# Provable Tempered Overfitting of
# Minimal Nets and Typical Nets

**Itamar Harel**
Technion
itamarharel01@gmail.com

**William M. Hoza**
The University of Chicago

**Gal Vardi**
Weizmann Institute of Science

**Itay Evron**
Technion

**Nathan Srebro**
Toyota Technological Institute at Chicago

**Daniel Soudry**
Technion

## Abstract

We study the overfitting behavior of fully connected deep Neural Networks (NNs) with binary weights fitted to perfectly classify a noisy training set. We consider interpolation using both the smallest NN (having the minimal number of weights) and a random interpolating NN. For both learning rules, we prove overfitting is tempered. Our analysis rests on a new bound on the size of a threshold circuit consistent with a partial function. To the best of our knowledge, ours are the first theoretical results on benign or tempered overfitting that: (1) apply to deep NNs, and (2) do not require a very high or very low input dimension.

## 1 Introduction

Neural networks (NNs) famously exhibit strong generalization capabilities, seemingly in defiance of traditional generalization theory. Specifically, NNs often generalize well empirically even when trained to interpolate the training data perfectly [97]. This motivated an extensive line of work attempting to explain the overfitting behavior of NNs, and particularly their generalization capabilities when trained to perfectly fit a training set with corrupted labels (*e.g.,* [5, 28, 57, 48]).

In an attempt to better understand the aforementioned generalization capabilities of NNs, Mallinar et al. [57] proposed a taxonomy of benign, tempered, and catastrophic overfitting. An algorithm that perfectly interpolates a training set with corrupted labels, *i.e.,* an interpolator, is said to have tempered overfitting if its generalization error is neither benign nor catastrophic — not optimal but much better than trivial. However, the characterization of overfitting in NNs is still incomplete, especially in *deep* NNs when the input dimension is neither very high nor very low. In this paper, we aim to understand the overfitting behavior of deep NNs in this regime.

We start by analyzing tempered overfitting in "min-size" NN interpolators, *i.e.,* whose neural layer widths are selected to minimize the total number of weights. The number of parameters in a model is a natural complexity measure in learning theory and practice. For instance, it is theoretically well understood that $L_1$ regularization in a sparse linear regression setting yields a sparse regressor. Practically, finding small-sized deep models is a common objective used in pruning (*e.g.,* [33]) and neural architecture search (*e.g.,* [54]). Recently, Manoj and Srebro [58] proved that the *shortest program* (Turing machine) that perfectly interpolates noisy datasets exhibits tempered overfitting, illustrating how a powerful model can avoid catastrophic overfitting by returning a min-size interpolator.

Furthermore, we study tempered overfitting in random ("typical") interpolators — NNs sampled uniformly from the set of parameters that perfectly fit the training set. Given a narrow teacher model and no label noise, Buzaglo et al. [13] recently proved that such typical interpolators, which may be *highly overparameterized*, generalize well. This is remarkable since these interpolators do not rely on

38th Conference on Neural Information Processing Systems (NeurIPS 2024).

explicit regularization or the implicit bias of any gradient algorithm. An immediate question arises — what kind of generalization behavior do typical interpolators exhibit in the presence of label noise? This is especially interesting in light of theoretical and empirical findings that typical NNs implement low-frequency functions [70, 83], while interpolating noisy training sets may require high frequencies.

For both the min-size and typical NN interpolators, we study the generalization behavior under an underlying *noisy* teacher model. We focus on deep NNs with binary weights and activations (similar NNs are used in resource-constrained environments; *e.g.,* [42]). Our analysis reveals that these models exhibit a tempered overfitting behavior that depends on the statistical properties of the label noise. For independent noise, in addition to an upper bound we also find a lower bound on the expected generalization error. Our results are illustrated in Figure 1 below, in which the yellow line in the right panel is similar to empirically observed linear behavior [e.g., 57, Figures 2, 3, and 6].

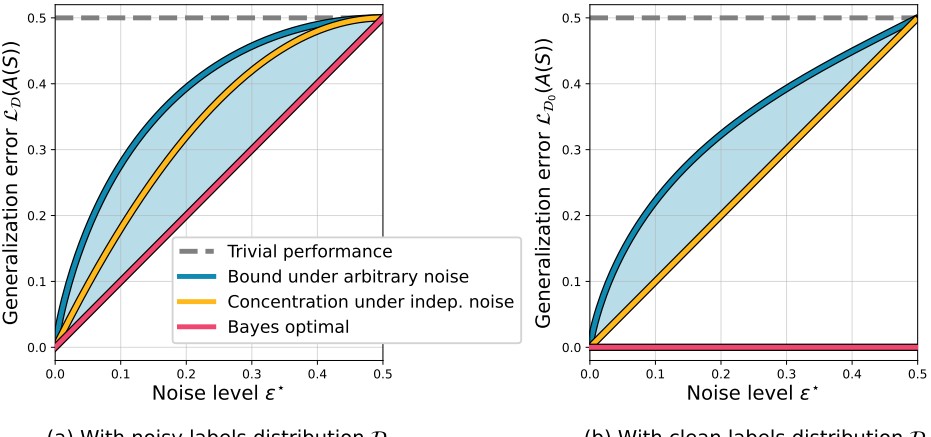

(a) With noisy labels distribution $\mathcal{D}$  (b) With clean labels distribution $\mathcal{D}_0$

Figure 1: **Types of overfitting behaviors.** Consider a binary classification problem of learning a realizable distribution $\mathcal{D}_0$. Let $\mathcal{D}$ be the distribution induced by adding an $\varepsilon^\star$-probability for a data point's label to be flipped relative to $\mathcal{D}_0$. Suppose a model is trained with data from $\mathcal{D}$. Then, assuming the classes are balanced, the trivial generalization performance is $0.5$ (in gray; *e.g.,* with a constant predictor). **Left.** Evaluating the model on $\mathcal{D}$, a Bayes-optimal hypothesis (in red) obtains a generalization error of $\varepsilon^\star$. For large enough training sets, our results (Section 4) dictate a tempered overfitting behavior illustrated above. For arbitrary noise, the error is approximately bounded by $1 - \varepsilon^{\star \varepsilon^\star} (1 - \varepsilon^\star)^{1-\varepsilon^\star}$ (blue). For independent noise, the error is concentrated around the tighter $2\varepsilon^\star (1 - \varepsilon^\star)$ (yellow). A similar figure was previously shown in Manoj and Srebro [58] for shortest-program interpolators. **Right.** Assuming independent noise, the left figure can be transformed into the error of the model on $\mathcal{D}_0$ (see Lemma A.9). The linear behavior in the independent setting (yellow) is similar to the behavior observed empirically in Mallinar et al. [57, Figures 2, 3, and 6].

The contributions of this paper are:

- Returning a min-size NN interpolator is a natural learning rule that follows the Occam's-razor principle. We show that this learning rule exhibits tempered overfitting (Section 4.1).

- We prove that overparameterized random NN interpolators typically exhibit tempered overfitting with generalization close to a min-size NN interpolator (Section 4.2).

- To the best of our knowledge, ours are the first theoretical results on benign or tempered overfitting that: (1) apply to deep NNs, and (2) do not require a very high or very low input dimension.

- The above results rely on a key technical result — datasets generated by a constant-size teacher model with label noise can be interpolated[1] using a NN of constant depth with threshold activations, binary weights, a width sublinear in $N$, and roughly $H(\varepsilon^\star) \cdot N$ weights, where $H(\varepsilon^\star)$ is the binary entropy function of the fraction of corrupted labels (Section 3).

---

[1]As long as it has no repeated data points with opposite labels. See our Def. 2.5 of consistent datasets.

## 2 Setting

**Notation.** We reserve bold lowercase characters for vectors, bold uppercase characters for matrices, and regular uppercase characters for random elements. We use $\log$ to denote the base 2 logarithm, and $\ln$ to denote the natural logarithm. For a pair of vectors $\underline{d} = (d_1, \ldots, d_L)$, $\underline{d}' = (d'_1, \ldots, d'_L) \in \mathbb{N}^L$ we denote $\underline{d} \leq \underline{d}'$ if for all $l \in [L]$, $d_l \leq d'_l$. We use $\oplus$ to denote the XOR between two binary $\{0, 1\}$ values, and $\odot$ to denote the Hadamard (elementwise) product between two vectors. We use $H(\mathcal{D})$ to denote the entropy of some distribution $\mathcal{D}$. Finally, we use $\text{Ber}(\varepsilon)$ for the Bernoulli distribution with parameter $\varepsilon$, and $H(\varepsilon)$ for its entropy, which is the binary entropy function.

### 2.1 Model: Fully connected threshold NNs with binary weights

Similarly to Buzaglo et al. [13], we define the following model.

**Definition 2.1** (Binary threshold networks). For a depth $L$, widths $\underline{d} = (d_1, \ldots, d_L)$, input dimension $d_0$, a scaled-neuron fully connected binary threshold NN, or binary threshold network, is a mapping $\boldsymbol{\theta} \mapsto h_{\boldsymbol{\theta}}$ such that $h_{\boldsymbol{\theta}} : \{0, 1\}^{d_0} \to \{0, 1\}^{d_L}$, parameterized by

$$\boldsymbol{\theta} = \left\{ \mathbf{W}^{(l)}, \mathbf{b}^{(l)}, \boldsymbol{\gamma}^{(l)} \right\}_{l=1}^{L} ,$$

where for every layer $l \in [L]$,

$$\mathbf{W}^{(l)} \in \mathcal{Q}_l^W = \{0, 1\}^{d_l \times d_{l-1}} , \ \boldsymbol{\gamma}^{(l)} \in \mathcal{Q}_l^{\gamma} = \{-1, 0, 1\}^{d_l} , \ \mathbf{b}^{(l)} \in \mathcal{Q}_l^b = \{-d_{l-1} + 1, \ldots, d_{l-1}\}^{d_l} .$$

This mapping is defined recursively as $h_{\boldsymbol{\theta}}(\mathbf{x}) = h^{(L)}(\mathbf{x})$ where

$$h^{(0)}(\mathbf{x}) = \mathbf{x} ,$$

$$\forall l \in [L] \quad h^{(l)}(\mathbf{x}) = \mathbb{I} \left\{ \left( \boldsymbol{\gamma}^{(l)} \odot \left( \mathbf{W}^{(l)} h^{(l-1)}(\mathbf{x}) \right) + \mathbf{b}^{(l)} \right) > \mathbf{0} \right\} .$$

We denote the number of weights by $w(\underline{d}) = \sum_{l=1}^{L} d_l d_{l-1}$, and the total number of neurons by $n(\underline{d}) = \sum_{l=1}^{L} d_l$. The total number of parameters in such a NN is $M(\underline{d}) = w(\underline{d}) + 2n(\underline{d})$. We denote the set of functions representable as binary networks of widths $\underline{d}$ by $\mathcal{H}_{\underline{d}}^{\text{BTN}}$ and their corresponding parameter space by $\Theta^{\text{BTN}}(\underline{d})$.

*Remark* 2.2. Our generalization results are for the above formulation of neuron scalars $\boldsymbol{\gamma}$, *i.e.,* ternary scaling *before* the activation. However, we could have derived similar results if, instead, we changed the scale $\boldsymbol{\gamma}$ to appear *after* the activation and also adjusted the range of the biases (see Appendix G). Although we chose the former for simplicity, the latter is similar to the ubiquitous phenomenon in neuroscience known as "Dale's Law" [82]. This law, in a simplified form, means that all outgoing synapses of a neuron have the same effect, *e.g.,* are all excitatory (positive) or all inhibitory (negative).

*Remark* 2.3 (Simple counting argument). Let $\underline{d}_{\max} \triangleq \max\{d_1, \ldots, d_{L-1}\}$ be the maximal hidden-layer width. Then, combinatorially, it holds that

$$\log \underbrace{\left| \mathcal{H}_{\underline{d}}^{\text{BTN}} \right|}_{\text{\# hypotheses}} \leq \log \underbrace{\left| \Theta^{\text{BTN}}(\underline{d}) \right|}_{\substack{\text{\# parameter} \\ \text{assignments}}} \leq \underbrace{w(\underline{d})}_{\text{\# weights}} + \underbrace{n(\underline{d})}_{\text{\# neurons}} \left( \log(3) + \log \underbrace{(2\underline{d}_{\max})}_{\substack{\text{maximal} \\ \text{quantization}}} \right) .$$

This implies, using classical PAC bounds [75], that the sample complexity of learning with the *finite* hypothesis class $\mathcal{H}_{\underline{d}}^{\text{BTN}}$ is $O(w(\underline{d}) + n(\underline{d}) \log \underline{d}_{\max})$ (a more refined bound on $\left| \mathcal{H}_{\underline{d}}^{\text{BTN}} \right|$ is given in Lemma F.1). In Section 4 we show how this generalization bound can be improved in our setting.

### 2.2 Data model: A teacher network and label-flip noise

**Data distribution.** Let $\mathcal{X} = \{0, 1\}^{d_0}$ and let $\mathcal{D}$ be some joint distribution over a finite sample space $\mathcal{X} \times \{0, 1\}$ of features and labels.

**Assumption 2.4** (Teacher assumption). We assume a "teacher NN" $h^{\star}$ generating the labels. A label flipping noise is then added with a noise level of $\varepsilon^{\star} = \mathbb{P}_{(X,Y) \sim \mathcal{D}}(Y \neq h^{\star}(X))$, or equivalently

$$Y \oplus h^{\star}(X) \sim \text{Ber}(\varepsilon^{\star}) .$$

The label noise is *independent* when $Y \oplus h^{\star}(X)$ is independent of the features $X$ (in Section 4 it leads to stronger generalization results compared to ones for arbitrary noise).

### 2.3 Learning problem: Classification with interpolators

We consider the problem of binary classification over a training set $S = \{(\mathbf{x}_i, y_i)\}_{i=1}^{N}$ with $N$ data points, sampled from the noisy joint distribution $\mathcal{D}$ described above. We always assume that $S$ is sampled i.i.d., and therefore, with some abuse of notation, we use $\mathcal{D}(S) = \mathcal{D}^N(S) = \prod_{i=1}^{N} \mathcal{D}(\mathbf{x}_i, y_i)$. For a hypothesis $h : \mathcal{X} \to \{0, 1\}$, we define the risk, *i.e.,* the generalization error w.r.t. $\mathcal{D}$, as

$$\mathcal{L}_{\mathcal{D}}(h) \triangleq \mathbb{P}_{(X,Y)\sim\mathcal{D}}(h(X) \neq Y) .$$

We also define the empirical risk, *i.e.,* the training error,

$$\mathcal{L}_S(h) \triangleq \frac{1}{N} \sum_{n=1}^{N} \mathbb{I}\{h(\mathbf{x}_n) \neq y_n\} .$$

We say a hypothesis is an *interpolator* if $\mathcal{L}_S(h) = 0$.

In this paper, we are specifically interested in *consistent* datasets that can be perfectly fit. This is formalized in the following definition.

**Definition 2.5** (Consistent datasets). A dataset $S = \{(\mathbf{x}_i, y_i)\}_{i=1}^{N}$ is consistent if

$$\forall i, j \in [N] \ \ \mathbf{x}_i = \mathbf{x}_j \implies y_i = y_j .$$

Motivated by modern NNs which are often extremely overparameterized, we are interested in the generalization behavior of interpolators, *i.e.,* models that fit a consistent training set perfectly. Specifically, we consider Framework 1. While this framework is general enough to fit any minimal training error models, we shall be interested in the generalization of $A(S)$ in cases where the training set is most likely consistent (Def. 2.5).

---

**Framework 1** Learning interpolators

---

**Input:** A training set $S$.
**Algorithm:**
    **if** $S$ is consistent:
        **return** an interpolator $A(S) = h$ (such that $\mathcal{L}_S(h) = 0$)
    **else:**
        **return** an arbitrary hypothesis $A(S) = h$ (*e.g.,* $h(\mathbf{x}) = 0, \forall \mathbf{x}$)

---

In Section 4, we analyze the generalization of two learning rules that fall under this framework: (1) learning min-size NN interpolators and (2) sampling random NN interpolators. Our analysis reveals a tempered overfitting behavior in both cases.

## 3 Interpolating a noisy training set

Our main generalization results rely on a key technical result, which shows how to memorize any consistent training set generated according to our noisy teacher model. We prove that the memorizing "student" NN can be small enough to yield meaningful generalization bounds in the next sections.

We begin by noticing that under a teacher model $h^\star$ (Assumption 2.4), the labels of a consistent dataset $S$ (Def. 2.5) can be decomposed as

$$\forall i \in [N] \ \ y_i = h^\star(\mathbf{x}_i) \oplus f(\mathbf{x}_i) , \tag{1}$$

where $f : \{0, 1\}^{d_0} \to \{0, 1\}$ indicates a label flip in the $i^{\text{th}}$ example, and can be defined arbitrarily for $\mathbf{x} \notin S$. Motivated by this observation, we now show an upper bound for the dimensions of a network interpolating $S$, by bounding the dimensions of an NN implementing an arbitrary "partial" function $f$ defined on $N$ points.

**Theorem 3.1** (Memorizing the label flips). *Let* $f\colon \{0,1\}^{d_0} \rightarrow \{0,1,\star\}$ *be any function.*[2] *Let* $N = |f^{-1}(\{0,1\})|$ *and* $N_1 = |f^{-1}(1)|$. *There exists a depth-*$14$ *binary threshold network* $\tilde{h}\colon \{0,1\}^{d_0} \rightarrow \{0,1\}$, *with widths* $\underline{\tilde{d}}$, *satisfying the following.*

1. *$\tilde{h}$ is consistent with $f$, i.e., for every $\mathbf{x} \in \{0,1\}^{d_0}$, if $f(\mathbf{x}) \in \{0,1\}$, then $\tilde{h}(\mathbf{x}) = f(\mathbf{x})$.*

2. *The total number of weights in $\tilde{h}$ is at most $(1 + o(1)) \cdot \log \binom{N}{N_1} + \mathrm{poly}(d_0)$. More precisely,*

$$w\left(\underline{\tilde{d}}\right) = \log \binom{N}{N_1} + \left(\log \binom{N}{N_1}\right)^{3/4} \cdot \mathrm{polylog}\, N + O(d_0^2 \cdot \log N).$$

3. *Every layer of $\tilde{h}$ has width at most $(\log \binom{N}{N_1})^{3/4} \cdot \mathrm{poly}(d_0)$. More precisely,*

$$\underline{\tilde{d}}_{\max} = \left(\log \binom{N}{N_1}\right)^{3/4} \cdot \mathrm{polylog}\, N + O(d_0 \cdot \log N).$$

The main takeaway from Theorem 3.1 is that label flips can be memorized with networks with a number of parameters that is optimal in the leading order $N \cdot \mathcal{L}_S(h^\star)$, i.e., not far from the minimal information-theoretical value. The proofs for this section are given in Appendix D.

**Proof idea.** Denote $S = f^{-1}(\{0,1\})$. We employ established techniques from the pseudorandomness literature to construct an efficient *hitting set generator* (HSG)[3] for the class of all conjunctions of literals. The HSG definition implies that there exists a seed on which the generator outputs a truth table that agrees with $f$ on $S$. The network $\tilde{h}$ computes any requested bit of that truth table.

*Remark* 3.2 (Dependence on $d_0$). In Appendix E we show that the $O\left(d_0^2 \cdot \log N\right)$ term is nearly tight, yet it can be relaxed when using some closely related NN architectures. For example, with a single additional layer of width $\Omega\left(\sqrt{d_0} \cdot \log N\right)$ with ternary weights in the first layer, *i.e.,* $\mathcal{Q}_1^W = \{-1, 0, 1\}$ instead of $\{0, 1\}$, the $O\left(d_0^2 \cdot \log N\right)$ term of Theorem 3.1 can be improved to $O\left(d_0^{3/2} \cdot \log N + d_0 \cdot \log^3 N\right)$.

Next, with the bound on the dimensions of a NN implementing $f$, we can bound the dimensions of a min-size interpolating NN by bounding the dimensions of a NN implementing the XOR of $\tilde{h}$ and $h^\star$.

**Lemma 3.3** (XOR of two NNs). *Let $h_1, h_2$ be two binary NNs with depths $L_1 \leq L_2$ and widths $\underline{d}^{(1)}, \underline{d}^{(2)}$, respectively. Then, there exists a NN $h$ with depth $L_{\mathrm{XOR}} \triangleq L_2 + 2$ and widths*

$$\underline{d}_{\mathrm{XOR}} \triangleq \left(d_1^{(1)} + d_1^{(2)}, \, \ldots, \, d_{L_1}^{(1)} + d_{L_1}^{(2)}, \, d_{L_1+1}^{(2)} + 1, \, \ldots, \, d_{L_2}^{(2)} + 1, \, 2, \, 1\right),$$

*such that for all inputs $\mathbf{x} \in \{0,1\}^{d_0}$, $h(\mathbf{x}) = h_1(\mathbf{x}) \oplus h_2(\mathbf{x})$.*

Combining Theorem 3.1 and Lemma 3.3 results in the following corollary.

**Corollary 3.4** (Memorizing a consistent dataset). *For any teacher $h^\star$ of depth $L^\star$ and dimensions $\underline{d}^\star$ and any consistent training set $S$ generated from it, there exists an interpolating NN $h$ (i.e., $\mathcal{L}_S(h) = 0$) of depth $L = \max\{L^\star, 14\} + 2$ and dimensions $\underline{d}$, such that the number of weights is*

$$w(\underline{d}) \leq w(\underline{d}^\star) + N \cdot H(\mathcal{L}_S(h^\star)) + 2n(\underline{d}^\star) N^{3/4} H(\mathcal{L}_S(h^\star))^{3/4} \mathrm{polylog} N$$
$$+ O(d_0(d_0 + n(\underline{d}^\star)) \cdot \log N)$$

*and the maximal width is*

$$\underline{d}_{\max} \leq \underline{d}_{\max}^\star + N^{3/4} \cdot H(\mathcal{L}_S(h^\star))^{3/4} \cdot \mathrm{polylog}(N) + O(d_0 \cdot \log(N)).$$

**Proof idea.** We explicitly construct a NN with the desired properties. We can choose a subset of neurons to implement the teacher NN and another subset to implement the NN memorizing the label flips. Furthermore, we zero the weights between the two subsets. Two additional layers compute the XOR of the outputs, thus yielding the labels as in (1). This is illustrated in Figure 2.

---

[2]When $f(\mathbf{x}) = \star$, the interpretation is that $f$ is "undefined" on $\mathbf{x}$, *i.e.,* $f$ is a "partial" function.

[3]A variant of the *pseudorandom generator* (PRG) concept.

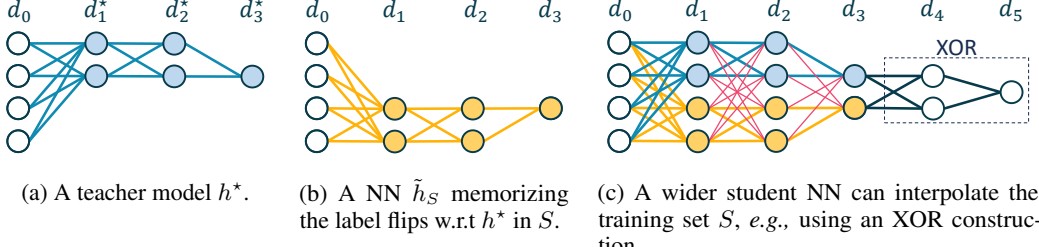

(a) A teacher model $h^\star$.

(b) A NN $\tilde{h}_S$ memorizing the label flips w.r.t $h^\star$ in $S$.

(c) A wider student NN can interpolate the training set $S$, *e.g.,* using an XOR construction.

Figure 2: **Interpolating a dataset.** To memorize the training set, we use a subset of the parameters to match those of the teacher and another subset to memorize the noise (label flips). Then, we "merge" these subsets to interpolate the noisy training set. In our figure, (1) blue edges represent weights identical to the teacher's; (2) yellow edges memorize the noise; (3) red edges are set to 0; and two additional layers implement the XOR between outputs, thus memorizing the training set.

# 4 Tempered overfitting of min-size and random interpolators

In this section, we provide our main results on the overfitting behavior of interpolating NNs. We consider min-size NN interpolators and random NN interpolators. For both learning rules, we prove tempered overfitting. Namely, we show that the test performance of the learned interpolators is not much worse than the Bayes optimal error.

First, for the sake of readability, let us define the marginal peak probability of the distribution.

**Definition 4.1** (Peak marginal probability). $\mathcal{D}_{\max} \triangleq \max_{\mathbf{x} \in \mathcal{X}} \mathbb{P}_{(X,Y) \sim \mathcal{D}} (X = \mathbf{x})$.

Our results in this section focus on cases where the number of training samples is $N = \omega\left(d_0^2 \log d_0\right)$ and $N = o\left(1/\sqrt{\mathcal{D}_{\max}}\right)$. In such regimes, the data consistency probability is high[4] and our bounds are meaningful. Note that given the binarization of the data, $N = o\left(1/\sqrt{\mathcal{D}_{\max}}\right)$ implies an exponential upper bound of $N = o\left(2^{d_0/2}\right)$, achieved by the uniform distribution, *i.e.,* when $\mathcal{D}_{\max} = 2^{-d_0}$. Due to the exponential growth of the sample space w.r.t. the input dimension, we find this assumption to be reasonable. Also, $N = \omega\left(d_0^2 \log d_0\right)$ implies that the input dimension cannot be arbitrarily large, but may still be non trivially small (see comparison to previous work in Section 5).

## 4.1 Min-size interpolators

We consider min-size NN interpolators of a fixed depth, *i.e.,* networks with the smallest number of weights for a certain depth that interpolate a given training set. In realizable settings, achieving good generalization performance by restricting the number of parameters in the learned interpolating model is a natural and well-understood approach. Indeed, in such cases, generalization follows directly from standard VC-dimension bounds [4, 75]. However, when interpolating *noisy* data, the size of the returned model increases with the number of samples (in order to memorize the noise; see *e.g.,* Vardi et al. [88]), making it challenging to guarantee generalization. In what follows, we prove that even when interpolating noisy data, min-size NNs exhibit good generalization performance.

**Learning rule: Min-size NN interpolator.** Given a consistent dataset $S$ and a fixed depth $L$, a min-size NN interpolator, or min-#weights interpolator, is a binary threshold network $h$ (see Def. 2.1) that achieves $\mathcal{L}_S(h) = 0$ using a minimal number of weights. Recall that $w(\underline{d}) = \sum_{l=1}^{L} d_l d_{l-1}$ and define the *minimal* number of weights required to implement a given hypothesis $h$,

$$w_L(h) \triangleq \min_{\underline{d} \in \mathbb{N}^L} w(\underline{d}) \text{ s.t. } h \in \mathcal{H}_{\underline{d}}^{\mathrm{BTN}}.$$

The learning rule is then defined as

$$A_L(S) \in \mathrm{argmin}_h w_L(h) \text{ s.t. } \mathcal{L}_S(h) = 0.$$

---

[4] $N = o\left(1/\sqrt{\mathcal{D}_{\max}}\right)$ implies a rough bound on the consistency probability via the union bound. For example, when the marginal of $X$ under $\mathcal{D}$ is uniform ($\mathcal{D}_{\max} = 1/|\mathcal{X}|$), and the inconsistency probability corresponds to the well-known birthday problem.

**Theorem 4.2** (Tempered overfitting of min-size NN interpolators). *Let $\mathcal{D}$ be a distribution induced by a noisy teacher of depth $L^\star$, widths $\underline{d}^\star$, $n(\underline{d}^\star)$ neurons, and a noise level of $\varepsilon^\star < 1/2$ (Assumption 2.4). There exists $c > 0$ such that the following holds. Let $S \sim \mathcal{D}^N$ be a training set such that $N = \omega\left(n\left(\underline{d}^\star\right)^4 H\left(\varepsilon^\star\right)^3 \log\left(n\left(\underline{d}^\star\right)\right)^c + d_0^2 \log d_0\right)$ and $N = o(\sqrt{1/\mathcal{D}_{\max}})$. Then, for any fixed depth $L \geq \max\{L^\star, 14\} + 2$, the generalization error of the min-size depth-L NN interpolator satisfies the following.*

- *Under arbitrary label noise,* $\qquad\qquad\qquad \mathbb{E}_S\left[\mathcal{L}_\mathcal{D}\left(A_L\left(S\right)\right)\right] \leq 1 - 2^{-H(\varepsilon^\star)} + o\left(1\right).$

- *Under independent label noise,* $\qquad\qquad \left|\mathbb{E}_S\left[\mathcal{L}_\mathcal{D}\left(A_L\left(S\right)\right)\right] - 2\varepsilon^\star\left(1-\varepsilon^\star\right)\right| = o\left(1\right).$

Here, $o\left(1\right)$ indicates terms that become insignificant when the number of samples $N$ is large. We illustrate these behaviors in Figure 1. Moreover, we discuss these results and the proof idea in Section 4.3 after presenting the corresponding results for posterior sampling. The complete proof with detailed characterization of the $o(1)$ terms is given in Appendix F.1.

## 4.2 Random NN interpolators (posterior sampling)

Recent empirical [87, 20] and theoretical [13] works have shown that, somewhat surprisingly, randomly sampled deep NNs that interpolate a training set often generalize well. We now turn to analyzing such random interpolators under our teacher assumption and noisy labels (Assumption 2.4). As with min-size NN interpolators, our analysis here reveals a tempered overfitting behavior.

**Prior distribution.** A distribution over parameters induces a prior distribution over hypotheses by

$$\mathcal{P}\left(h\right) = \mathbb{P}_{\boldsymbol{\theta}}\left(h_{\boldsymbol{\theta}} = h\right).$$

We focus on the prior induced by the *uniform prior* over the parameters of binary threshold networks. Specifically, for a fixed depth $L$ and dimensions $\underline{d}$, we consider $\boldsymbol{\theta} \sim \text{Uniform}\left(\Theta^{\text{BTN}}\left(\underline{d}\right)\right)$. In other words, to generate $h \sim \mathcal{P}$, each weight, bias, and neuron scalar in the NN is sampled independently and uniformly from its respective domain.

**Learning rule: Posterior sampling.** For any training set $S$, denote the probability to sample an interpolating NN by $p_S \triangleq \mathcal{P}\left(\mathcal{L}_S\left(h\right) = 0\right)$. When $p_S > 0$, define the posterior distribution $\mathcal{P}_S$ as

$$\mathcal{P}_S\left(h\right) \triangleq \mathcal{P}\left(h \mid \mathcal{L}_S\left(h\right) = 0\right) = \frac{\mathcal{P}(h)}{p_S}\mathbb{I}\left\{\mathcal{L}_S\left(h\right) = 0\right\}. \tag{2}$$

When $p_S = 0$, use an arbitrary $\mathcal{P}_S$. Finally, the posterior sampling rule is $A_{\underline{d}}\left(S\right) \sim \mathcal{P}_S$.

*Remark* 4.3 (Hypothesis expressivity). The following result requires that the student NN is large enough to interpolate *any* consistent $S$ (see Corollary 3.4), thus, $p_S > 0$ and $\mathcal{P}_S$ is defined as in (2).

**Theorem 4.4** (Tempered overfitting of random NN interpolators). *Let $\mathcal{D}$ be a distribution induced by a noisy teacher of depth $L^\star$, widths $\underline{d}^\star$, $n(\underline{d}^\star)$ neurons, and a noise level of $\varepsilon^\star < 1/2$ (Assumption 2.4). There exists a constant $c > 0$ such that the following holds. Let $S \sim \mathcal{D}^N$ be a training set such that $N = \omega\left(n\left(\underline{d}^\star\right)^4 \log\left(n\left(\underline{d}^\star\right)\right)^c + d_0^2 \log d_0\right)$ and $N = o(\sqrt{1/\mathcal{D}_{\max}})$. Then, for any student network of depth $L \geq \max\{L^\star, 14\} + 2$ and widths $\underline{d} \in \mathbb{N}^L$ holding*

$$\forall l = 1, \ldots, L^\star - 1 \quad d_l \geq d_l^\star + N^{3/4} \cdot \left(\log N\right)^c + c \cdot d_0 \cdot \log\left(N\right), \tag{3}$$

*the generalization error of posterior sampling satisfies the following.*

- *Under arbitrary label noise,*

$$\mathbb{E}_{S, A_{\underline{d}}(S)}\left[\mathcal{L}_\mathcal{D}\left(A_{\underline{d}}\left(S\right)\right)\right] \leq 1 - 2^{-H(\varepsilon^\star)} + O\left(\frac{n\left(\underline{d}\right) \cdot \log\left(d_{\max} + d_0\right)}{N}\right).$$

- *Under independent label noise,*

$$\left|\mathbb{E}_{S, A_{\underline{d}}(S)}\left[\mathcal{L}_\mathcal{D}\left(A_{\underline{d}}\left(S\right)\right)\right] - 2\varepsilon^\star\left(1-\varepsilon^\star\right)\right| \leq O\left(\sqrt{\frac{n\left(\underline{d}\right) \cdot \log\left(d_{\max} + d_0\right)}{N}}\right).$$

The proof and a detailed description of the error terms are given in Appendix F.2.

Remarkably, note that the interpolating NN in the theorem might be highly overparameterized, and that for such NNs good generalization is not guaranteed by standard generalization bounds [4, 75]. This theorem complements a similar result by Buzaglo et al. [13] for the realizable setting.

### 4.3 Discussion

The overfitting behaviors described in this section are illustrated in Figure 1.

**Proof idea.** We extend the information-theoretical generalization bounds from [58] to this paper's setting in which label collisions in the datasets have a non-zero probability. In particular, we bound the interpolator's complexity from below by the mutual information between the model and the training set. Since the model is interpolating, we can further bound the mutual information by a quantity dependent on the population error. From the other direction, we bound the model's complexity from above by (1) its size in the min-size setting of Section 4.1, and (2) by the negative log interpolation probability for the posterior sampling of Section 4.2. Together with Corollary 3.4 we obtain the bounds above on the expected generalization error.

In Figure 2 we illustrated the construction of a memorizing network used to bound the complexity of the min-size interpolator. In the following Figure 3 we illustrate how the interpolation probability $p_S$ can be bounded to induce a meaningful generalization bound.

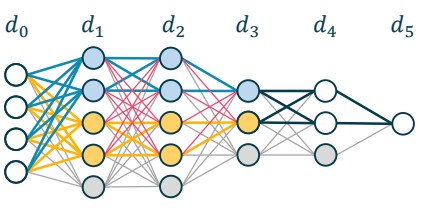

Figure 3: **Interpolating a dataset with an overparameterized student.** We build on the construction from Figure 2 that memorizes a dataset using a subset of the parameters (blue, yellow, and red edges). Then, redundant neurons (gray) can be effectively ignored by setting their neuron scaling parameters ($\gamma$) to 0, leaving the redundant weights (gray edges) unconstrained. Thus, the interpolation probability $p_S$ can be bounded by a quantity exponentially decaying in the number of neurons $n\left(\underline{d}\right)$ rather than in the number of weights $w\left(\underline{d}\right) = \omega\left(N\right)$.

Following Remark 3.2, the assumption $N = \omega\left(d_0^2 \log d_0\right)$ can be relaxed in some related architectures. For example, with a single additional layer of width $O\left(\sqrt{d_0} \cdot \log N\right)$ and ternary weights in the first layer $\mathcal{Q}_1^W = \{-1, 0, 1\}$, the requirement can be relaxed to $N = \omega\left(d_0^{3/2} \log d_0\right)$.

*Remark* 4.5 (Higher weight quantization). The bounds in the arbitrary noise setting can easily be extended to NNs with higher quantization levels. For example, letting $\hat{\mathcal{Q}}_l^W$ such that $\left|\mathcal{Q}_l^W\right| = Q$ and $\{0, 1\} \subseteq \mathcal{Q}_l^W$, under the appropriate assumptions, we get that

$$\mathbb{E}_{(S, A(S))}\left[\mathcal{L}_\mathcal{D}\left(A\left(S\right)\right)\right] \lesssim 1 - Q^{-H(\varepsilon^\star)},$$

which is a meaningful bound for noise levels $\varepsilon^\star \leq \varepsilon\left(Q\right)$ for some $\varepsilon\left(Q\right) < 1/2$.[5] Tighter results would require utilizing the additional quantization levels to achieve smaller dimensions of the interpolating network, and are left to future work.

## 5 Related work

**Benign and tempered overfitting.** The benign overfitting phenomenon has been extensively studied in recent years. Previous works analyzed the conditions in which benign overfitting occurs in linear regression [34, 11, 5, 65, 67, 21, 47, 93, 86, 98, 44, 90, 19, 3, 76, 31], kernel regression [51, 61, 53, 57, 72, 10, 60, 8, 50, 99, 6], and linear classification [18, 91, 14, 66, 64, 76, 52, 85, 92, 25]. Moreover, several works proved benign overfitting in classification using nonlinear NNs [28, 29, 15, 49, 94, 95, 62, 48, 30, 46]. All the aforementioned benign overfitting results require high-dimensional settings, namely, the input dimension is larger than the number of training samples.

---

[5]Specifically, $\varepsilon\left(Q\right)$ such that $1 - Q^{-H(\varepsilon(Q))} \leq 1/2$.

Mallinar et al. [57] suggested the taxonomy of benign, tempered, and catastrophic overfitting, which we use in this work. They demonstrated empirically that nonlinear NNs in classification tasks exhibit tempered overfitting. As mentioned in the introduction, our theoretical results for the independent noise case closely resemble these empirical findings (see Figure 1). Tempered overfitting in kernel ridge regression was theoretically studied in Mallinar et al. [57], Zhou et al. [99], Barzilai and Shamir [6]. In univariate ReLU NNs (namely, for input dimension 1), tempered overfitting was obtained for both classification [48] and regression [43]. Manoj and Srebro [58] proved tempered overfitting for a learning rule returning short programs in some programming language. Finally, tempered overfitting is well understood for the 1-nearest-neighbor learning rule, where the asymptotic risk is roughly twice the Bayes risk [23].

**Circuit complexity.** Theorem 3.1 (our NN for memorizing label flips) is in a similar spirit as several prior theorems in the area of *circuit complexity*. For example, Lupanov famously proved that every function $f \colon \{0,1\}^{d_0} \to \{0,1\}$ can be computed by a circuit consisting of $(1 + o(1)) \cdot 2^{d_0}/d_0$ many AND/OR/NOT gates, where the AND/OR gates have fan-in two [55]. Lupanov's bound, which is tight [77], is analogous to Theorem 3.1, because a NN can be considered a type of circuit.

Even more relevant is a line of work that analyzes the circuit complexity of an arbitrary partial function $f \colon \{0,1\}^{d_0} \to \{0,1,\star\}$ with a given domain size $N$ and a given number of 1-inputs $N_1$, similar to the setup of Theorem 3.1. See Jukna's textbook for an overview [45, Section 1.4.2]. We highlight the work of Chashkin, who showed that every such function can be computed by a circuit (of unbounded depth and bounded fan-in) with $(1 + o(1)) \cdot \frac{\log\binom{N}{N_1}}{\log\log\binom{N}{N_1}} + O(d_0)$ gates [17].

To the best of our knowledge, prior to our work, nothing analogous to Chashkin's theorem [17] was known regarding constant-depth threshold networks. It is conceivable that one could adapt Chashkin's construction [17] to the binary threshold network setting as a method of proving Theorem 3.1, but our proof of Theorem 3.1 uses a different approach. Our proof relies on shallow threshold networks computing *k-wise independent generators* [36] and an *error-reduction* technique that was developed in the context of space-bounded derandomization [38], among other ingredients.

**Memorization.** Our construction shows how noisy data can be interpolated using a small threshold NN with binary weights. It essentially requires memorizing the noisy examples. The task of memorization, namely, finding a smallest NN that allows for interpolation of arbitrary data points, has been extensively studied in recent decades. Memorization of $N$ arbitrary points in general position in $\mathbb{R}^d$ with a two-layer NN can be achieved using $O\left(\lceil \frac{N}{d} \rceil\right)$ hidden neurons [7, 81, 12]. Memorizing arbitrary $N$ points, even if they are not in general position, can be done using two-layer networks with $O(N)$ neurons [41, 74, 40, 97]. With three-layer networks, $O(\sqrt{N})$ neurons suffice, but the number of parameters is still linear in $N$ [39, 96, 89, 71]. Using deeper networks allows for memorization with a sublinear number of parameters [68, 88]. For example, memorization with networks of depth $\sqrt{N}$ requires only $\tilde{O}(\sqrt{N})$ parameters [88]. However, we note that in the aforementioned results, the number of quantization levels is not constant, namely, the number of bits in the representation of each weight depends on $N$.[6] Moreover, even in the sublinear constructions of [68, 88], the number of bits required to represent the network is $\omega(N)$. As a result, in this work we cannot rely on these constructions to obtain meaningful bounds.

**Posterior sampling and guess and check.** The generalization of random interpolating neural networks has previously been studied, both empirically and theoretically [87, 63, 84, 20, 13]. Theisen et al. [84] studied the generalization of interpolating random linear and random features classifiers. Valle-Perez et al. [87], Mingard et al. [63] considered the Gaussian process approximation to random NNs which typically requires networks with infinite width. Buzaglo et al. [13] provided a method to obtain generalization results for quantized random NNs of general architectures — possibly deep and with finite width, under the assumption of a narrow teacher model. A variant of this approach was used to prove our generalization results of posterior sampling, with the XOR network (Lemma 3.3) used in the role of the teacher.

---

[6]We note that in most papers, the required number of quantization levels is implicit in the constructions, and is not discussed explicitly.

# 6 Extensions, limitations, and future work

In this work, we focused on binary (fully connected) threshold networks of depth $L \geq 16$ (Section 2.1) with binary input features (Section 2.2), for which we were able to derive nontrivial generalization bounds.

Our results can be extended with simple modifications to derive bounds in other settings. For instance, to NNs with higher weight quantization (see Remark 4.5), or to ReLU networks (since any threshold network with binary weights can be computed by a not-much-larger ReLU network with a constant quantization level). Unfortunately, without more sophisticated arguments these extensions result in looser generalization bounds. The "bottleneck" of our approach is the reliance on (nearly) tight bounds on the widths of interpolating NNs.

Extending the results to other architectures (*e.g.,* CNNs or fully connected without neuron scaling) and other quantization schemes (*e.g.,* floating point representations) will mainly require utilizing their specific structure to derive tighter bounds on the complexity (*e.g.,* number of weights or number of bits) needed to interpolate consistent datasets. Furthermore, our bounds require the depth of the networks to be at least 16, and the width to be $\omega\left(N^{3/4}\right)$, which might be deemed impractical for real datasets.[7] The key to alleviating these requirements is, again, obtaining tighter complexity results.

Our paper focused on consistent training sets (Def. 2.5), in order to allow perfect interpolation. Realistically, models do not always perfectly interpolate the training set, and therefore it is interesting to find generalization bounds for non-interpolating models, depending on the training error. In addition, it is interesting to relate the generalization to the training *loss*, and not just to the training accuracy. Such extensions will require either broadening our generalization results or deriving new ones.

## Acknowledgments and Disclosure of Funding

We thank Alexander Chashkin for generously providing English-language expositions of some results from his work [17] as well as some results from Lupanov's work [56] (personal communication). The research of DS was Funded by the European Union (ERC, A-B-C-Deep, 101039436). Views and opinions expressed are however those of the author only and do not necessarily reflect those of the European Union or the European Research Council Executive Agency (ERCEA). Neither the European Union nor the granting authority can be held responsible for them. DS also acknowledges the support of the Schmidt Career Advancement Chair in AI. GV is supported by research grants from the Center for New Scientists at the Weizmann Institute of Science, and the Shimon and Golde Picker – Weizmann Annual Grant. Part of this work was done as part of the NSF-Simons funded Collaboration on the Mathematics of Deep Learning. NS was partially supported by the NSF TRIPOD Institute on Data Economics Algorithms and Learning (IDEAL) and an NSF-IIS award.

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

# A Preliminaries and Auxiliary Results

## A.1 Preliminaries

Before moving to the proofs of the main results, we recall and introduce some notation that will be used throughout the supplementary material.

**Notation.** We denote a (possibly random) learning algorithm by $A(S)$, and assume that it takes values in some hypothesis class $\mathcal{H}$. We use $\mathcal{D}$ to denote the joint distribution over a finite sample space $\mathcal{X} \times \{0, 1\}$ of the features and labels, $\nu$ to denote the marginal distribution of the algorithm, and $p$ to denote the joint distribution of a training set $S \sim \mathcal{D}^N$ and the algorithm $A(S)$. Specifically, the training set is a random element

$$S = \{(X_1, Y_1), \ldots, (X_N, Y_N)\} \sim \mathcal{D}^N$$

where $(X_i, Y_i)$ is reserved for the $i$-th example in $S$. That is $(X_i, Y_i)$ is always a sample in $S$, whereas $(X, Y)$ is used to denote a data point which is independent of $S$. We use $d\mathcal{D}(x, y)$, $d\nu(h)$ and $dp(s, h) = dp(\{(x_1, y_1), \ldots, (x_N, y_N)\}, h)$ to denote the corresponding probability mass functions. With some abuse of notation, we use $d\mathcal{D}(x)$ for the probability mass function of the marginal of $\mathcal{D}$ over $\mathcal{X}$

$$d\mathcal{D}(x) = \mathbb{P}_{(X,Y)\sim\mathcal{D}}(X = x) ,$$

and $dp((x_1, y_1), h)$ for the marginal of the joint probability of a single point from $S$ and the output of the algorithm, *i.e.,*

$$dp((x_1, y_1), h) = \mathbb{P}_{(S, A(S))\sim p}(X_1 = x_1, Y_1 = y_1, A(S) = h) .$$

Similarly, we use $dp(x_1, h)$, $dp(y_1 \mid x_1, h)$, etc., for the probability mass functions of the appropriate marginal and conditional distributions.

**Interpolating algorithm.** In order to simplify the analysis, we introduce a framework of interpolation learning related to the one introduced in Framework 1.

Let $\tilde{A}(S)$ be a learning rule satisfying Framework 1, and let $\star$ be some arbitrary token distinct from any hypothesis the algorithm may produce. We define a modified learning rule $A(S)$[8] such that

- If $S$ is inconsistent then $A(S) = \star$.

- Otherwise, if $S$ is consistent then $A(S) = \tilde{A}(S)$, so in particular $\mathcal{L}_S(A(S)) = 0$.

Notice that since the $A(S) = \tilde{A}(S)$ when $S$ is consistent

$$\mathbb{E}[\mathcal{L}_\mathcal{D}(A(S)) \mid \text{consistent } S] = \mathbb{E}\left[\mathcal{L}_\mathcal{D}\left(\tilde{A}(S)\right) \mid \text{consistent } S\right]$$

and therefore we can find bounds for the generalization error of $\tilde{A}(S)$ by analyzing $A(S)$. In addition, when it can be inferred from context we use $A(S)$ to denote the min-size and posterior sampling interpolators (instead of $A_L(S)$ or $A_{\underline{d}}(S)$, respectively).

For ease of exposition, throughout the appendix, we rephrase the assumptions made in Section 4, namely, that $N = \omega\left(d_0^2 \log d_0\right)$ and $N = o\left(1/\sqrt{\mathcal{D}_{\max}}\right)$, as follows.

**Assumption A.1** (Bounded input dimension). $d_0 = o\left(\sqrt{N/\log N}\right)$.

**Assumption A.2** (Data distribution flatness). $\mathcal{D}_{\max} = o\left(1/N^2\right)$.

---

[8]As most of the appendix deals with the modified learning rule, we use $\tilde{A}(S)$ for the original one and $A(S)$ for the modified one.

## A.2  Auxiliary results

We start by citing several standard results from information theory and lemmas from Manoj and Srebro [58] which will be useful throughout our supplementary materials.

**Lemma A.3** (Chain rule of mutual information). *For any random variables $A_1, A_2$ and $B$*

$$I\left(\left(A_1, A_2\right); B\right) = I\left(A_2; B \mid A_1\right) + I\left(A_1; B\right) .$$

**Lemma A.4.** *Let $A$ and $B$ be any two random variables with associated marginal distributions $p_A$, $p_B$, and joint $p_{A,B}$. Let $q_{A|B}$ be any conditional distribution (i.e. such that for any $b$, $q_{A|B}\left(\cdot, b\right)$ is a normalized non-negative measure). Then:*

$$I\left(A; B\right) \geq \mathbb{E}_{A,B \sim p_{A,B}}\left[\log\left(\frac{dq_{A|B}\left(A|B\right)}{dp_A\left(A\right)}\right)\right] .$$

**Lemma A.5.** *Let $A_1, A_2, B$ be random variables where $A_1$ and $A_2$ are independent. Then*

$$I\left(\left(A_1, A_2\right); B\right) \geq I\left(A_1; B\right) + I\left(A_2; B\right) .$$

**Lemma A.6** (Lemma A.4 from Manoj and Srebro [58]). *For $C \geq 0$ and $0 \leq \alpha \leq 1$ it holds that*

$$1 - 2^{-H(\alpha)-C} \leq 1 - 2^{-H(\alpha)} + C .$$

**Lemma A.7.** *Let $\varepsilon \in \left(0, \frac{1}{2}\right)$ and*

$$\phi\left(t\right) \triangleq \phi_\varepsilon\left(t\right) = \frac{\varepsilon^t}{\varepsilon^t + \left(1-\varepsilon\right)^t} = \frac{1}{1 + \left(\frac{1}{\varepsilon} - 1\right)^t} .$$

*Then, $\phi$ is monotonically decreasing as a function of $t$, and convex in $(0, \infty)$.*

*Proof.* Denote $\alpha \triangleq \frac{1}{\varepsilon} - 1$ then

$$\phi\left(t\right) = \frac{1}{1 + \alpha^t}$$

$$\phi'\left(t\right) = \frac{-\ln\left(\alpha\right)\alpha^t}{\left(1+\alpha^t\right)^2} = -\ln\left(\alpha\right) \cdot \frac{\alpha^t}{1 + 2\alpha^t + \alpha^{2t}}$$

$$\phi''\left(t\right) = -\ln\left(\alpha\right) \cdot \frac{\ln\left(\alpha\right)\alpha^t\left(1+\alpha^t\right)^2 - \alpha^t \cdot 2\left(1+\alpha^t\right)\cdot\ln\left(\alpha\right)\alpha^t}{\left(1+\alpha^t\right)^4}$$

$$= -\ln\left(\alpha\right)^2 \cdot \alpha^t \cdot \frac{\left(1+\alpha^t\right) - 2\alpha^t}{\left(1+\alpha^t\right)^3} = \ln\left(\alpha\right)^2 \cdot \alpha^t \cdot \frac{\alpha^t - 1}{\left(1+\alpha^t\right)^3} .$$

Notice that for any $\varepsilon \in \left(0, \frac{1}{2}\right)$, $\alpha = \frac{1}{\varepsilon} - 1 > 1$ so for all $t > 0$

$$\alpha^t - 1 > 0$$

and $\phi''\left(t\right) > 0$ so the function is indeed convex, and $-\ln\left(\alpha\right) < 0$ so $\phi$ is decreasing. □

**Corollary A.8.** *For all $t > 0$ it holds that*

$$\phi(t) \geq \phi(1) + \phi'(1)(t-1) = \varepsilon + \ln 2 \left(\varepsilon \log(\varepsilon) + \varepsilon H(\varepsilon)\right)(t-1) .$$

*Proof.* Substituting $t = 1$,

$$\phi(1) = \frac{\varepsilon}{\varepsilon + (1-\varepsilon)} = \varepsilon$$

$$\phi'(1) = -\ln(\alpha) \cdot \frac{\alpha}{(1+\alpha)^2} = -\ln\left(\frac{1}{\varepsilon} - 1\right) \cdot \frac{\frac{1}{\varepsilon} - 1}{\left(1 + \left(\frac{1}{\varepsilon} - 1\right)\right)^2} = -\ln\left(\frac{1-\varepsilon}{\varepsilon}\right) \cdot \frac{\frac{1}{\varepsilon} - 1}{\left(\frac{1}{\varepsilon}\right)^2}$$

$$= -\left(\ln(1-\varepsilon) - \ln(\varepsilon)\right) \cdot \left(\varepsilon - \varepsilon^2\right) = \varepsilon(1-\varepsilon)\ln(\varepsilon) - \varepsilon(1-\varepsilon)\ln(1-\varepsilon)$$

$$= \varepsilon \ln(\varepsilon) - \varepsilon\left(\varepsilon \ln(\varepsilon) + (1-\varepsilon)\ln(1-\varepsilon)\right)$$

$$= \varepsilon \ln 2 \left(\log(\varepsilon) - \left(\varepsilon \log(\varepsilon) + (1-\varepsilon)\log(1-\varepsilon)\right)\right)$$

$$= \varepsilon \ln 2 \left(\log(\varepsilon) + H(\varepsilon)\right) = \ln 2 \left(\varepsilon \log(\varepsilon) + \varepsilon H(\varepsilon)\right) .$$

The inequality then holds due to convexity. $\qquad\square$

Finally, for completeness, we derive the relationship between the generalization error with respect to the noisy distribution $\mathcal{L}_{\mathcal{D}}(h)$, and the generalization error with respect to the clean distribution $\mathcal{L}_{\mathcal{D}_0}(h)$.

**Lemma A.9.** *Let $\mathcal{D}$ be a distribution as in Section 2.2, with independent noise with label flipping probability $\varepsilon^\star \in \left(0, \frac{1}{2}\right)$. Let $\mathcal{D}_0$ be the clean distribution, i.e., the distribution with label flipping probability $0$. If*

$$\mathcal{L}_{\mathcal{D}}(h) = \mathbb{P}_{(X,Y)\sim\mathcal{D}}(h(X) \neq Y)$$

*and*

$$\mathcal{L}_{\mathcal{D}_0}(h) = \mathbb{P}_{(X,Y)\sim\mathcal{D}_0}(h(X) \neq Y) = \mathbb{P}_{(X)\sim\mathcal{D}}(h(X) \neq h^\star(X))$$

*then*

$$\mathcal{L}_{\mathcal{D}_0}(h) = \frac{\mathcal{L}_{\mathcal{D}}(h) - \varepsilon^\star}{1 - 2\varepsilon^\star} .$$

*Proof.* By definition,

$$\mathcal{L}_{\mathcal{D}}(h) = \mathbb{P}_{(X,Y)\sim\mathcal{D}}(h(X) \neq Y)$$

$$= \mathbb{P}_{(X,Y)\sim\mathcal{D}}(h(X) \neq Y \mid h^\star(X) \oplus Y = 0)\, \mathbb{P}_{(X,Y)\sim\mathcal{D}}(h^\star(X) \oplus Y = 0)$$

$$+ \mathbb{P}_{(X,Y)\sim\mathcal{D}}(h(X) \neq Y \mid h^\star(X) \oplus Y = 1)\, \mathbb{P}_{(X,Y)\sim\mathcal{D}}(h^\star(X) \oplus Y = 1) .$$

Since we assume that the noise is independent,

$$\mathcal{L}_{\mathcal{D}}(h) = (1 - \varepsilon^\star)\, \mathbb{P}_{(X,Y)\sim\mathcal{D}}(h(X) \neq Y \mid h^\star(X) \oplus Y = 0)$$

$$+ \varepsilon^\star \mathbb{P}_{(X,Y)\sim\mathcal{D}}(h(X) \neq Y \mid h^\star(X) \oplus Y = 1)$$

$$= (1 - \varepsilon^\star)\, \mathbb{P}_{(X,Y)\sim\mathcal{D}}(h(X) \neq h^\star(X)) + \varepsilon^\star \mathbb{P}_{(X,Y)\sim\mathcal{D}}(h(X) = h^\star(X))$$

$$= (1 - \varepsilon^\star)\, \mathcal{L}_{\mathcal{D}_0}(h) + \varepsilon^\star\left(1 - \mathcal{L}_{\mathcal{D}_0}(h)\right)$$

$$= \varepsilon^\star + (1 - 2\varepsilon^\star)\, \mathcal{L}_{\mathcal{D}_0}(h) ,$$

or equivalently,

$$\mathcal{L}_{\mathcal{D}_0}(h) = \frac{\mathcal{L}_{\mathcal{D}}(h) - \varepsilon^\star}{1 - 2\varepsilon^\star} .$$

$\qquad\square$

*Remark A.10.* In particular, under the assumptions of the lemma, $\mathcal{L}_{\mathcal{D}}(h) = 2\varepsilon^\star(1 - \varepsilon^\star)$ is equivalent to $\mathcal{L}_{\mathcal{D}_0}(h) = \varepsilon^\star$.

# B Data consistency

Before moving on to generalization, we address some key properties of the training set's consistency.

**Lemma B.1.** *For any distribution over the data $\mathcal{D}$, $\mathbb{P}\left(\text{inconsistent } S\right) \leq \frac{1}{2}N^2 \mathcal{D}_{\max}$ .*

*Proof.* Using the union bound,

$$
\begin{aligned}
\mathbb{P}\left(\text{inconsistent } S\right) &= \mathbb{P}\left(\exists i \neq j \in [N] \,:\, X_i = X_j, Y_i \neq Y_j\right) \\
&\leq \mathbb{P}\left(\exists i \neq j \in [N] \,:\, X_i = X_j\right) \\
&\leq \sum_{i \neq j} \mathbb{P}\left(X_i = X_j\right) = \binom{N}{2}\mathbb{P}\left(X_1 = X_2\right) = \binom{N}{2}\sum_{x \in \mathcal{X}}\mathbb{P}\left(X_1 = x\right)\mathbb{P}\left(X_2 = x\right) \\
&\leq \binom{N}{2}\sum_{x \in \mathcal{X}}\mathcal{D}_{\max}\mathbb{P}\left(X = x\right) = \binom{N}{2}\mathcal{D}_{\max} \leq \frac{1}{2}N^2 \mathcal{D}_{\max} \,.
\end{aligned}
$$

$\square$

Hence, under Assumption A.2 we have $\mathbb{P}\left(\text{inconsistent } S\right) = o\left(1\right)$, *i.e.,* the inconsistency probability is asymptotically small.

## B.1 Independent label noise

We now focus on the case of independent label noise, *i.e.,* $Y \oplus h^\star\left(X\right) \mid \{X = x\} \sim \text{Ber}\left(\varepsilon^\star\right)$ for any $x \in \mathcal{X}$. Recall the noise level

$$
\varepsilon^\star = \mathbb{P}_{(X,Y)\sim\mathcal{D}}\left(Y \neq h^\star\left(X\right)\right) = \mathbb{P}_S\left(Y_1 \neq h^\star\left(X_1\right)\right)
$$

and we define the "effective" noise level in a *consistent* training set

$$
\hat{\varepsilon}_{\text{tr}} \triangleq \mathbb{P}_S\left(Y_1 \neq h^\star\left(X_1\right) \mid \text{consistent } S\right) \,. \tag{4}
$$

We relate $\hat{\varepsilon}_{\text{tr}}$ to $\varepsilon^\star$ in the following lemma.

**Lemma B.2.** *In the independent noise setting, it holds that*

$$
\left|\hat{\varepsilon}_{tr} - \varepsilon^\star\right| \leq \left|\ln 2\left(\varepsilon^\star\log\left(\varepsilon^\star\right) + \varepsilon^\star H\left(\varepsilon^\star\right)\right)\right| \cdot \left(N - 1\right)\frac{\mathcal{D}_{\max}}{\mathbb{P}\left(\text{consistent } S\right)} \,,
$$

*and moreover, $\hat{\varepsilon}_{tr} \leq \varepsilon^\star$.*

*Proof.* Conditioning on $S$ being consistent (having no label "collisions"), all occurrences of $x$ in $S$ must have the same label so

$$
\mathbb{P}_S\left(Y_1 \neq h^\star\left(X_1\right) \mid (X_1, Y_1) \text{ appears } k \text{ times in } S, \text{ consistent } S\right) = \frac{\varepsilon^{\star k}}{\varepsilon^{\star k} + \left(1 - \varepsilon^\star\right)^k}
$$

Therefore,

$$
\begin{aligned}
\hat{\varepsilon}_{\text{tr}} &= \mathbb{P}_S\left(Y_1 \neq h^\star\left(X_1\right) \mid \text{consistent } S\right) \\
&= \sum_{k=1}^{N} \mathbb{P}_S\left(Y_1 \neq h^\star\left(X_1\right) \mid (X_1, Y_1) \text{ appears } k \text{ times in } S, \text{ consistent } S\right) \\
&\qquad \cdot \mathbb{P}\left((X_1, Y_1) \text{ appears } k \text{ times in } S \mid \text{consistent } S\right) \\
&= \sum_{k=1}^{N} \frac{\varepsilon^{\star k}}{\varepsilon^{\star k} + \left(1 - \varepsilon^\star\right)^k} \cdot \mathbb{P}\left((X_1, Y_1) \text{ appears } k \text{ times in } S \mid \text{consistent } S\right) \\
&\leq \sum_{k=1}^{N} \frac{\varepsilon^{\star 1}}{\varepsilon^{\star 1} + \left(1 - \varepsilon^\star\right)^1} \cdot \mathbb{P}\left((X_1, Y_1) \text{ appears } k \text{ times in } S \mid \text{consistent } S\right) \\
&= \varepsilon^\star \underbrace{\sum_{k=1}^{N} \cdot\mathbb{P}\left((X_1, Y_1) \text{ appears } k \text{ times in } S \mid \text{consistent } S\right)}_{\text{sums to 1}} = \varepsilon^\star \,.
\end{aligned}
\tag{5}
$$

On the other hand, define

$$K(S) \triangleq |\{i \in [N] \mid X_i = X_1\}| = \sum_{i=1}^{N} \mathbb{I}\{X_i = X_1\}$$

then

$$\mathbb{E}_S[K(S) \mid \text{consistent } S] = 1 + \sum_{i=2}^{N} \mathbb{E}_S[\mathbb{I}\{X_i = X_1\} \mid \text{consistent } S]$$

$$= 1 + (N-1)\mathbb{P}_S(X_2 = X_1 \mid \text{consistent } S).$$

Next,

$$\mathbb{P}(X_2 = X_1 \mid \text{consistent } S) = \frac{\mathbb{P}(X_2 = X_1, \text{consistent } S)}{\mathbb{P}(\text{consistent } S)} \leq \frac{\mathbb{P}(X_1 = X_2)}{\mathbb{P}(\text{consistent } S)}.$$

Since $d\mathcal{D}(x) \leq \mathcal{D}_{\max}$ for all $x \in \mathcal{X}$, as in the proof of Lemma B.1

$$\mathbb{E}_S[K(S) \mid \text{consistent } S] \leq 1 + (N-1)\frac{\mathbb{P}(X_1 = X_2)}{\mathbb{P}(\text{consistent } S)} \leq 1 + (N-1)\frac{\mathcal{D}_{\max}}{\mathbb{P}(\text{consistent } S)}.$$

Then, using Lemma A.7 we get,

$$\hat{\varepsilon}_{\text{tr}} = \sum_{k=1}^{N} \frac{\varepsilon^{\star k}}{\varepsilon^{\star k} + (1-\varepsilon^\star)^k} \cdot \mathbb{P}(X_1 \text{ appears } k \text{ times in } S \mid \text{consistent } S)$$

$$= \sum_{k=1}^{N} \phi_{\varepsilon^\star}(k) \cdot \mathbb{P}(X_1 \text{ appears } k \text{ times in } S \mid \text{consistent } S)$$

$$= \mathbb{E}_S\left[\underbrace{\phi_{\varepsilon^\star}(K(S))}_{\text{convex in } k} \mid \text{consistent } S\right]$$

$$[\text{Jensen}] \geq \phi_{\varepsilon^\star}(\mathbb{E}_S[K(S) \mid \text{consistent } S])$$

$$[\text{decreasing}] \geq \phi_{\varepsilon^\star}\left(1 + (N-1) \cdot \frac{\mathcal{D}_{\max}}{\mathbb{P}(\text{consistent } S)}\right).$$

Corollary A.8 implies that

$$\hat{\varepsilon}_{\text{tr}} \geq \phi_{\varepsilon^\star}\left(1 + (N-1) \cdot \frac{\mathcal{D}_{\max}}{\mathbb{P}(\text{consistent } S)}\right)$$

$$\geq \varepsilon^\star + \ln 2\left(\varepsilon^\star \log(\varepsilon^\star) + \varepsilon^\star H(\varepsilon^\star)\right) \cdot (N-1)\frac{\mathcal{D}_{\max}}{\mathbb{P}(\text{consistent } S)}.$$

Combining the bounds we get

$$|\hat{\varepsilon}_{\text{tr}} - \varepsilon^\star| \leq |\ln 2\left(\varepsilon^\star \log(\varepsilon^\star) + \varepsilon^\star H(\varepsilon^\star)\right)| \cdot (N-1)\frac{\mathcal{D}_{\max}}{\mathbb{P}(\text{consistent } S)}.$$

$$\square$$

## C   Generalization bounds

We present two generalization bounds for the population error of an interpolating algorithm in terms of the mutual information of it with the training set.

*Remark* C.1 (High consistency probability). Throughout the appendix we assume that the consistency satisfies $\mathbb{P}_S (\text{consistent } S) \geq \frac{1}{2}$. While this assumption is not without loss of generality, it is a weaker version of Assumption A.2 and implied by it (asymptotically). As Assumption A.2 is assumed in all "downstream results" that this appendix aims to support, we find it is reasonable to assume here.

### C.1   Arbitrary label noise

In this subsection, we provide a generalization bound for interpolating algorithms without any assumptions on the distribution of the noise $Y \oplus h^\star (X) \mid \{X = x\}$, other than $\mathcal{L}_\mathcal{D} (h^\star) = \varepsilon^\star$.

**Lemma C.2.** *For any interpolating learning algorithm $A (S)$,*

$$- \log \left( 1 - \mathbb{E}_{S, A(S)} \left[ \mathcal{L}_\mathcal{D} (A (S)) \mid \text{consistent } S \right] \right) \leq \frac{I (S; A (S))}{N \cdot \mathbb{P}_S (\text{consistent } S)} .$$

*Proof.* We rely on Lemma A.4. Specifically, we shall use the following suggested conditional distribution. For $h \neq \star$ let

$$dq (s|h) = \frac{1}{Z_h} \mathbb{I} \{\mathcal{L}_s (h) = 0\} d\mathcal{D}^N (s)$$

where

$$\begin{aligned}
Z_h &= \sum_s \mathbb{I} \{\mathcal{L}_s (h) = 0\} d\mathcal{D}^N (s) \\
&= \mathbb{E}_S \mathbb{I} \{\mathcal{L}_S (h) = 0\} \\
&= \mathbb{P}_S (\mathcal{L}_S (h) = 0) \\
&= (1 - \mathcal{L}_\mathcal{D} (h))^N .
\end{aligned}$$

For $h = \star$ let

$$dq (s|\star) = \frac{\mathbb{I} \{\text{inconsistent } s\} d\mathcal{D}^N (s)}{\sum_{s'} \mathbb{I} \{\text{inconsistent } s'\} d\mathcal{D}^N (s')} = \mathbb{I} \{\text{inconsistent } s\} \frac{d\mathcal{D}^N (s)}{\mathbb{P}_S (\text{inconsistent } S)} .$$

Clearly, if $h \neq \star$ and $dq (s|h) = 0$ then either $d\mathcal{D}^N (s) = 0$ so $dp (s, h) = 0$ as well, or $\mathcal{L}_s (h) \neq 0$. So, since $h \neq \star$, $s$ can be interpolated and $dp (s \mid h) = 0$. That is, the proposed conditional distribution is absolutely continuous w.r.t. the true conditional distribution. When $h = \star$, $q$ is the true conditional distribution given that $h = \star$ so it is also absolutely continuous w.r.t. it. That is, the proposed distribution is

$$dq (s \mid h) = \frac{\mathbb{I} \{\text{inconsistent } s\}}{\mathbb{P}_S (\text{inconsistent } S)} \mathbb{I} \{h = \star\} d\mathcal{D}^N (s) + \frac{\mathbb{I} \{\mathcal{L}_s (h) = 0\}}{(1 - \mathcal{L}_\mathcal{D} (h))^N} \mathbb{I} \{h \neq \star\} d\mathcal{D}^N (s) .$$

From Lemma A.4

$$\begin{aligned}
I (S; A (S)) &\geq \mathbb{E}_{S, A(S)} \left[ \log \left( \frac{dq (S|A (S))}{d\mathcal{D}^N (S)} \right) \right] \\
&= \mathbb{E}_{S, A(S)} \left[ \log \left( \frac{\mathbb{I} \{\text{inconsistent } S\}}{\mathbb{P}_{S'} (\text{inconsistent } S')} \mathbb{I} \{A (S) = \star\} + \frac{\mathbb{I} \{\mathcal{L}_S (A(S)) = 0\}}{(1 - \mathcal{L}_\mathcal{D} (A(S)))^N} \mathbb{I} \{A (S) \neq \star\} \right) \right] .
\end{aligned}$$

$\mathbb{I} \{A (S) = \star\}$ and $\mathbb{I} \{A (S) \neq \star\}$ are mutually exclusive so

$$\begin{aligned}
&I (S; A (S)) \\
&\geq \mathbb{E}_{S, A(S)} \left[ \log \left( \frac{\mathbb{I} \{\text{inconsistent } S\}}{\mathbb{P}_{S'} (\text{inconsistent } S')} \right) \mathbb{I} \{A (S) = \star\} + \log \left( \frac{\mathbb{I} \{\mathcal{L}_S (A(S)) = 0\}}{(1 - \mathcal{L}_\mathcal{D} (A(S)))^N} \right) \mathbb{I} \{A (S) \neq \star\} \right] .
\end{aligned}$$

The first term is 0 when $A (S) \neq \star$ and positive when $A (S) = \star$ (and so always non-negative). Furthermore, whenever $dp (S, A (S)) > 0$ and $\mathbb{I} \{A (S) \neq \star\} = 1$ hold together, they imply that

$\mathbb{I}\{\mathcal{L}_S(A(S)) = 0\} = 1$ so we have

$$I(S; A(S)) \geq \mathbb{E}_{S,A(S)}\left[\log\left(\frac{1}{(1 - \mathcal{L}_{\mathcal{D}}(A(S)))^N}\right)\mathbb{I}\{A(S) \neq \star\}\right]$$
$$= -\mathbb{E}_{S,A(S)}[N\log(1 - \mathcal{L}_{\mathcal{D}}(A(S)))\mathbb{I}\{A(S) \neq \star\}] .$$

Using Jensen's inequality,

$$-\mathbb{E}_{S,A(S)}[N\log(1 - \mathcal{L}_{\mathcal{D}}(A(S)))\mathbb{I}\{A(S) \neq \star\}]$$
$$= -N\mathbb{E}_{S,A(S)}[\log(1 - \mathcal{L}_{\mathcal{D}}(A(S))) \mid \mathbb{I}\{A(S) \neq \star\}]\mathbb{P}_{S,A(S)}(A(S) \neq \star)$$
$$\geq -N\log\left(1 - \mathbb{E}_{S,A(S)}[\mathcal{L}_{\mathcal{D}}(A(S)) \mid \mathbb{I}\{A(S) \neq \star\}]\right)\mathbb{P}_{S,A(S)}(A(S) \neq \star)$$
$$= -N\log\left(1 - \mathbb{E}_{S,A(S)}[\mathcal{L}_{\mathcal{D}}(A(S)) \mid \text{consistent } S]\right)\mathbb{P}_S(\text{consistent } S)$$

so

$$I(S; A(S)) \geq -N\log\left(1 - \mathbb{E}_{S,A(S)}[\mathcal{L}_{\mathcal{D}}(A(S)) \mid \text{consistent } S]\right)\mathbb{P}_S(\text{consistent } S) .$$

Rearranging the inequality

$$-\log\left(1 - \mathbb{E}_{S,A(S)}[\mathcal{L}_{\mathcal{D}}(A(S)) \mid \text{consistent } S]\right) \leq \frac{I(S; A(S))}{N \cdot \mathbb{P}_S(\text{consistent } S)}$$

$\square$

## C.2 Independent Noise

**Lemma C.3.** *Assuming independent noise, the generalization error of interpolating learning rules satisfies the following.*

$$\left| \mathbb{E}_{S,A(S)} \left[ \mathcal{L}_\mathcal{D} \left( A\left( S \right) \right) \right] \mid \text{consistent } S \right] - 2\varepsilon^\star \left( 1 - \varepsilon^\star \right) \right|$$

$$\leq \left( 1 - 2\varepsilon^\star \right) \sqrt{C\left( N \right)} + \frac{\left( N - 1 \right) \mathcal{D}_{\max}}{3},$$

*where $C\left( N \right) \triangleq \frac{I(S;A(S)) - N \cdot (H(\varepsilon^\star) - \mathbb{P}(\text{inconsistent } S))}{N(1 - \mathbb{P}(\text{inconsistent } S))}$.*

*Proof.* As in the proof of Lemma 4.2 in Manoj and Srebro [58], since $S$ is sampled i.i.d., we have

$$I\left( S; A\left( S \right) \right) \overset{A.5}{\geq} \sum_{i=1}^N I\left( \left( X_i, Y_i \right); A\left( S \right) \right) = N \cdot I\left( \left( X_1, Y_1 \right); A\left( S \right) \right)$$

$$\overset{A.3}{=} N \cdot I\left( X_1; A\left( S \right) \right) + N \cdot I\left( Y_1; A\left( S \right) \mid X_1 \right). \tag{6}$$

Using properties of conditional mutual information,

$$I\left( Y_1; A\left( S \right) \mid X_1 \right) = H\left( Y_1 \mid X_1 \right) - H\left( Y_1 \mid A\left( S \right), X_1 \right). \tag{7}$$

For the first term in (7), we employ the fact that for any $x \in \mathcal{X}$, either $Y_1 \mid X_1 = x \sim \text{Ber}\left( \varepsilon^\star \right)$ or $Y_1 \mid X_1 = x \sim \text{Ber}\left( 1 - \varepsilon^\star \right)$ to get

$$H\left( Y_1 \mid X_1 \right) = \mathbb{E}_{(X,Y) \sim \mathcal{D}} \left[ H\left( Y_1 \mid X_1 = X \right) \right] = \mathbb{E}_{(X,Y) \sim \mathcal{D}} \left[ H\left( \varepsilon^\star \right) \right] = H\left( \varepsilon^\star \right).$$

For the second term in (7), we again employ the definition of conditional entropy,

$$H\left( Y_1 \mid A\left( S \right), X_1 \right) = - \sum_{x_1 \in \mathcal{X}} \sum_{h \in \mathcal{H} \cup \{\star\}} \left[ dp\left( \left( x_1, 0 \right), h \right) \log \left( \frac{dp\left( \left( x_1, 0 \right), h \right)}{dp\left( x_1, h \right)} \right) \right.$$

$$\left. + dp\left( \left( x_1, 1 \right), h \right) \log \left( \frac{dp\left( \left( x_1, 1 \right), h \right)}{dp\left( x_1, h \right)} \right) \right].$$

When $h \neq \star$, the marginal distribution of a training data point and a hypothesis is

$$dp\left( \left( x, y \right), h \right) = dp\left( y \mid x, h \right) dp\left( x, h \right) = \mathbb{I}\left\{ y = h\left( x \right) \right\} dp\left( x, h \right),$$

and the inner sum becomes a sum over expressions of the form:

$$dp\left( \left( x_1, h\left( x_1 \right) \right), h \right) \log \left( \frac{dp\left( \left( x_1, h\left( x_1 \right) \right), h \right)}{dp\left( x_1, h \right)} \right) = dp\left( x_1, h \right) \underbrace{\log \left( \frac{dp\left( x_1, h \right)}{dp\left( x_1, h \right)} \right)}_{=0} = 0.$$

Therefore, we have that,

$$H\left( Y_1 \mid A\left( S \right), X_1 \right)$$

$$= - \sum_{x_1 \in \mathcal{X}} \left[ dp\left( \left( x_1, 0 \right), \star \right) \log \left( \frac{dp\left( \left( x_1, 0 \right), \star \right)}{dp\left( x_1, \star \right)} \right) + dp\left( \left( x_1, 1 \right), \star \right) \log \left( \frac{dp\left( \left( x_1, 1 \right), \star \right)}{dp\left( x_1, \star \right)} \right) \right]$$

Employing conditional probabilities (notice that $\frac{dp((x_1, 0), \star)}{dp(x_1, \star)} = \frac{dp(0 \mid x_1, \star) dp(x_1, \star)}{dp(x_1, \star)} = dp\left( 0 \mid x_1, \star \right)$), we get,

$$H\left( Y_1 \mid A\left( S \right), X_1 \right)$$

$$= - \sum_{x_1 \in \mathcal{X}} dp\left( x_1, \star \right) \left[ dp\left( 0 \mid x_1, \star \right) \log \left( dp\left( 0 \mid x_1, \star \right) \right) + dp\left( 1 \mid x_1, \star \right) \log \left( dp\left( 1 \mid x_1, \star \right) \right) \right]$$

$$= \sum_{x_1 \in \mathcal{X}} dp\left( x_1, \star \right) H\left( dp\left( 0 \mid x_1, \star \right) \right) = dp\left( \star \right) \sum_{x_1 \in \mathcal{X}} dp\left( x_1 \mid \star \right) H\left( dp\left( 0 \mid x_1, \star \right) \right)$$

$$= \mathbb{P}\left( A\left( S \right) = \star \right) \mathbb{E}_{(S, A(S)) \sim p} \left[ \underbrace{H\left( dp\left( 0 \mid X_1, \star \right) \right)}_{\leq 1} \mid \text{inconsistent } S \right] \leq \mathbb{P}_S\left( \text{inconsistent } S \right).$$

Overall, the right term in (6) is lower bounded by,

$$I\left(Y_1; A\left(S\right) \mid X_1\right) = H\left(Y_1 \mid X_1\right) - H\left(Y_1 \mid A\left(S\right), X_1\right) \geq H\left(\varepsilon^\star\right) - \mathbb{P}\left(\text{inconsistent } S\right) .$$

For the left term, *i.e.,* $I\left(X_1; A\left(S\right)\right)$, we use the variational bound Lemma A.4 with the following suggested conditional distribution.

- For $h = \star$, choose $dq\left(x_1 \mid \star\right) = dp\left(x_1\right)$ (notice that $\sum_{x_1 \in \mathcal{X}} dq\left(x_1 \mid \star\right) = \sum_{x_1} dp\left(x_1\right) = 1$.

- Otherwise, if $h \neq \star$, denote $q_\varepsilon = \text{Ber}\left(\varepsilon\right)$, and

$$\hat{\varepsilon}_{\text{tr}} = \mathbb{P}_S\left(Y_1 \neq h^\star\left(X_1\right) \mid \text{consistent } S\right)$$
$$\hat{\varepsilon}_{\text{gen}} = \mathbb{E}_{(S, A(S)) \sim p}\left[\mathbb{P}_{X \sim \mathcal{D}}\left(A\left(S\right)\left(X\right) \neq h^\star\left(X\right)\right) \mid A\left(S\right) \neq \star\right] .$$

Note that $\hat{\varepsilon}_{\text{tr}}$ may differ from $\varepsilon^\star$. We choose the following conditional distribution

$$dq\left(x_1 \mid h\right) = \frac{1}{Z_h} \cdot \frac{dq_{\hat{\varepsilon}_{\text{tr}}}\left(h\left(x_1\right) \oplus h^\star\left(x_1\right)\right)}{dq_{\hat{\varepsilon}_{\text{gen}}}\left(h\left(x_1\right) \oplus h^\star\left(x_1\right)\right)} dp\left(x_1\right) .$$

In total, we choose,

$$dq\left(x_1 \mid h\right) = \frac{1}{Z_h} \cdot \frac{dq_{\hat{\varepsilon}_{\text{tr}}}\left(h\left(x_1\right) \oplus h^\star\left(x_1\right)\right)}{dq_{\hat{\varepsilon}_{\text{gen}}}\left(h\left(x_1\right) \oplus h^\star\left(x_1\right)\right)} \cdot \mathbb{I}\left\{h \neq \star\right\} dp\left(x_1\right) + \mathbb{I}\left\{h = \star\right\} dp\left(x_1\right) ,$$

where $Z_h$ is the corresponding partition function.

Then, we use (A.4) and properties of logarithms and indicators to show that,

$$I\left(X_1; A\left(S\right)\right) \geq \mathbb{E}_{S, A(S)}\left[\log\left(\frac{dq\left(X_1 \mid A\left(S\right)\right)}{dp\left(X_1\right)}\right)\right]$$

$$= \mathbb{E}_{S, A(S)}\left[\log\left(\frac{\frac{1}{Z_{A(S)}}\frac{dq_{\hat{\varepsilon}_{\text{tr}}}(h(X_1) \oplus h^\star(X_1))}{dq_{\hat{\varepsilon}_{\text{gen}}}(h(X_1) \oplus h^\star(X_1))}dp\left(X_1\right)\mathbb{I}\left\{A\left(S\right) \neq \star\right\} + dp\left(X_1\right)\mathbb{I}\left\{A\left(S\right) = \star\right\}}{dp\left(X_1\right)}\right)\right]$$

$$= \mathbb{E}_{S, A(S)}\left[\log\left(\frac{1}{Z_{A(S)}}\frac{dq_{\hat{\varepsilon}_{\text{tr}}}\left(h\left(X_1\right) \oplus h^\star\left(X_1\right)\right)}{dq_{\hat{\varepsilon}_{\text{gen}}}\left(h\left(X_1\right) \oplus h^\star\left(X_1\right)\right)}\mathbb{I}\left\{A\left(S\right) \neq \star\right\} + \mathbb{I}\left\{A\left(S\right) = \star\right\}\right)\right]$$

$$= \mathbb{E}_{S, A(S)}\left[\log\left(\frac{1}{Z_{A(S)}}\frac{dq_{\hat{\varepsilon}_{\text{tr}}}\left(h\left(X_1\right) \oplus h^\star\left(X_1\right)\right)}{dq_{\hat{\varepsilon}_{\text{gen}}}\left(h\left(X_1\right) \oplus h^\star\left(X_1\right)\right)}\right)\mathbb{I}\left\{A\left(S\right) \neq \star\right\}\right] +$$

$$\mathbb{E}_{S, A(S)}\Big[\underbrace{\log\left(1\right)\mathbb{I}\left\{A\left(S\right) = \star\right\}}_{=0}\Big]$$

$$= \mathbb{E}_{S, A(S)}\left[\log\left(\frac{1}{Z_{A(S)}}\frac{dq_{\hat{\varepsilon}_{\text{tr}}}\left(h\left(X_1\right) \oplus h^\star\left(X_1\right)\right)}{dq_{\hat{\varepsilon}_{\text{gen}}}\left(h\left(X_1\right) \oplus h^\star\left(X_1\right)\right)}\right)\mathbb{I}\left\{A\left(S\right) \neq \star\right\}\right] .$$

Using the law of total expectation, the above becomes,

$$= \mathbb{P}\left(A\left(S\right) \neq \star\right)\mathbb{E}_{S, A(S)}\left[\log\left(\frac{1}{Z_{A(S)}} \cdot \frac{dq_{\hat{\varepsilon}_{\text{tr}}}\left(h\left(X_1\right) \oplus h^\star\left(X_1\right)\right)}{dq_{\hat{\varepsilon}_{\text{gen}}}\left(h\left(X_1\right) \oplus h^\star\left(X_1\right)\right)}\right) \,\Big|\, A\left(S\right) \neq \star\right] ,$$

where we also use Jensen's inequality to show,

$$\mathbb{E}_{S, A(S)}\left[\log\left(\frac{1}{Z_{A(S)}} \cdot \frac{dq_{\hat{\varepsilon}_{\text{tr}}}\left(h\left(X_1\right) \oplus h^\star\left(X_1\right)\right)}{dq_{\hat{\varepsilon}_{\text{gen}}}\left(h\left(X_1\right) \oplus h^\star\left(X_1\right)\right)}\right) \,\Big|\, A\left(S\right) \neq \star\right]$$

$$= \mathbb{E}_{S, A(S)}\left[\log\left(\frac{dq_{\hat{\varepsilon}_{\text{tr}}}\left(h\left(X_1\right) \oplus h^\star\left(X_1\right)\right)}{dq_{\hat{\varepsilon}_{\text{gen}}}\left(h\left(X_1\right) \oplus h^\star\left(X_1\right)\right)}\right) \,\Big|\, A\left(S\right) \neq \star\right] - \mathbb{E}_{S, A(S)}\left[\log\left(Z_{A(S)}\right) \,\big|\, A\left(S\right) \neq \star\right]$$

$$\geq \mathbb{E}_{S, A(S)}\left[\log\left(\frac{dq_{\hat{\varepsilon}_{\text{tr}}}\left(h\left(X_1\right) \oplus h^\star\left(X_1\right)\right)}{dq_{\hat{\varepsilon}_{\text{gen}}}\left(h\left(X_1\right) \oplus h^\star\left(X_1\right)\right)}\right) \,\Big|\, A\left(S\right) \neq \star\right] - \log\left(\mathbb{E}_{S, A(S)}\left[Z_{A(S)} \,\big|\, A\left(S\right) \neq \star\right]\right) .$$

The partition function satisfies for all $h \neq \star$,

$$Z_h = \sum_{x_1 \in \mathcal{X}} \frac{dq_{\hat{\varepsilon}_{\mathrm{tr}}}(h(x_1) \oplus h^\star(x_1))}{dq_{\hat{\varepsilon}_{\mathrm{gen}}}(h(x_1) \oplus h^\star(x_1))} dp(x_1) = \mathbb{E}_{X \sim \mathcal{D}}\left[\frac{dq_{\hat{\varepsilon}_{\mathrm{tr}}}(h(X) \oplus h^\star(X))}{dq_{\hat{\varepsilon}_{\mathrm{gen}}}(h(X) \oplus h^\star(X))}\right]$$

$$= \mathbb{P}_{X \sim \mathcal{D}}(h(X) = h^\star(X)) \cdot \frac{1 - \hat{\varepsilon}_{\mathrm{tr}}}{1 - \hat{\varepsilon}_{\mathrm{gen}}} + \mathbb{P}_{X \sim \mathcal{D}}(h(X) \neq h^\star(X)) \cdot \frac{\hat{\varepsilon}_{\mathrm{tr}}}{\hat{\varepsilon}_{\mathrm{gen}}}.$$

Taking the expectation w.r.t. $(S, A(S)) \sim p$, we get

$$\mathbb{E}_{(S,A(S)) \sim p}\left[Z_{A(S)} \mid A(S) \neq \star\right]$$
$$= \frac{1 - \hat{\varepsilon}_{\mathrm{tr}}}{1 - \hat{\varepsilon}_{\mathrm{gen}}} \cdot \underbrace{\mathbb{E}_{S,A(S)}\left[\mathbb{P}(A(S)(X) = h^\star(X)) \mid A(S) \neq \star\right]}_{=1-\hat{\varepsilon}_{\mathrm{gen}}} +$$
$$+ \frac{\hat{\varepsilon}_{\mathrm{tr}}}{\hat{\varepsilon}_{\mathrm{gen}}} \cdot \underbrace{\mathbb{E}_{S,A(S)}\left[\mathbb{P}(A(S)(X) \neq h^\star(X)) \mid A(S) \neq \star\right]}_{=\hat{\varepsilon}_{\mathrm{gen}}} = 1.$$

Combining the above, we have that,

$$I(X_1; A(S)) \geq \mathbb{P}(A(S) \neq \star)\, \mathbb{E}_{S,A(S)}\left[\log\left(\frac{dq_{\hat{\varepsilon}_{\mathrm{tr}}}(A(S)(X_1) \oplus h^\star(X_1))}{dq_{\hat{\varepsilon}_{\mathrm{gen}}}(A(S)(X_1) \oplus h^\star(X_1))}\right) \mid A(S) \neq \star\right].$$

Notice that

$$(A(S)(X_1) \oplus h^\star(X_1) \mid \{A(S) \neq \star\}) = (Y_1 \oplus h^\star(X_1) \mid \{\text{consistent } S\}),$$

so $A(S)(X_1) \oplus h^\star(X_1) \mid \{A(S) \neq \star\} \sim \mathrm{Ber}(\hat{\varepsilon}_{\mathrm{tr}})$, and thus

$$I(X_1; A(S)) \geq \mathbb{P}(A(S) \neq \star)\, D_{KL}\left(q_{\hat{\varepsilon}_{\mathrm{tr}}} || q_{\hat{\varepsilon}_{\mathrm{gen}}}\right) = \left(1 - \mathbb{P}(\text{inconsistent } S)\right) D_{KL}\left(q_{\hat{\varepsilon}_{\mathrm{tr}}} || q_{\hat{\varepsilon}_{\mathrm{gen}}}\right).$$

Putting this all together, (6) is lower bounded by,

$$I(S; A(S)) \geq N\left(1 - \mathbb{P}(\text{inconsistent } S)\right) D_{KL}\left(q_{\hat{\varepsilon}_{\mathrm{tr}}} || q_{\hat{\varepsilon}_{\mathrm{gen}}}\right) + N\left(H(\varepsilon^\star) - \mathbb{P}(\text{inconsistent } S)\right).$$

Rearranging the inequality

$$D_{KL}\left(q_{\hat{\varepsilon}_{\mathrm{tr}}} || q_{\hat{\varepsilon}_{\mathrm{gen}}}\right) \leq \frac{I(S; A(S)) - N \cdot (H(\varepsilon^\star) - \mathbb{P}(\text{inconsistent } S))}{N(1 - \mathbb{P}(\text{inconsistent } S))} \triangleq C(N).$$

Using Pinsker's inequality, we have,

$$|\hat{\varepsilon}_{\mathrm{tr}} - \hat{\varepsilon}_{\mathrm{gen}}| \leq \sqrt{\frac{1}{2} D_{KL}\left(q_{\hat{\varepsilon}_{\mathrm{tr}}} || q_{\hat{\varepsilon}_{\mathrm{gen}}}\right)} \leq \sqrt{C(N)}.$$

We proceed to bound $\mathbb{E}_{S,A(S)}\left[\mathcal{L}_{\mathcal{D}}(A(S)) \mid \text{consistent } S\right]$ in terms of $|\hat{\varepsilon}_{\mathrm{tr}} - \hat{\varepsilon}_{\mathrm{gen}}|$. Notice that

$$\mathbb{E}_{S,A(S)}\left[\mathcal{L}_{\mathcal{D}}(A(S)) \mid \text{consistent } S\right]$$
$$= \mathbb{E}_{S,A(S)}\left[\mathbb{P}_{(X,Y) \sim \mathcal{D}}(A(S)(X) \neq Y) \mid \text{consistent } S\right]$$
$$= \mathbb{E}_{S,A(S)}\left[\mathbb{P}(A(S)(X) \neq Y \mid Y = h^\star(X)) \underbrace{\mathbb{P}(Y = h^\star(X))}_{\text{no label flip}} \mid \text{consistent } S\right] +$$
$$+ \mathbb{E}_{S,A(S)}\left[\mathbb{P}(A(S)(X) \neq Y \mid Y \neq h^\star(X)) \underbrace{\mathbb{P}(Y \neq h^\star(X))}_{\text{label flip}} \mid \text{consistent } S\right]$$
$$= \mathbb{E}_{S,A(S)}\left[\mathbb{P}(A(S)(X) \neq h^\star(X))(1 - \varepsilon^\star) + \mathbb{P}(A(S)(X) = h^\star(X))\varepsilon^\star \mid \text{consistent } S\right]$$
$$= (1 - \varepsilon^\star)\mathbb{E}_{S,A(S)}\left[\mathbb{P}(A(S)(X) \neq h^\star(X)) \mid \text{consistent } S\right] +$$
$$+ \varepsilon^\star \mathbb{E}_{S,A(S)}\left[\mathbb{P}(A(S)(X) = h^\star(X)) \mid \text{consistent } S\right]$$
$$= (1 - \varepsilon^\star)\hat{\varepsilon}_{\mathrm{gen}} + \varepsilon^\star(1 - \hat{\varepsilon}_{\mathrm{gen}}).$$

Then, using the triangle inequality,

$$\left|\mathbb{E}_{S,A(S)}\left[\mathcal{L}_{\mathcal{D}}\left(A\left(S\right)\right) \mid \text{consistent } S\right] - 2\varepsilon^{\star}\left(1-\varepsilon^{\star}\right)\right|$$
$$= \left|\left(1-\varepsilon^{\star}\right)\hat{\varepsilon}_{\text{gen}} + \varepsilon^{\star}\left(1-\hat{\varepsilon}_{\text{gen}}\right) - 2\varepsilon^{\star}\left(1-\varepsilon^{\star}\right)\right| = \left|\hat{\varepsilon}_{\text{gen}} - \hat{\varepsilon}_{\text{gen}}\varepsilon^{\star} + \varepsilon^{\star} - \hat{\varepsilon}_{\text{gen}}\varepsilon^{\star} - 2\varepsilon^{\star} + 2\varepsilon^{\star 2}\right|$$
$$= \left|\hat{\varepsilon}_{\text{gen}} - 2\hat{\varepsilon}_{\text{gen}}\varepsilon^{\star} - \varepsilon^{\star} + 2\varepsilon^{\star 2}\right| = \left|\hat{\varepsilon}_{\text{gen}}\left(1-2\varepsilon^{\star}\right) - \varepsilon^{\star}\left(1-2\varepsilon^{\star}\right)\right| = \left(1-2\varepsilon^{\star}\right)\left|\hat{\varepsilon}_{\text{gen}} - \varepsilon^{\star}\right|$$
$$\leq \left(1-2\varepsilon^{\star}\right)\left(\left|\hat{\varepsilon}_{\text{gen}} - \hat{\varepsilon}_{\text{tr}}\right| + \left|\hat{\varepsilon}_{\text{tr}} - \varepsilon^{\star}\right|\right).$$

Combining with the result from Lemma B.2 and Remark C.1

$$\left|\hat{\varepsilon}_{\text{tr}} - \varepsilon^{\star}\right| \leq \left|\ln 2\left(\varepsilon^{\star}\log\left(\varepsilon^{\star}\right) + \varepsilon^{\star}H\left(\varepsilon^{\star}\right)\right)\right| \cdot \left(N-1\right)\frac{\mathcal{D}_{\max}}{\mathbb{P}\left(\text{consistent } S\right)}$$
$$\leq \left|\ln 2\left(\varepsilon^{\star}\log\left(\varepsilon^{\star}\right) + \varepsilon^{\star}H\left(\varepsilon^{\star}\right)\right)\right| \cdot 2\left(N-1\right)\mathcal{D}_{\max}.$$

we conclude that

$$\left|\mathbb{E}_{S,A(S)}\left[\mathcal{L}_{\mathcal{D}}\left(A\left(S\right)\right) \mid \text{consistent } S\right] - 2\varepsilon^{\star}\left(1-\varepsilon^{\star}\right)\right|$$
$$\leq \left(1-2\varepsilon^{\star}\right)\left(\sqrt{C\left(N\right)} + \ln 2\left|\left(\varepsilon^{\star}\log\left(\varepsilon^{\star}\right) + \varepsilon^{\star}H\left(\varepsilon^{\star}\right)\right)\right| \cdot 2\left(N-1\right)\mathcal{D}_{\max}\right).$$

Finally, we can use the algebraic property that $\left(1-2\varepsilon^{\star}\right)\left|\ln 2\left(\varepsilon^{\star}\log\left(\varepsilon^{\star}\right) + \varepsilon^{\star}H\left(\varepsilon^{\star}\right)\right)\right| \leq \frac{1}{6}$, to get

$$\left|\mathbb{E}_{S,A(S)}\left[\mathcal{L}_{\mathcal{D}}\left(A\left(S\right)\right) \mid \text{consistent } S\right] - 2\varepsilon^{\star}\left(1-\varepsilon^{\star}\right)\right|$$
$$\leq \left(1-2\varepsilon^{\star}\right)\sqrt{C\left(N\right)} + \frac{\left(N-1\right)\mathcal{D}_{\max}}{3}.$$

$\square$

We can now bound the expected generalization error without conditioning on the consistency of the training set.

**Lemma C.4.** *It holds that,*

$$\left|\mathbb{E}_{S,A(S)}\left[\mathcal{L}_{\mathcal{D}}\left(A\left(S\right)\right)\right] - 2\varepsilon^{\star}\left(1-\varepsilon^{\star}\right)\right|$$
$$\leq \left|\mathbb{E}_{S,A(S)}\left[\mathcal{L}_{\mathcal{D}}\left(A\left(S\right)\right) \mid \text{consistent } S\right] - 2\varepsilon^{\star}\left(1-\varepsilon^{\star}\right)\right| + \mathbb{P}_{S}\left(\text{inconsistent } S\right).$$

*Proof.* Let $X$ be an arbitrary RV in $[0,1]$ and $Y$ be a binary RV. Then,

$$\mathbb{E}\left[X\right] = \mathbb{P}\left(Y\right)\mathbb{E}\left[X|Y\right] + \mathbb{P}\left(\neg Y\right)\mathbb{E}\left[X|\neg Y\right]$$
$$\mathbb{E}\left[X\right] - \mathbb{E}\left[X|Y\right] = \mathbb{P}\left(Y\right)\mathbb{E}\left[X|Y\right] - \mathbb{E}\left[X\mid Y\right] + \mathbb{P}\left(\neg Y\right)\mathbb{E}\left[X|\neg Y\right]$$
$$= \mathbb{E}\left[X|Y\right]\left(\mathbb{P}\left(Y\right) - 1\right) + \mathbb{P}\left(\neg Y\right)\mathbb{E}\left[X|\neg Y\right]$$
$$= -\mathbb{E}\left[X|Y\right]\mathbb{P}\left(\neg Y\right) + \mathbb{P}\left(\neg Y\right)\mathbb{E}\left[X|\neg Y\right] = \mathbb{P}\left(\neg Y\right)\left(\mathbb{E}\left[X|\neg Y\right] - \mathbb{E}\left[X|Y\right]\right)$$
$$\left|\mathbb{E}\left[X\right] - \mathbb{E}\left[X|Y\right]\right| = \mathbb{P}\left(\neg Y\right)\underbrace{\left|\mathbb{E}\left[X|\neg Y\right] - \mathbb{E}\left[X|Y\right]\right|}_{\leq 1} \leq \mathbb{P}\left(\neg Y\right).$$

As a result

$$\left|\mathbb{E}_{S,A(S)}\left[\mathcal{L}_{\mathcal{D}}\left(A\left(S\right)\right)\right] - \mathbb{E}_{S,A(S)}\left[\mathcal{L}_{\mathcal{D}}\left(A\left(S\right)\right) \mid \text{consistent } S\right]\right| \leq \mathbb{P}_{S}\left(\text{inconsistent } S\right).$$

Then, the required inequality is obtained by simply using the triangle inequality on

$$\left|\mathbb{E}_{S,A(S)}\left[\mathcal{L}_{\mathcal{D}}\left(A\left(S\right)\right)\right] - 2\varepsilon^{\star}\left(1-\varepsilon^{\star}\right)\right|.$$

$\square$

# D  Memorizing the label flips (proofs for Section 3)

In this section, we prove Theorem 3.1. We begin with an informal outline of the proof idea. Inspired by Manoj and Srebro's analysis [58], our proof of Theorem 3.1 is based on the concept of a *pseudorandom generator*, defined below.

**Definition D.1** (Pseudorandom generator). Let $G\colon \{0,1\}^r \to \{0,1\}^R$ be a function, let $\mathcal{V}$ be a class of functions $V\colon \{0,1\}^R \to \{0,1\}$, let $\mathcal{D}$ be a distribution over $\{0,1\}^R$, and let $\epsilon > 0$. We say that $G$ is an *$\epsilon$-pseudorandom generator* ($\epsilon$-PRG) for $\mathcal{V}$ with respect to $\mathcal{D}$ if for every $V \in \mathcal{V}$, we have

$$\left| \mathbb{P}_{\mathbf{y}\sim\mathcal{D}}(V(\mathbf{y}) = 1) - \mathbb{P}_{\mathbf{u}\in\{0,1\}^r}(V(G(\mathbf{u})) = 1) \right| \leq \epsilon,$$

where $\mathbf{u}$ is sampled uniformly at random from $\{0,1\}^r$.

To connect Definition D.1 to Theorem 3.1, let $R = 2^{d_0}$. Let $\mathcal{V}$ be the class of all conjunctions of literals, such as $V(\mathbf{y}) = \mathbf{y}_1 \wedge \bar{\mathbf{y}}_2 \wedge \mathbf{y}_4$. Let $\hat{\mathcal{X}} = f^{-1}(\{0,1\})$. There is a function $V_f \in \mathcal{V}$ such that given the entire truth table of a NN $\tilde{h}$, the function $V_f$ verifies that $\tilde{h}$ agrees with $f$ on $\hat{\mathcal{X}}$. This function $V_f$ is a conjunction of $N_1$ many variables and $(N - N_1)$ many negated variables.

Let $\alpha = N_1/N$, and let $\mathcal{D} = \mathrm{Ber}(\alpha)^R$. Suppose $G$ is an $\epsilon$-PRG for $\mathcal{V}$ with respect to $\mathcal{D}$, where $\epsilon < \mathbb{P}_{\mathbf{y}\sim\mathcal{D}}(V_f(\mathbf{y}) = 1)$. Then $\mathbb{P}_{\mathbf{u}\in\{0,1\}^r}(V_f(G(\mathbf{u})) = 1) \neq 0$, i.e., there exists some $\mathbf{u}^\star \in \{0,1\}^r$ such that $V_f(G(\mathbf{u}^\star)) = 1$. Therefore, if we let $\tilde{h}$ be a NN that computes the function

$$\tilde{h}(\mathbf{x}) = G(\mathbf{u}^\star)_{\mathbf{x}}, \tag{8}$$

then $\tilde{h}$ agrees with $f$ on $\hat{\mathcal{X}}$. In the equation above, $G(\mathbf{u}^\star)_{\mathbf{x}}$ denotes the $\mathbf{x}$-th bit of $G(\mathbf{u}^\star)$, thinking of $\mathbf{x}$ as a number from 0 to $R - 1$ represented by its binary expansion.

There is a large body of well-established techniques for constructing PRGs. (See, for example, Hatami and Hoza's recent survey [35].) Therefore, constructing a suitable PRG might seem like a promising approach to proving Theorem 3.1. However, this approach is flawed. The issue concerns the seed length ($r$). According to the plan outlined above, the seed $\mathbf{u}^\star$ is effectively hard-coded into the neural network $\tilde{h}$, which means that, realistically, the number of weights in $\tilde{h}$ will be at least $r$. Meanwhile, for the plan above to make sense, our PRG's error parameter ($\epsilon$) must satisfy

$$\epsilon < \mathbb{P}_{\mathbf{y}\sim\mathcal{D}}(V_f(\mathbf{y}) = 1) = 2^{-H(\alpha)\cdot N} \approx 2^{-\binom{N}{N_1}}. \tag{9}$$

Comparing (9) to Theorem 3.1, we see that we would need a PRG with seed length

$$r = (1 + o(1)) \cdot \log(1/\epsilon).$$

But this is too much to ask. There is no real reason to expect such a PRG to exist, even if we ignore explicitness considerations. Indeed, in some cases, it is provably impossible to achieve a seed length smaller than $(2 - o(1)) \cdot \log(1/\epsilon)$ [2].

To circumvent this issue, we will work with a more flexible variant of the PRG concept called a *hitting set generator* (HSG).

**Definition D.2** (Hitting set generator). Let $G\colon \{0,1\}^r \to \{0,1\}^R$ be a function, let $\mathcal{V}$ be a class of functions $V\colon \{0,1\}^R \to \{0,1\}$, let $\mathcal{D}$ be a distribution over $\{0,1\}^R$, and let $\epsilon > 0$. We say that $G$ is an *$\epsilon$-hitting set generator* ($\epsilon$-HSG) for $\mathcal{V}$ with respect to $\mathcal{D}$ if for every $V \in \mathcal{V}$ such that $\mathbb{P}_{\mathbf{y}\sim\mathcal{D}}(V(\mathbf{y}) = 1) > \epsilon$, there exists $\mathbf{u}^\star \in \{0,1\}^r$ such that $V(G(\mathbf{u}^\star)) = 1$.

Definition D.2 is weaker than Definition D.1, but an HSG is sufficient for our purposes. Crucially, one can show nonconstructively that for every $\mathcal{V}$, $\mathcal{D}$, and $\epsilon$, there exists an HSG with seed length

$$1 \cdot \log(1/\epsilon) + \log\log|\mathcal{V}| + O(1),$$

whereas the nonconstructive PRG seed length is $2 \cdot \log(1/\epsilon) + \cdots$. To prove Theorem 3.1, we will construct an *explicit* HSG for conjunctions of literals with respect to $\mathrm{Ber}(\alpha)^R$ with a seed length of $(1 + o(1)) \cdot \log(1/\epsilon) + \mathrm{polylog}\, R$. We will ensure that our HSG is "explicit enough" to enable computing the function $\tilde{h}$ defined by (8) using a constant-depth NN with approximately $r$ many weights.

Our HSG construction uses established techniques from the pseudorandomness literature. In brief, we use $k$-wise independence to construct an initial HSG with a poor dependence on $\epsilon$, and then we apply an error reduction technique due to Hoza and Zuckerman [38]. Details follow.

## D.1 Preprocessing the input to reduce the dimension

Before applying an HSG as outlined above, the first step of the proof of Theorem 3.1 is actually a preprocessing step that reduces the dimension to approximately $2 \log N$. This step is not completely essential, but it helps to improve the dependence on $d_0$ in Theorem 3.1. The preprocessing step is based on a standard trick, namely, we treat the input as a vector in $\mathbb{F}_2^{d_0}$ and apply a random matrix, where $\mathbb{F}_2$ denotes the field with two elements. That is:

**Definition D.3** ($\mathbb{F}_2$-linear and $\mathbb{F}_2$-affine functions). A function $C \colon \{0,1\}^d \to \{0,1\}^{d'}$ is $\mathbb{F}_2$-linear if it has the form
$$C(\mathbf{x}) = \mathbf{W}\mathbf{x},$$
where $\mathbf{W} \in \{0,1\}^{d' \times d}$ and the arithmetic is mod 2. More generally, we say that $C$ is $\mathbb{F}_2$-affine if it has the form
$$C(\mathbf{x}) = \mathbf{W}\mathbf{x} + \mathbf{b},$$
where $\mathbf{W} \in \{0,1\}^{d' \times d}$, $\mathbf{b} \in \{0,1\}^{d'}$, and the arithmetic is mod 2.

The following fact is standard; we include the proof only for completeness.

**Lemma D.4** (Preprocessing to reduce the dimension). *Let $d_0 \in \mathbb{N}$, let $\hat{\mathcal{X}} \subseteq \{0,1\}^{d_0}$, and let $N = |\hat{\mathcal{X}}|$. There exists an $\mathbb{F}_2$-linear function $C_0 \colon \{0,1\}^{d_0} \to \{0,1\}^{2\lceil \log N \rceil}$ that is injective on $\hat{\mathcal{X}}$.*

*Proof.* Pick $\mathbf{W} \in \{0,1\}^{2\lceil \log N \rceil \times d_0}$ uniformly at random and let $C_0(\mathbf{x}) = \mathbf{W}\mathbf{x}$. For each pair of distinct points $\mathbf{x}, \mathbf{x}' \in \hat{\mathcal{X}}$, we have
$$\mathbb{P}(\mathbf{W}\mathbf{x} = \mathbf{W}\mathbf{x}') = \mathbb{P}(\mathbf{W}(\mathbf{x} - \mathbf{x}') = \mathbf{0}) = 2^{-2\lceil \log N \rceil} < 1/N^2.$$
Therefore, by the union bound over all pairs $\mathbf{x}, \mathbf{x}'$, there is a nonzero chance that $C_0$ is injective on $\hat{\mathcal{X}}$. Therefore, there is some fixing of $\mathbf{W}$ such that $C_0$ is injective on $\hat{\mathcal{X}}$. $\square$

We will choose $C_0$ to be injective on the domain of $f$. That way, after applying $C_0$ to the input, our remaining task is to compute some other partial function $f' \colon \{0,1\}^{2\lceil \log N \rceil} \to \{0,1,\star\}$, namely, the function $f'$ such that $f' \circ C_0 = f$. This function $f'$ has the same domain size ($N$), and it takes the value 1 on the same number of points ($N_1$), so the net effect is that we have decreased the dimension from $d_0$ down to $2\lceil \log N \rceil$. This same technique appears in the circuit complexity literature, along with more sophisticated variants. For example, see Jukna's textbook [45, Section 1.4.2].

To apply Lemma D.4 in our setting, we rely on the well-known fact that $\mathbb{F}_2$-linear functions, and more generally $\mathbb{F}_2$-affine functions, can be computed by depth-two binary threshold networks. More precisely, we have the following.

**Lemma D.5** (Binary threshold networks computing $\mathbb{F}_2$-linear functions). *If $C \colon \{0,1\}^d \to \{0,1\}$ is the parity function or its negation, then there exists a depth-one binary threshold network $C_0 \colon \{0,1\}^d \to \{0,1\}^{(d+2)}$ and a number $b \in \mathbb{R}$ such that for every $\mathbf{x} \in \{0,1\}^d$, we have*
$$C(x) = \mathbf{1}^T C_0(\mathbf{x}) + b,$$
*where $\mathbf{1}$ denotes the all-ones vector. Moreover, every affine function $C \colon \{0,1\}^d \to \{0,1\}^{d'}$ can be computed by a depth-two binary threshold network with $d' \cdot (d+2)$ nodes in the hidden layer.*

*Proof.* First, suppose $C$ is the parity function. For each $i \in [d]$, let $\phi_{\leq i} \colon \{0,1\}^d \to \{0,1\}$ be the function
$$\phi_{\leq i}(\mathbf{x}) = 1 \iff \sum_{j=1}^d \mathbf{x}_j \leq i,$$
and similarly define $\phi_{\geq i} \colon \{0,1\}^d \to \{0,1\}$ by
$$\phi_{\geq i}(\mathbf{x}) = 1 \iff \sum_{j=1}^d \mathbf{x}_j \geq i.$$

Then

$$\phi_{\leq 1}(\mathbf{x}) + \phi_{\geq 1}(\mathbf{x}) + \phi_{\leq 3}(\mathbf{x}) + \phi_{\geq 3}(\mathbf{x}) + \cdots = \begin{cases} \lceil d/2 \rceil + 1 & \text{if PARITY}(\mathbf{x}) = 1 \\ \lceil d/2 \rceil & \text{if PARITY}(\mathbf{x}) = 0, \end{cases}$$

so we can take $b = -\lceil d/2 \rceil$. Now, suppose $C$ is the negation of the parity function. This reduces to the case of the parity function because $1 - \text{PARITY}(\mathbf{x}) = \text{PARITY}(\mathbf{x}, 1)$. Finally, the "moreover" statement follows because if $C$ is $\mathbb{F}_2$-affine, then every output bit of $C$ is either the parity function or the negated parity function applied to some subset of the inputs. $\qquad\square$

Lemma D.5 can be generalized to the case of any symmetric function instead of PARITY. This technique is well-known in the circuit complexity literature; for example, see the work of Hajnal, Maass, Pudlák, Szegedy, and Turán [32].

## D.2 Threshold networks computing $k$-wise independent generators

One of the ingredients of our HSG will be a family of pairwise independent hash functions. We will use the following family, notable for its low computational complexity.

**Lemma D.6** (Affine pairwise independent hash functions). *For every $a, r \in \mathbb{N}$, there is a family $\mathcal{H}$ of hash functions* $\text{hash} \colon \{0,1\}^a \to \{0,1\}^r$ *with the following properties.*

- $|\mathcal{H}| \leq 2^{O(a+r)}$.

- $\mathcal{H}$ *is pairwise independent. That is, for every two distinct* $\mathbf{w}, \mathbf{w}' \in \{0,1\}^a$, *if we pick* $\text{hash} \in \mathcal{H}$ *uniformly at random, then* $\text{hash}(\mathbf{w})$ *and* $\text{hash}(\mathbf{w}')$ *are independent and uniformly distributed over* $\{0,1\}^r$.

- *Each function* $\text{hash} \in \mathcal{H}$ *is* $\mathbb{F}_2$*-affine.*

*Proof.* See the work of Mansour, Nisan, and Tiwari [59, Claim 2.2]. $\qquad\square$

*Remark* D.7 (Alternative hash families). By Lemma D.5, each function $\text{hash} \in \mathcal{H}$ can be computed by a depth-two binary threshold network with $O(r^2 a + r a^2)$ wires (weights). There exist alternative pairwise independent hash function families with lower wire complexity. In particular, one could use hash functions based on integer arithmetic [24], which can be implemented with wire complexity $(a + r)^{1+\gamma}$ for any arbitrarily small constant $\gamma > 0$ [73]. This would lead to slightly better width and wire complexity bounds in Theorem 3.1: each occurrence of $3/4$ could be replaced with $2/3 + \gamma$. However, the downside of this approach is that the depth of the network would increase to a very large constant depending on $\gamma$.

Another ingredient of our HSG will be a threshold network computing a "$k$-wise uniform generator," defined below.

**Definition D.8** ($k$-wise uniform generator). A $k$-*wise uniform generator* is a function $G \colon \{0,1\}^r \to \{0,1\}^R$ such that if we sample $\mathbf{u} \in \{0,1\}^r$ uniformly at random, then every $k$ of the output coordinates of $G(\mathbf{u})$ are independent and uniform. In other words, $G$ is a 0-PRG for $\mathcal{V}$ with respect to the uniform distribution, where $\mathcal{V}$ consists of all Boolean functions that only depend on $k$ bits.

Prior work has shown that $k$-wise uniform generators can be implemented by constant-depth threshold networks [36]. We will need to re-analyze the construction to get sufficiently fine-grained bounds. In the remainder of this subsection, we will prove the following.

**Lemma D.9** (Constant-depth $k$-wise uniform generator). *Let $k, R \in \mathbb{N}$ where $R$ is a power of two. There exists a $k$-wise uniform generator $G \colon \{0,1\}^r \to \{0,1\}^R$, where $r = O(k \cdot \log R)$, such that for every $\mathbb{F}_2$-affine function $\mathrm{hash} \colon \{0,1\}^a \to \{0,1\}^r$, there exists a depth-5 binary threshold network $C \colon \{0,1\}^{a + \log R} \to \{0,1\}^{k \cdot \mathrm{polylog}\, R}$ with widths $\underline{d}$ satisfying the following.*

1. *For every $\mathbf{w} \in \{0,1\}^a$ and every $\mathbf{z} \in \{0,1\}^{\log R}$, we have*

$$G(\mathrm{hash}(\mathbf{w}))_{\mathbf{z}} = \mathsf{PARITY}(C(\mathbf{w}, \mathbf{z})),$$

*thinking of $\mathbf{z}$ as a number in $\{0, 1, \ldots, R - 1\}$.*

2. *The maximum width $\underline{d}_{\max}$ is at most $ak \cdot \mathrm{polylog}\, R$.*

3. *The total number of weights $w\,(\underline{d})$ is at most $(a + k) \cdot ak \cdot \mathrm{polylog}\, R$.*

*Remark* D.10 (The role of the parity functions). One can combine Lemma D.9 with Lemma D.5 to obtain threshold networks computing the function $(\mathbf{u}, \mathbf{z}) \mapsto G(\mathbf{u})_{\mathbf{z}}$. In Lemma D.9, instead of describing a network that computes the function $(\mathbf{u}, \mathbf{z}) \mapsto G(\mathbf{u})_{\mathbf{z}}$, we describe a network $C$ satisfying $G(\mathrm{hash}(\mathbf{w}))_{\mathbf{z}} = \mathsf{PARITY}(C(\mathbf{w}, \mathbf{z}))$. The only reason for this more complicated statement is that it leads to a slightly better depth complexity in Theorem 3.1.

We reiterate that the proof of Lemma D.9 heavily relies on prior work. For the most part, this prior work studies a Boolean circuit model that is closely related to, but distinct from, the "binary threshold network" model in which we are interested. We introduce the circuit model next.

**Definition D.11** ($\widehat{\mathrm{LT}}_L$ circuits). An $\widehat{\mathrm{LT}}_L$ circuit is defined just like a depth-$L$ binary threshold network (Definition 2.1), except that we allow arbitrary integer weights ($\mathbf{W}^{(l)} \in \mathbb{Z}^{d_l \times d_{l-1}}$); we allow arbitrary integer thresholds ($\mathbf{b}^{(l)} \in \mathbb{Z}^{d_l}$); and we do not allow any scaling ($\gamma^{(l)} = \mathbf{1}^{d_l}$). The *size* of the circuit is the sum of the absolute values of the weights, i.e.,

$$\sum_{l=1}^{L} \sum_{i=1}^{d_l} \sum_{j=1}^{d_{l-1}} |\mathbf{W}_{ij}^{(l)}|.$$

*Remark* D.12 (Parallel wires). In the context of circuit complexity, it is perhaps more natural to stipulate that the weights are always $\{\pm 1\}$; there can be any number of *parallel wires* between two nodes, including zero; and the size of the circuit is the total number of wires. This is completely equivalent to Definition D.11.

The proof of Lemma D.9 relies on circuits performing arithmetic. A long line of research investigated the depth complexity of (iterated) integer multiplication [16, 9, 69, 78, 37, 73, 32, 80, 79], culminating in the following result by Siu and Roychowdhury [79].

**Theorem D.13** (Iterated multiplication in depth four [79]). *For every $n \in \mathbb{N}$, there exists an $\widehat{\mathrm{LT}}_4$ circuit of size $\mathrm{poly}(n)$ that computes the product of $n$ given $n$-bit integers.*

By a standard trick [26], Theorem D.13 implies circuits of the same complexity that compute the iterated product of *polynomials* over $\mathbb{F}_2$. We include a proof sketch for completeness.

**Corollary D.14** (Iterated multiplication of polynomials over $\mathbb{F}_2$). *For every $n \in \mathbb{N}$, there exists an $\widehat{\mathrm{LT}}_4$ circuit of size $\mathrm{poly}(n)$ that computes the product of $n$ given polynomials in $\mathbb{F}_2[x]$, each of which has degree less than $n$ and is represented by an $n$-bit vector of coefficients.*

*Proof sketch.* Think of the given polynomials as polynomials over $\mathbb{Z}$, say $q_1(x), \ldots, q_n(x)$. If we evaluate one of these polynomials on a power of two, say $q_i(2^s)$, and then write the output in binary, the resulting string consists of the coefficients of $q_i$, with $s - 1$ zeroes inserted between every two bits. Therefore, by using the $\mathrm{poly}(ns)$-size circuit of Theorem D.13 (with some of its inputs fixed to zeroes), we can compute the product $q_1(2^s) \cdot q_2(2^s) \cdots q_n(2^s) = q(2^s)$, where $q = q_1 \cdot q_2 \cdots q_n$. Every coefficient of $q$ is a nonnegative integer bounded by $n^n$, so if we choose $s = \lceil n \log n \rceil$, then the binary expansion of $q(2^s)$ is the concatenation of all of the binary expansions of the coefficients of $q$. To reduce mod 2, we simply discard all but the lowest-order bit of each of those coefficients. $\qquad\square$

At this point, we are ready to construct a circuit that computes a $k$-wise uniform generator. The construction is based on the work of Healy and Viola [36].

**Lemma D.15** (A $k$-wise uniform generator in the $\widehat{\mathrm{LT}}_L$ model). *Let $k, R \in \mathbb{N}$ where $R$ is a power of two. There exists a $k$-wise uniform generator $G\colon \{0,1\}^r \to \{0,1\}^R$, an $\mathbb{F}_2$-linear function $C_0\colon \{0,1\}^{\log R} \to \{0,1\}^{O(\log R \cdot \log k)}$, and an $\widehat{\mathrm{LT}}_4$ circuit $C_1\colon \{0,1\}^{O(k \cdot \log R)} \to \{0,1\}^{O(k \cdot \log R)}$ with the following properties.*

- *The seed length is $r = O(k \cdot \log R)$.*

- *For every seed $\mathbf{u} \in \{0,1\}^r$ and every $\mathbf{z} \in \{0,1\}^{\log R}$, we have*
$$G(\mathbf{u})_{\mathbf{z}} = \mathrm{PARITY}(C_1(\mathbf{u}, C_0(\mathbf{z}))),$$
  *thinking of $\mathbf{z}$ as a number in $\{0, 1, \ldots, R-1\}$.*

- *The circuit $C_1$ has size $k \cdot \mathrm{polylog}\, R$.*

*Proof.* If $k \geq R$, the lemma is trivial, so assume $k < R$. We use the following standard example of a $k$-wise independent generator [22, 1]. Let $n = \log R$, let $E(x) \in \mathbb{F}_2[x]$ be an irreducible polynomial of degree $n$, and let $\mathbb{F}_{2^n}$ be the finite field consisting of all polynomials in $\mathbb{F}_2[x]$ modulo $E(x)$. The seed of the generator is interpreted as a list of field elements: $\mathbf{u} = (p_0, p_1, \ldots, p_{k-1}) \in \mathbb{F}_{2^n}^k$. Each index $\mathbf{z} \in \{0, 1, \ldots, R-1\}$ can be interpreted as a field element $\mathbf{z} \in \mathbb{F}_{2^n}$. The output of the generator is given by

$$G(\mathbf{u})_{\mathbf{z}} = \text{the lowest order bit of } p_0 \cdot \mathbf{z}^0 + p_1 \cdot \mathbf{z}^1 + \cdots + p_{k-1} \cdot \mathbf{z}^{k-1},$$

where the arithmetic takes place in $\mathbb{F}_{2^n}$.

To study the circuit complexity of this generator, let us first focus on a single term $p_i \cdot \mathbf{z}^i$. Since we are thinking of $\mathbf{z}$ as a field element $\mathbf{z} \in \mathbb{F}_{2^n}$, we can also think of it as a polynomial $\mathbf{z}(x) \in \mathbb{F}_2[x]$ of degree less than $n$. Write $\mathbf{z}(x) = \sum_{j=0}^{n-1} \mathbf{z}_j \cdot x^j$. We compute the power $\mathbf{z}^i$ by a "repeated squaring" approach. Write $i = \sum_{m \in M} 2^m$, where $M \subseteq \{0, 1, \ldots, \lfloor \log i \rfloor\}$. Then

$$p_i(x) \cdot \mathbf{z}(x)^i = p_i(x) \cdot \prod_{m \in M} \left( \sum_{j=0}^{n-1} \mathbf{z}_j \cdot x^j \right)^{2^m} = p_i(x) \cdot \prod_{m \in M} \sum_{j=0}^{n-1} \mathbf{z}_j \cdot x^{j \cdot 2^i},$$

since we are working in characteristic two. For each $m \in M$ and each $j < n$, let $e_{m,j}(x) = x^{j \cdot 2^m} \bmod E(x)$, a polynomial of degree less than $n$. That way,

$$p_i(x) \cdot \mathbf{z}(x)^i \equiv p_i(x) \cdot \prod_{m \in M} \sum_{j=0}^{n-1} \mathbf{z}_j \cdot e_{m,j}(x) \pmod{E(x)}. \tag{10}$$

The function $C_0(\mathbf{z})$ computes $\sum_{j=0}^{n-1} \mathbf{z}_j \cdot e_{m,j}$ for every $m \in \{0, 1, \ldots, \lfloor \log k \rfloor\}$. This function is $\mathbb{F}_2$-linear, because for each $m \in \{0, 1, \ldots, \lfloor \log k \rfloor\}$ and each $s < n$, the $s$-th bit of $\sum_{j=0}^{n-1} \mathbf{z}_j \cdot e_{m,j}$ is given by

$$\bigoplus_{j : e_{m,j,s} = 1} \mathbf{z}_j,$$

where $e_{m,j,s}$ denotes the $s$-th coefficient of the polynomial $e_{m,j}$.

Next, the circuit $C_1$ applies $k$ copies of the iterated multiplication circuit from Corollary D.14, in parallel, to compute the polynomial on the right-hand side of (10) for each $0 \leq i < k$. Each iterated multiplication circuit has size $\mathrm{polylog}\, R$, so altogether, $C_1$ has size $k \cdot \mathrm{polylog}\, R$.

At this point, we have computed polynomials $r_0, r_1, \ldots, r_{k-1} \in \mathbb{F}_2[x]$, each of degree $O(n \log k)$, such that $r_i(x) \equiv p_i(x) \cdot \mathbf{z}(x)^i \pmod{E(x)}$. Next, we need to sum these terms up, reduce mod $E(x)$, and output the lowest-order bit. For each $j \leq O(n \log k)$, let $r_{ij}$ be the $x^j$ coefficient of $r_i$. Our circuit needs to output the lowest-order bit of

$$\sum_{i=0}^{k-1} r_i \bmod E(x) = \sum_{i=0}^{k-1} \sum_{j=0}^{O(n \log k)} r_{ij} \cdot e_{0,j}.$$

Now, we are working over characteristic two, so $\sum$ means bitwise XOR. In other words, the output is given by

$$\bigoplus_{j:e_{0,j,0}=1} \bigoplus_{i=0}^{k-1} r_{ij},$$

i.e., it is the parity function applied to some subset of the output bits of $C_1$. To complete the proof, modify $C_1$ by deleting the unused output gates. $\qquad\square$

We have almost completed the proof of Lemma D.9. The last step is to bridge the gap between $\widehat{\mathrm{LT}}_L$ circuits and binary threshold networks. We do so via the following lemma.

**Lemma D.16** (Simulating $\widehat{\mathrm{LT}}_L$ circuits using binary threshold networks)**.** *Let $L \geq 1$ be a constant. Let $C_0 \colon \{0,1\}^{d_0} \to \{0,1\}^{d_1}$ be an $\mathbb{F}_2$-affine function, and let $C_1 \colon \{0,1\}^{d_1} \to \{0,1\}^{d_2}$ be an $\widehat{\mathrm{LT}}_L$ circuit of size $S$. Then the composition $C_1 \circ C_0$ can be computed by a depth-$(L+1)$ binary threshold network with widths $\underline{d}$ satisfying the following.*

1. *The maximum width $\underline{d}_{\max}$ is at most $S \cdot (d_0 + 2)$.*

2. *The total number of weights $w\,(\underline{d})$ is at most $O(S^2 d_0 + S d_0^2)$.*

*Proof.* Let us define the *cost* of a layer in an $\widehat{\mathrm{LT}}_L$ circuit to be the sum of the absolute values of the weights in that layer, so the size of the circuit is the sum of the costs. Lemma D.5 implies that $C_1 \circ C_0 = C_1' \circ C_0'$, where $C_0'$ is a depth-one binary threshold network and $C_1'$ is an $\widehat{\mathrm{LT}}_L$ circuit in which the first layer has cost at most $S \cdot (d_0 + 2)$ and all subsequent layers have cost at most $S$.

To complete the proof, let us show by induction on $L$ that in general, if $C_0'$ is a depth-one binary threshold network and $C_1'$ is an $\widehat{\mathrm{LT}}_L$ circuit in which the layers have costs $S_1, S_2, \ldots, S_L$, then $C_1' \circ C_0'$ can be computed by a depth-$(L+1)$ binary threshold network in which the layers after the input layer have widths $S_1, S_2, \ldots, S_L$. Let us write $C_1'$ as $C_3 \circ C_2$, where $C_3$ is the last layer of $C_1'$. By induction, $C_2 \circ C_0'$ can be computed by a depth-$L$ binary threshold network $C$ in which the layers after the input layer have widths $S_1, S_2, \ldots, S_{L-1}$. Now let us modify $C_3$ and $C$ so that every wire in $C_3$ has weight either $0$ or $1$. If a wire in $C_3$ has an integer weight $w \notin \{0, 1\}$, then we make $|w|$ many copies of the appropriate output gate of $C$, negate them if $w < 0$, and then split the wire into $|w|$ many wires, each with weight $+1$. This process has no effect on the cost of $C_3$. The process could potentially increase the width of the output layer of $C$, but its width will not exceed $S_L$, the cost of $C_3$. After this modification, we can simply think of $C_3$ as one more layer in our binary threshold network. $\qquad\square$

Lemma D.9 follows immediately from Lemmas D.15 and D.16.

## D.3 A hitting set generator with a non-optimal dependence on $\epsilon$

In this subsection, we will use the $k$-wise independent generators that we developed in the previous subsection to construct our first HSG:

**Lemma D.17** (Non-optimal HSG for conjunctions of literals). *Let $R$ be a power of two and let $\alpha, \epsilon \in (0,1)$. Assume that $1/R \leq \alpha \leq 1 - 1/R$. Let $\mathcal{V}$ be the class of functions $V: \{0,1\}^R \to \{0,1\}$ that can be expressed as a conjunction of literals. There exists a generator $G: \{0,1\}^r \to \{0,1\}^R$ satisfying the following.*

1. *For every $V \in \mathcal{V}$, if $\mathbb{P}_{\mathbf{y} \sim \mathrm{Ber}(\alpha)^R}(V(y) = 1) \geq 2\epsilon$, then $\mathbb{P}_{\mathbf{u} \in \{0,1\}^r}(V(G(\mathbf{u})) = 1) \geq \epsilon$.*

2. *The seed length is $r = O(\log(1/\epsilon) \cdot \log^2 R)$.*

3. *For every $\mathbb{F}_2$-affine function $\mathrm{hash}: \{0,1\}^a \to \{0,1\}^r$, the function $C(\mathbf{w}, \mathbf{z}) = G(\mathrm{hash}(\mathbf{w}))_{\mathbf{z}}$ can be computed by a depth-8 binary threshold network with widths $\underline{d}$ such that the maximum width $\underline{d}_{\max}$ at most $a \cdot \log(1/\epsilon) \cdot \mathrm{polylog}\, R$ and the total number of weights $w\,(\underline{d})$ is at most $(\log(1/\epsilon) \cdot a^2 + \log^2(1/\epsilon) \cdot a) \cdot \mathrm{polylog}\, R$.*

*Remark* D.18. The parameters of Lemma D.17 are not yet sufficient to prove Theorem 3.1. Remember, we need the number of weights to be only $(1 + o(1)) \cdot \log(1/\epsilon)$. On the other hand, Item 1 is stronger than what the HSG definition requires. This will enable us to improve the seed length of the generator later.

The proof of Lemma D.17 is based on the work of Even, Goldreich, Luby, Nisan, and Veličković [27]. In particular, we use the following lemma from their work.

**Lemma D.19** (Implications of $k$-wise independence [27]). *Let $X_1, \ldots, X_R$ be independent $\{0,1\}$-valued random variables. Let $\tilde{X}_1, \ldots, \tilde{X}_R$ be $k$-wise independent $\{0,1\}$-valued random variables such that $\tilde{X}_i$ is distributed identically to $X_i$ for every $i$. Then*

$$|\mathbb{P}(X_1 = X_2 = \cdots = X_R = 1) - \mathbb{P}(\tilde{X}_1 = \tilde{X}_2 = \cdots = \tilde{X}_R = 1)| \leq 2^{-\Omega(k)}.$$

*Proof of Lemma D.17.* Let $Q$ be the smallest positive integer such that $Q \geq 4R^2$ and $\log \log Q$ is an integer. Let $\phi: \{0, 1, \ldots, Q-1\} \to \{0,1\}$ be the function

$$\phi(x) = 1 \iff x \leq \alpha \cdot Q.$$

We think of $\phi$ as a function $\phi: \{0,1\}^{\log Q} \to \{0,1\}$ by representing $x$ in binary.

Let $\vec{\phi}: \{0,1\}^{R \log Q} \to \{0,1\}^R$ be $R$ copies of $\phi$ applied to $R$ disjoint input blocks. Let $G_0: \{0,1\}^r \to \{0,1\}^{R \log Q}$ be a $k$-wise independent generator for a suitable value $k = O(\log(1/\epsilon) \cdot \log R)$. Our generator $G$ is the composition $\vec{\phi} \circ G_0$.

Now let us prove that $G$ has the claimed properties. The seed length bound is clear. Now let us analyze the computational complexity of $G$. To compute $G(\mathrm{hash}(\mathbf{w}))_{\mathbf{z}}$, we begin by computing $C_1(\mathbf{w}, \mathbf{z} \log Q + i)$ for every $i \in \{0, 1, \ldots, \log Q - 1\}$, all in parallel, where $C_1$ is the depth-5 network from Lemma D.9. Since $\log Q$ is a power of two, the binary expansions of the numbers $\mathbf{z} \log Q, \mathbf{z} \log Q + 1, \mathbf{z} \log Q + 2, \ldots, \mathbf{z} \log Q + \log Q - 1$ simply consist of $\mathbf{z}$ followed by all possible bitstrings of length $\log \log Q$. The maximum width of one of these layers is bounded by $ak \cdot \mathrm{polylog}\, R = a \cdot \log(1/\epsilon) \cdot \mathrm{polylog}\, R$, and the total number of weights among these layers is at most $(a + k) \cdot ak \cdot \mathrm{polylog}\, R = (a + \log(1/\epsilon)) \cdot a \cdot \log(1/\epsilon) \cdot \mathrm{polylog}\, R$. Furthermore, the number of output bits is $\log(1/\epsilon) \cdot \mathrm{polylog}\, R$.

Next, recall that to compute a single bit of the output of $G_0$, we need to apply the parity function to the outputs of $C_1$. Therefore, to compute an output bit of our generator $G$, we need to apply an $\mathbb{F}_2$-linear function followed by $\phi$. Observe that $\phi$ can be computed by a depth-two "$\mathrm{AC}^0$ circuit," i.e., a circuit consisting of unbounded-fan-in AND and OR gates applied to literals, in which the total number of gates is $O(\log Q) = O(\log R)$. This can be viewed as a special case of an $\widehat{\mathrm{LT}}_2$ circuit of size $O(\log^2 R)$. Therefore, by Lemma D.16, the $\mathbb{F}_2$-linear function followed by $\phi$ can be computed by a depth-3 binary threshold network in which every layer has width at most $\log(1/\epsilon) \cdot \mathrm{polylog}\, R$ and the total number of weights is at most $\log^2(1/\epsilon) \cdot \mathrm{polylog}\, R$. This completes the analysis of the computational complexity of $G$.

Next, let us prove the correctness of $G$, i.e., let us prove Item 1 of Lemma D.17. Let $V \in \mathcal{V}$ and assume $\mathbb{P}_{\mathbf{y} \sim \mathrm{Ber}(\alpha)^R}(V(\mathbf{y}) = 1) \geq 2\epsilon$. Since $V$ is a conjunction of literals, we can write $V(\mathbf{y}) = V_1(\mathbf{y}_1) \cdot V_2(\mathbf{y}_2) \cdots V_R(\mathbf{y}_R)$ for some functions $V_1, V_2, \ldots, V_R \colon \{0,1\} \to \{0,1\}$.

We will analyze $\mathbb{P}_{\mathbf{u} \in \{0,1\}^r}(V(G(\mathbf{u})) = 1)$ in two stages. First, we compare $V(\vec{\phi}(G_0(\mathbf{u})))$ to $V(\vec{\phi}(\bar{\mathbf{y}}))$, where $\bar{\mathbf{y}} \in \{0,1\}^{R \log Q}$ is uniform random. Then, in the second stage, we will compare $V(\vec{\phi}(\bar{\mathbf{y}}))$ to $V(\mathbf{y})$, where $\mathbf{y} \sim \mathrm{Ber}(\alpha)^R$.

For the first stage, we are in the situation of Lemma D.19, because the $R$ many $(\log Q)$-bit blocks of $G_0(\mathbf{u})$ are $(k/\log Q)$-wise independent. Therefore,

$$\left| \mathbb{P}_{\mathbf{u} \in \{0,1\}^r}(V(G(\mathbf{u})) = 1) - \mathbb{P}_{\bar{\mathbf{y}} \in \{0,1\}^{R \log Q}}(V(\vec{\phi}(\bar{\mathbf{y}})) = 1) \right| \leq \exp\left( -\Omega\left( \frac{k}{\log Q} \right) \right) \leq 0.5\epsilon,$$

provided we choose a suitable value $k = O(\log(1/\epsilon) \cdot \log R)$.

Now, for the second stage, observe that if we sample $\bar{\mathbf{y}} \in \{0,1\}^{\log Q}$ uniformly at random, then $|\mathbb{P}(\phi(\bar{\mathbf{y}}) = 1) - \alpha| \leq \frac{1}{Q} \leq \frac{1}{4R^2}$. For each $i$, since $1/R \leq \alpha \leq 1 - 1/R$, we have

$$\mathbb{P}_{\bar{\mathbf{y}} \in \{0,1\}^{\log Q}}(V_i(\phi(\bar{\mathbf{y}})) = 1) \geq \mathbb{P}_{\mathbf{y} \sim \mathrm{Ber}(\alpha)}(V_i(\mathbf{y}) = 1) - \frac{1}{4R^2}$$

$$\geq \left( 1 - \frac{1}{4R} \right) \cdot \mathbb{P}_{\mathbf{y} \sim \mathrm{Ber}(\alpha)}(V_i(\mathbf{y}) = 1).$$

Therefore,

$$\mathbb{P}_{\bar{\mathbf{y}} \in \{0,1\}^{R \log Q}}(V(\vec{\phi}(\bar{\mathbf{y}})) = 1) \geq \left( 1 - \frac{1}{4R} \right)^R \cdot \mathbb{P}_{\mathbf{y} \sim \mathrm{Ber}(\alpha)^R}(V(\mathbf{y}) = 1)$$

$$\geq 1.5\epsilon$$

by Bernoulli's inequality. Combining the bounds from the two stages completes the proof. $\qquad \square$

### D.4 Networks for computing functions that are constant on certain intervals

At this point, we have constructed an HSG for conjunctions of literals with a non-optimal dependence on the threshold parameter $\epsilon$ (Lemma D.17). To improve the dependence on $\epsilon$, we will use a technique introduced by Hoza and Zuckerman [38]. They introduced this "error-reduction" technique in the context of derandomizing general space-bounded algorithms, but it is simpler in our context (conjunctions of literals).

The basic idea is as follows. Let $V$ be a conjunction of literals with a low acceptance probability: $\mathbb{P}_{\mathbf{y} \sim \mathrm{Ber}(\alpha)^R}(V(\mathbf{y}) = 1) = \epsilon$. We will split $V$ up as a product,

$$V(\mathbf{y}) = V^{(0)}(\mathbf{y}^{(0)}) \cdot V^{(1)}(\mathbf{y}^{(1)}) \cdots V^{(T-1)}(\mathbf{y}^{(T-1)}),$$

where each $V^{(i)}$ is a conjunction of literals with a considerably higher acceptance probability:

$$\mathbb{P}_{\mathbf{y}^{(i)} \sim \mathrm{Ber}(\alpha)^{R_i}}(V^{(i)}(\mathbf{y}^{(i)}) = 1) \approx \epsilon_0 \gg \epsilon.$$

We choose $V^{(0)}$ to be the conjunction of the first few literals in $V$; $V^{(1)}$ is the conjunction of the next few literals; etc. To hit a single $V^{(i)}$, we can use our initial HSG with a relatively high threshold parameter ($\epsilon_0$). Then, we use pairwise independent hash functions to "recycle" the seed of our initial HSG from one $V^{(i)}$ to the next.

To implement this technique, one of the ingredients we need is a network that figures out which block $V^{(i)}$ contains a particular given index $\mathbf{z} \in \{0, 1, \ldots, R-1\}$. In this subsection, we describe networks that handle that key ingredient. The constructions are elementary and straightforward.

First, we review standard circuits for integer comparisons.

**Lemma D.20** (DNFs for comparing integers). *Let $R$ be a power of two, let $I \subseteq [0, R)$ be an interval, and let $g_I \colon \{0,1\}^{\log R} \to \{0,1\}$ be the indicator function for $I \cap \{0, 1, \ldots, R-1\}$ (identifying numbers with their binary expansions). Then $g_I$ can be expressed as a DNF formula consisting of $O(\log^2 R)$ terms.*

*Proof.* First, consider the case that $I = [0, r)$ for some $r \in \{1, 2, \ldots, R\}$. If $r = R$, then the lemma is trivial, so assume $r < R$. Let $S$ be the set of indices at which the binary expansion of $r$ has a one. For each $i \in S$, we introduce a term that asserts that the input $\mathbf{z}$ agrees with the binary expansion of $r$ prior to position $i$, and then $\mathbf{z}$ has a zero in position $i$. The disjunction of these $|S|$ many terms computes $g_I$.

The case $I = [\ell, R)$ for some $\ell \in \{0, 1, \ldots, R - 1\}$ is symmetric. Finally, in the general case, we can assume that $I$ is an intersection of an interval of the form $[\ell, R)$ with an interval of the form $[0, r)$. Therefore, $g_I$ can be expressed in the form $\mathsf{AND}_2 \circ \mathsf{OR}_{\log R} \circ \mathsf{AND}_{\log R}$, where $\mathsf{AND}_k$ / $\mathsf{OR}_k$ denotes an AND / OR gate with fan-in $k$. To complete the proof, observe that every $\mathsf{AND}_a \circ \mathsf{OR}_b$ formula can be re-expressed as an $\mathsf{OR}_{b^a} \circ \mathsf{AND}_a$ formula. $\square$

**Lemma D.21** (Computing a function that is constant on intervals)**.** *Let $T$ and $R$ be powers of two. Suppose the interval $[0, R)$ has been partitioned into $T$ subintervals, say $[0, R) = I_0 \cup I_1 \cup \cdots \cup I_{T-1}$. Let $g \colon \{0, 1, \ldots, R-1\} \to \{0, 1\}^a$ be a function that is constant on each subinterval $I_j$. Then for every $\mathbb{F}_2$-affine function $C_0 \colon \{0, 1\}^{d_0} \to \{0, 1\}^{\log R}$, there is a depth-6 binary threshold network $C \colon \{0, 1\}^{d_0} \to \{0, 1\}^{a + \log R}$ with widths $\underline{d}$ satisfying the following.*

1. *For every $\mathbf{x} \in \{0, 1\}^{d_0}$, we have*

$$C(\mathbf{x}) = (g(C_0(\mathbf{x})), C_0(\mathbf{x})).$$

2. *The maximum width $\underline{d}_{\max}$ is at most $O(T \cdot \log^3 R + a + d_0 \cdot \log R)$.*

3. *The total number of weights $w\,(\underline{d})$ is at most*

$$aT + O(T \cdot \log^4 R + d_0^2 \cdot \log R + d_0 \cdot \log^2 R + a \cdot \log R).$$

We emphasize that the leading term in the weights bound is $aT$, with a coefficient of 1. This is crucial. It is also important that the weights bound has only a linear dependence on $T$, the number of intervals.

*Proof.* We begin by computing $C_0(\mathbf{x})$ and the negations of all of those bits. By Lemma D.5, we can compute these bits using a depth-two network where the hidden layer has width $O(d_0 \cdot \log R)$ and the output layer has width $O(\log R)$.

Let $\mathbf{z} = C_0(\mathbf{x}) \in \{0, 1\}^{\log R}$, and think of $\mathbf{z}$ as a number $\mathbf{z} \in \{0, 1, \ldots, R - 1\}$. Our next goal is to compute the $(\log T)$-bit binary expansion of the unique $j_* \in \{0, 1, \ldots, T - 1\}$ such that $\mathbf{z} \in I_{j_*}$. To do so, for each position $i \in \{0, 1, \ldots, \log T - 1\}$, let $S_i$ be the set of $j \in \{0, 1, \ldots, T - 1\}$ such that $j$ has a 1 in position $i$ of its binary expansion. We have a disjunction, over all $j \in S_i$, of the DNF computing $g_{I_j}$ from Lemma D.20. We also compute all the negations of the bits of $j_*$, and we also copy $\mathbf{z}$. Altogether, this is a depth-two network where the hidden layer has width $O(T \cdot \log T \cdot \log^2 R) = O(T \cdot \log^3 R)$ and the output layer has width $O(\log R)$.

Our final goal is to compute $g(\mathbf{z})$, which can be written in the form $g'(j_*)$ since $g$ is constant on each subinterval. We use a "brute-force DNF" to compute $g'$. First, for every $j \in \{0, 1, \ldots, T - 1\}$, we have an AND gate that checks whether $j_* = j$. Then each output bit of $g'$ is a disjunction of some of those AND gates. We also copy $\mathbf{z}$. Altogether, this is a depth-two network where the hidden layer has width $T + \log R$ and the output layer has width $a + \log R$. $\square$

## D.5 Error reduction

In this subsection, we improve our HSG's dependence on $\epsilon$, as described in the previous subsection. The following theorem should be compared to Lemma D.17. As discussed previously, the proof is based on a technique due to Hoza and Zuckerman [38].

**Theorem D.22** (HSG with near-optimal dependence on $\epsilon$). *Let $R$ be a power of two and let $\alpha, \epsilon \in (0, 1)$. Assume that $1/R \leq \alpha \leq 1 - 1/R$. Let $\mathcal{V}$ be the class of functions $V: \{0,1\}^R$ that can be expressed as a conjunction of literals. There exists a generator $G: \{0,1\}^r \to \{0,1\}^R$ satisfying the following.*

1. *$G$ is an $\epsilon$-HSG for $\mathcal{V}$ with respect to $\mathrm{Ber}(\alpha)^R$. That is, if $\mathbb{P}_{\mathbf{y} \sim \mathrm{Ber}(\alpha)^R}(V(\mathbf{y}) = 1) > \epsilon$ for every $V \in \mathcal{V}$, then there exists a seed $\sigma \in \{0,1\}^r$ such that $V(G(\sigma)) = 1$.*

2. *The seed length is $r = \log(1/\epsilon) + \log^{3/4}(1/\epsilon) \cdot \mathrm{polylog}\, R$.*

3. *For every $\mathbb{F}_2$-affine function $C_0: \{0,1\}^{d_0} \to \{0,1\}^{\log R}$ and every fixed seed $\sigma \in \{0,1\}^r$, the function $\tilde{h}(\mathbf{x}) = G(\sigma)_{C_0(\mathbf{x})}$ can be computed by a depth-14 binary threshold network with widths $\underline{d}$ such that the maximum width $\underline{d}_{\max}$ is at most*

$$\log^{3/4}(1/\epsilon) \cdot \mathrm{polylog}\, R + O(d_0 \cdot \log R),$$

*and the total number of weights $w\,(\underline{d})$ is at most*

$$\log(1/\epsilon) + \log^{3/4}(1/\epsilon) \cdot \mathrm{polylog}\, R + O(d_0^2 \cdot \log R + d_0 \cdot \log^2 R).$$

*Proof.* First we will describe the construction of $G$; then we will verify its seed length and computational complexity; and finally we will verify its correctness.

**Construction.** Let $T$ be the smallest power of two such that $T \geq \log^{3/4}(1/\epsilon)$. Let

$$\epsilon_0 = \frac{\epsilon^{1/T}}{2R},$$

and note that $\log(1/\epsilon_0) = \Theta(\log^{1/4}(1/\epsilon) + \log R)$. Let $G_0: \{0,1\}^{r_0} \to \{0,1\}^R$ be the generator of Lemma D.17 with error parameter $\epsilon_0$, i.e., for every $V \in \mathcal{V}$, if $\mathbb{P}_{\mathbf{y} \sim \mathrm{Ber}(\alpha)^R}(V(\mathbf{y}) = 1) \geq 2\epsilon_0$, then $\mathbb{P}_{\mathbf{u} \in \{0,1\}^{r_0}}(V(G_0(\mathbf{u})) = 1) \geq \epsilon_0$. Let $a$ be the smallest positive integer such that $2^a > R/\epsilon_0$. Let $\mathcal{H}$ be the family of $\mathbb{F}_2$-affine hash functions $\mathrm{hash}: \{0,1\}^a \to \{0,1\}^{r_0}$ from Lemma D.6.

A seed for our generator $G$ consists of a function $\mathrm{hash} \in \mathcal{H}$, inputs $\mathbf{w}^0, \ldots, \mathbf{w}^{T-1} \in \{0,1\}^a$, and nonnegative integers $0 = \ell_0 \leq \ell_1 \leq \cdots \leq \ell_T = R$. Given this data $\sigma = (\mathrm{hash}, \mathbf{w}^0, \ldots, \mathbf{w}^{T-1}, \ell_0, \ldots, \ell_T)$, the output of the generator is given by

$$G(\sigma) = G_0(\mathrm{hash}(\mathbf{w}^0))_{0 \cdots \ell_1 - 1} \;\&\; G_0(\mathrm{hash}(\mathbf{w}^1))_{\ell_1 \cdots \ell_2 - 1} \;\&\; \cdots \;\&\; G_0(\mathrm{hash}(\mathbf{w}^{T-1}))_{\ell_{T-1} \cdots \ell_T - 1}.$$

In the equation above, $\mathbf{y}_{a \cdots b}$ denotes the substring of $\mathbf{y}$ consisting of the bits at positions $a, a+1, a+2, \ldots, b$, and $\&$ denotes concatenation.

**Seed length and computational complexity.** Since $|\mathcal{H}| \leq 2^{O(a+r_0)}$, the description length of $\mathrm{hash}$ is $O(a + r_0)$. The description length of $\mathbf{w}^0, \ldots, \mathbf{w}^{T-1}$ is $aT$, and the description length of $\ell^0, \ldots, \ell^T$ is $O(T \log R)$. By our choices of $a$ and $\epsilon_0$, we have

$$a \leq \log(1/\epsilon_0) + O(\log R) = \frac{\log(1/\epsilon)}{T} + O(\log R).$$

Furthermore, by Lemma D.17, we have

$$r_0 = O(\log(1/\epsilon_0) \cdot \log^2 R).$$

Therefore, the overall seed length of our generator is

$$aT + O(r_0 + T \log R + a) \leq \log(1/\epsilon) + \log^{3/4}(1/\epsilon) \cdot \mathrm{polylog}\, R.$$

To analyze the computational complexity, fix an arbitrary seed

$$\sigma = (\mathrm{hash}, \mathbf{w}^0, \ldots, \mathbf{w}^{T-1}, \ell_0, \ldots, \ell_T).$$

The numbers $\ell_0, \ldots, \ell_T$ partition the interval $[0, R)$ into subintervals, namely $[0, R) = [\ell_0, \ell_1) \cup [\ell_1, \ell_2) \cup \cdots \cup [\ell_{T-1}, \ell_T)$. Define $g \colon \{0, 1, \ldots, R-1\} \to \{0, 1\}^a$ by the rule

$$g(\mathbf{z}) = \mathbf{w}^j \text{ where } j \text{ is such that } \mathbf{z} \in [\ell_j, \ell_{j+1}).$$

Then $g$ is constant on each subinterval $[\ell_j, \ell_{j+1})$, so we may apply Lemma D.21 to obtain a depth-6 binary threshold network $C_1 \colon \{0, 1\}^{d_0} \to \{0, 1\}^{a + \log R}$ computing the function $C(\mathbf{x}) = (g(C_0(\mathbf{x})), C_0(\mathbf{x}))$. In this network, every layer has width at most

$$O(T \cdot \log^3 R + a + d_0 \cdot \log R) = \log^{3/4}(1/\epsilon) \cdot \mathrm{polylog}\, R + O(d_0 \cdot \log R),$$

and the total number of weights is at most

$$aT + O(T \cdot \log^4 R + d_0^2 \cdot \log R + d_0 \cdot \log^2 R + a \cdot \log R)$$

$$\leq \log(1/\epsilon) + \log^{3/4}(1/\epsilon) \cdot \mathrm{polylog}\, R + O(d_0^2 \cdot \log R + d_0 \cdot \log^2 R).$$

Let $\mathbf{z} = C_0(\mathbf{x})$, and let $\mathbf{w} = g(\mathbf{z})$. Our remaining goal is to compute $G(\sigma)_{\mathbf{z}}$, which is equal to $G_0(\mathrm{hash}(\mathbf{w}))_{\mathbf{z}}$. To do so, we use the network guaranteed to exist by Lemma D.17. This network, which we call $C_2$, has depth 8. Every layer in this network has width at most

$$a \cdot \log(1/\epsilon_0) \cdot \mathrm{polylog}\, R = \sqrt{\log(1/\epsilon)} \cdot \mathrm{polylog}\, R.$$

The total number of weights in this network is at most

$$(\log(1/\epsilon_0) \cdot a^2 + \log^2(1/\epsilon_0) \cdot a) \cdot \mathrm{polylog}\, R = \log^{3/4}(1/\epsilon) \cdot \mathrm{polylog}\, R.$$

Composing $C_2$ with $C_1$ completes the analysis of the computational complexity of our HSG.

**Correctness.** Finally, let us prove the correctness of our HSG. For convenience, for any $n \in \mathbb{N}$ and any function $V \colon \{0, 1\}^n \to \{0, 1\}$, we write $\mathbb{E}(V)$ to denote the quantity $\mathbb{P}_{\mathbf{y} \sim \mathrm{Ber}(\alpha)^n}(V(\mathbf{y}) = 1)$.

Fix any $V \in \mathcal{V}$ such that $\mathbb{E}(V) > \epsilon$. Since $V$ is a conjunction of literals, we can write $V$ in the form

$$V(\mathbf{y}) = V_0(\mathbf{y}_0) \cdot V_1(\mathbf{y}_1) \cdots V_{R-1}(\mathbf{y}_{R-1})$$

for some functions $V_0, V_1, \ldots, V_{R-1} \colon \{0, 1\} \to \{0, 1\}$. For each $0 \leq a \leq b \leq R - 1$, define

$$V_{a \cdots b} = V_a \cdot V_{a+1} \cdots V_b.$$

We inductively define numbers $0 = \ell_0 \leq \ell_1 \leq \cdots \leq \ell_T$ as follows. Assume that we have already defined $\ell_0, \ldots, \ell_i$. Let $\ell_{i+1}$ be the smallest integer in $\{\ell_i + 1, \ldots, R - 1\}$ such that

$$\mathbb{E}(V_{\ell_i \cdots \ell_{i+1}-1}) \leq \epsilon^{1/T},$$

or let $\ell_{i+1} = R$ if no such $\ell_{i+1}$ exists. Define $V^{(i)} = V_{\ell_i \cdots \ell_{i+1}-1}$. Observe that $\ell_T = R$, because otherwise we would have

$$\epsilon < \mathbb{E}(V) \leq \prod_{i=0}^{T-1} \mathbb{E}(V_i) \leq (\epsilon^{1/T})^T = \epsilon,$$

a contradiction. Furthermore, $\mathbb{E}(V_i) > \epsilon^{1/T}/R = 2\epsilon_0$, because each literal in $V$ is satisfied with probability at least $\min\{\alpha, 1 - \alpha\} \geq 1/R$. Therefore, if we define

$$S_i = \{\mathbf{u} \in \{0, 1\}^{r_0} : V_i(G_0(\mathbf{u})_{\ell_i \cdots \ell_{i+1}-1}) = 1\}$$

and $\rho_i = |S_i|/2^{r_0}$, then the correctness of $G_0$ ensures that $\rho_i \geq \epsilon_0$.

Next, we will show that there exist $\mathrm{hash}, \mathbf{w}^0, \ldots, \mathbf{w}^{T-1}$ such that for every $i$, we have $\mathrm{hash}(\mathbf{w}^i) \in S_i$. To prove it, pick $\mathrm{hash} \in \mathcal{H}$ at random. For each $i \in \{0, 1, \ldots, T-1\}$ and each $\mathbf{w} \in \{0, 1\}^a$, let $X_{i,\mathbf{w}}$ be the indicator random variable for the "good" event $\mathrm{hash}(\mathbf{w}) \in S_i$. Define $X_i = \sum_{\mathbf{w} \in \{0,1\}^a} X_{i,\mathbf{w}}$. Then for every $i$, by pairwise independence, we have

$$\mathbb{E}(X_i) = 2^a \cdot \rho_i$$

$$\text{and } \mathrm{Var}(X_i) = 2^a \cdot \rho_i \cdot (1 - \rho_i) \leq 2^a \cdot \rho_i.$$

Therefore, by Chebyshev's inequality,

$$\mathbb{P}(X_i = 0) \leq \frac{1}{2^a \cdot \rho_i} \leq \frac{1}{2^a \cdot \epsilon_0} < \frac{1}{R}.$$

Consequently, by the union bound over all $i$, there is a nonzero chance that $X_0 = X_1 = \cdots = X_{T-1} = 0$, in which case there exist $\mathbf{w}^0, \ldots, \mathbf{w}^{T-1}$ such that $\mathrm{hash}(\mathbf{w}^i) \in S_i$ for every $i$.

At this point, we have constructed our seed $\sigma = (\mathrm{hash}, \mathbf{w}^0, \ldots, \mathbf{w}^{T-1}, \ell_0, \ldots, \ell_T)$. By the construction of $G$, we have $V(G(\sigma)) = 1$. $\qquad \square$

Theorem 3.1 readily follows from Theorem D.22, as we now explain.

**Recall Theorem 3.1.** Let $f\colon \{0,1\}^{d_0} \to \{0,1,\star\}$ be any function.[9] Let $N = |f^{-1}(\{0,1\})|$ and $N_1 = |f^{-1}(1)|$. There exists a depth-14 binary threshold network $\tilde{h}\colon \{0,1\}^{d_0} \to \{0,1\}$, with widths $\underline{\tilde{d}}$, satisfying the following.

1. $\tilde{h}$ is consistent with $f$, *i.e.,* for every $\mathbf{x} \in \{0,1\}^{d_0}$, if $f(\mathbf{x}) \in \{0,1\}$, then $\tilde{h}(\mathbf{x}) = f(\mathbf{x})$.

2. The total number of weights in $\tilde{h}$ is at most $(1 + o(1)) \cdot \log \binom{N}{N_1} + \mathrm{poly}(d_0)$. More precisely,

$$w\left(\underline{\tilde{d}}\right) = \log \binom{N}{N_1} + \left(\log \binom{N}{N_1}\right)^{3/4} \cdot \mathrm{polylog}\, N + O(d_0^2 \cdot \log N).$$

3. Every layer of $\tilde{h}$ has width at most $(\log \binom{N}{N_1})^{3/4} \cdot \mathrm{poly}(d_0)$. More precisely,

$$\underline{\tilde{d}}_{\max} = \left(\log \binom{N}{N_1}\right)^{3/4} \cdot \mathrm{polylog}\, N + O(d_0 \cdot \log N).$$

*Proof.* Let $R = 2^{2\lceil \log N \rceil}$. Let $C_0\colon \{0,1\}^{d_0} \to \{0,1\}^{\log R}$ be an $\mathbb{F}_2$-affine function that is injective on $\mathcal{X}$; such a function is guaranteed to exist by Lemma D.4. Define $V\colon \{0,1\}^R \to \{0,1\}$ by the rule

$$V(\mathbf{y}) = 1 \iff \forall \mathbf{x} \in \mathcal{X},\ \mathbf{y}_{C_0(\mathbf{x})} = f(x).$$

This function $V$ is a conjunction of $N_1$ variables and $N - N_1$ negated variables.

If $N_1 \in \{0, N\}$, then the theorem is trivial, because we can take $\tilde{h}$ to be a constant function. Assume, therefore, that $0 < N_1 < N$. Let $\alpha = N_1/N$, and note that $1/R \leq \alpha \leq 1 - 1/R$. Let $\epsilon = \frac{1}{2}\alpha^{N_1} \cdot (1-\alpha)^{N-N_1} = 2^{-H(\alpha) \cdot N - 1}$, and note that

$$\mathbb{P}_{\mathbf{y} \sim \mathrm{Ber}(\alpha)^R}(V(\mathbf{y}) = 1) = 2\epsilon.$$

Let $G\colon \{0,1\}^r \to \{0,1\}^R$ be the HSG from Theorem D.22. There exists a seed $\sigma \in \{0,1\}^r$ such that $V(G(\sigma)) = 1$. Our network $\tilde{h}$ computes the function $\tilde{h}(\mathbf{x}) = G(\sigma)_{C_0(\mathbf{x})}$. Since $V(G(\sigma)) = 1$, we must have $\tilde{h}(\mathbf{x}) = f(\mathbf{x})$ for every $x \in \mathcal{X}$.

To bound the computational complexity, observe that $\log(1/\epsilon) = H(\alpha) \cdot N + 1 \leq \log \binom{N}{N_1} + O(\log N)$. Therefore, every layer of $\tilde{h}$ has width at most

$$\left(\log \binom{N}{N_1}\right)^{3/4} \cdot \mathrm{polylog}\, N + O(d_0 \cdot \log N),$$

and the total number of weights in $\tilde{h}$ is at most

$$\log \binom{N}{N_1} + \left(\log \binom{N}{N_1}\right)^{3/4} \cdot \mathrm{polylog}\, N + O(d_0^2 \cdot \log N + d_0 \cdot \log^2 N).$$

Finally, we have $N \leq 2^{d_0}$, so the last term above can be simplified to $O(d_0^2 \cdot \log N)$. □

*Remark D.23.* In Theorem 3.1, the weights bound has an $O(d_0^2 \cdot \log N)$ term. This term is close to optimal; see Appendix E for further details.

---

[9]When $f(\mathbf{x}) = \star$, the interpretation is that $f$ is "undefined" on $\mathbf{x}$, *i.e.,* $f$ is a "partial" function.

## D.6 XOR networks

In what follows, we denote the activation function $\sigma(x) = \mathbb{I}\{x > 0\}$.

**Lemma D.24** (XOR NN). *The* XOR *function can be implemented with a single-hidden-layer fully connected binary threshold network with input dimension 2 and $c_{\text{XOR}}$ parameters by*

$$h_{\text{XOR}}\begin{pmatrix} x_1 \\ x_2 \end{pmatrix} = \sigma\left(1 \odot (\begin{array}{cc} 1 & 1 \end{array}) \cdot \sigma\left(\left(\begin{array}{c} 1 \\ -1 \end{array}\right) \odot \left(\begin{array}{cc} 1 & 1 \\ 1 & 1 \end{array}\right)\left(\begin{array}{c} x_1 \\ x_2 \end{array}\right) + \left(\begin{array}{c} 0 \\ 2 \end{array}\right)\right) - 1\right).$$

*Proof.* We can simplify $h_{\text{XOR}}$ as

$$
\begin{aligned}
h_{\text{XOR}}\begin{pmatrix} x_1 \\ x_2 \end{pmatrix} &= \sigma\left((\begin{array}{cc} 1 & 1 \end{array}) \cdot \sigma\left(\left(\begin{array}{c} 1 \\ -1 \end{array}\right) \odot \left(\begin{array}{cc} 1 & 1 \\ 1 & 1 \end{array}\right)\left(\begin{array}{c} x_1 \\ x_2 \end{array}\right) + \left(\begin{array}{c} 0 \\ 2 \end{array}\right)\right) - 1\right) \\
&= \sigma\left((\begin{array}{cc} 1 & 1 \end{array}) \cdot \sigma\left(\left(\begin{array}{c} x_1 + x_2 \\ -x_1 - x_2 + 2 \end{array}\right)\right) - 1\right) \\
&= \sigma\left(\sigma(x_1 + x_2) + \sigma(2 - x_1 - x_2) - 1\right) \\
&= \mathbb{I}\{\mathbb{I}\{x_1 + x_2 > 0\} + \mathbb{I}\{x_1 + x_2 < 2\} > 1\} \\
&= \mathbb{I}\left\{\mathbb{I}\left\{\begin{pmatrix} x_1 \\ x_2 \end{pmatrix} \neq \begin{pmatrix} 0 \\ 0 \end{pmatrix}\right\} + \mathbb{I}\left\{\begin{pmatrix} x_1 \\ x_2 \end{pmatrix} \neq \begin{pmatrix} 1 \\ 1 \end{pmatrix}\right\} > 1\right\} \\
&= \mathbb{I}\left\{\begin{pmatrix} x_1 \\ x_2 \end{pmatrix} \neq \begin{pmatrix} 0 \\ 0 \end{pmatrix} \text{ and } \begin{pmatrix} x_1 \\ x_2 \end{pmatrix} \neq \begin{pmatrix} 1 \\ 1 \end{pmatrix}\right\} \\
&= \text{XOR}(x_1, x_2) .
\end{aligned}
$$

$\square$

*Remark* D.25. Notice that the function $\text{Id} : \{0, 1\} \to \{0, 1\}$ defined as $\text{Id}(0) = 0$ and $\text{Id}(1) = 1$ can be implemented using any depth $L$ network with a single input dimension and $c_{\text{Id}} \cdot L$ parameters.

Following this remark, for simplicity we shall assume that $h_1$ and $h_2$ in the following Lemma are of the same depth, as they can be elongated with $O(L)$ additional parameters, which are negligible in the subsequent analysis.

**Recall Lemma 3.3.** Let $h_1, h_2$ be two binary networks with depths $L_1 \leq L_2$ and widths $\underline{d}^{(1)}, \underline{d}^{(2)}$, respectively. Then, there exists a network $h$ with depth $L_{\text{XOR}} \triangleq L_2 + 2$ and widths

$$\underline{d}_{\text{XOR}} \triangleq \left( d_1^{(1)} + d_1^{(2)}, \ldots, d_{L_1}^{(1)} + d_{L_1}^{(2)}, d_{L_1+1}^{(2)} + 1, \ldots, d_{L_2}^{(2)} + 1, 2, 1 \right),$$

such that for all inputs $\mathbf{x} \in \{0,1\}^{d_0}$, $h(\mathbf{x}) = h_1(\mathbf{x}) \oplus h_2(\mathbf{x})$.

The lemma above is given immediately by the lemma we state and prove next.

**Lemma D.26** (XOR of Two NNs). *Let* $h_1, h_2 : \mathcal{X} \to \{0,1\}$ *be quantized fully connected binary threshold networks with depths $L'$ and widths $\underline{d}^{(1)}, \underline{d}^{(2)}$, respectively. Let $L \geq 2 + L'$ and $\underline{d} \geq \underline{d}_{\text{XOR}}$. Let $\Theta^{BTN}(\underline{d}; h_1, h_2)$ be the subset of $\Theta^{BTN}(\underline{d})$ such that for all $\boldsymbol{\theta} \in \Theta^{BTN}(\underline{d}; h_1, h_2)$, $\boldsymbol{\theta}$ has the following form:*

- *For $l = 1$:*

$$\mathbf{W}_1 = \begin{pmatrix} \mathbf{W}_1^{(1)} \\ \mathbf{W}_1^{(2)} \\ \tilde{\mathbf{W}}_1 \end{pmatrix}, \ \mathbf{b}_1 = \begin{pmatrix} \mathbf{b}_1^{(1)} \\ \mathbf{b}_1^{(2)} \\ \tilde{\mathbf{b}}_1 \end{pmatrix}, \ \gamma_1 = \begin{pmatrix} \mathbf{1}_{d_l^{(1)}} \\ \mathbf{1}_{d_l^{(2)}} \\ \mathbf{0}_{d_l - d_l^{(1)} - d_l^{(2)}} \end{pmatrix}$$

  *with arbitrary $\tilde{\mathbf{W}}_1, \tilde{\mathbf{b}}_1$.*

- *For $l = 2, \ldots, L'$:*

$$\mathbf{W}_l = \begin{pmatrix} \mathbf{W}_l^{(1)} & \mathbf{0}_{d_l^{(1)} \times d_{l-1}^{(2)}} & \tilde{\mathbf{W}}_l^1 \\ \mathbf{0}_{d_l^{(2)} \times d_{l-1}^{(1)}} & \mathbf{W}_l^{(2)} & \tilde{\mathbf{W}}_l^2 \\ \tilde{\mathbf{W}}_l^3 & \tilde{\mathbf{W}}_l^4 & \tilde{\mathbf{W}}_l^5 \end{pmatrix} \in \{0,1\}^{d_l \times d_{l-1}},$$

$$\mathbf{b}_l = \begin{pmatrix} \mathbf{b}_l^{(1)} \\ \mathbf{b}_l^{(2)} \\ \tilde{\mathbf{b}}_l \end{pmatrix} \in \{-d_{l-1}, \ldots, -1, 0, 1, \ldots, d_{l-1} - 1\}^{d_l},$$

$$\gamma_l = \begin{pmatrix} \mathbf{1}_{d_l^{(1)}} \\ \mathbf{1}_{d_l^{(2)}} \\ \mathbf{0}_{d_l - d_l^{(1)} - d_l^{(2)}} \end{pmatrix} \in \{0,1\}^{d_l},$$

  *with arbitrary $\tilde{\mathbf{W}}_l^1, \tilde{\mathbf{W}}_l^2, \tilde{\mathbf{W}}_l^3, \tilde{\mathbf{W}}_l^4, \tilde{\mathbf{W}}_l^5, \tilde{\mathbf{b}}_l$.*

- *For $l = L' + k$, $k = 1, 2$:*

$$\mathbf{W}_l = \begin{pmatrix} \mathbf{W}_k^{\text{XOR}} & \tilde{\mathbf{W}}_l^1 \\ \tilde{\mathbf{W}}_l^2 & \tilde{\mathbf{W}}_l^3 \end{pmatrix} \in \{0,1\}^{d_l \times d_{l-1}},$$

$$\mathbf{b}_l = \begin{pmatrix} \mathbf{b}_k^{\text{XOR}} \\ \tilde{\mathbf{b}}_l \end{pmatrix} \in \{-d_{l-1}, \ldots, -1, 0, 1, \ldots, d_{l-1} - 1\}^{d_l}, \gamma_l = \begin{pmatrix} \gamma_k^{\text{XOR}} \\ \mathbf{0} \end{pmatrix} \in \{0, \pm 1\}^{d_l}.$$

- *And for $l > L' + 2$:*

$$\mathbf{W}_l = \begin{pmatrix} \mathbf{W}^{Id} & \tilde{\mathbf{W}}_l^1 \\ \tilde{\mathbf{W}}_l^2 & \tilde{\mathbf{W}}_l^3 \end{pmatrix} \in \{0,1\}^{d_l \times d_{l-1}},$$

$$\mathbf{b}_l = \begin{pmatrix} \mathbf{b}^{Id} \\ \tilde{\mathbf{b}}_l \end{pmatrix} \in \{-d_{l-1}, \ldots, -1, 0, 1, \ldots, d_{l-1} - 1\}^{d_l}, \gamma_l = \begin{pmatrix} \gamma^{Id} \\ \mathbf{0} \end{pmatrix} \in \{0, \pm 1\}^{d_l}.$$

*Then for all $\boldsymbol{\theta} \in \Theta^{BTN}(\underline{d}; h_1, h_2)$ $h_{\boldsymbol{\theta}} = h_1 \oplus h_2$.*

An illustration of this construction is given in Figure 3.

*Proof.* We prove the claim by induction. For $l = 1$ we have $d_0 = d_0^{(1)} = d_0^{(2)}$ and

$$h_\theta^{(1)}(\mathbf{x}) = \gamma_1 \odot \sigma\left(\mathbf{W}_1 h_\theta^{(0)}(\mathbf{x}) + \mathbf{b}_1\right)$$

$$= \begin{pmatrix} \mathbf{1}_{d_1^\star} \\ \mathbf{1}_{d_1^f} \\ \mathbf{0}_{d_1 - d_1^\star - d_1^f} \end{pmatrix} \odot \sigma\left( \begin{pmatrix} \mathbf{W}_1^{(1)} \\ \mathbf{W}_1^{(2)} \\ \tilde{\mathbf{W}}_1^1 \end{pmatrix} \mathbf{x} + \begin{pmatrix} \mathbf{b}_1^{(1)} \\ \mathbf{b}_1^{(2)} \\ \tilde{\mathbf{b}}_1 \end{pmatrix} \right)$$

$$= \begin{pmatrix} \sigma\left(\mathbf{W}_1^{(1)}\mathbf{x} + \mathbf{b}_1^{(1)}\right) \\ \sigma\left(\mathbf{W}_1^{(2)}\mathbf{x} + \mathbf{b}_1^{(2)}\right) \\ \mathbf{0}_{d_1 - d_1^{(1)} - d_1^{(2)}} \end{pmatrix} = \begin{pmatrix} h_1^{(1)}(\mathbf{x}) \\ h_2^{(1)}(\mathbf{x}) \\ \mathbf{0}_{d_1 - d_1^\star - d_1^f} \end{pmatrix}.$$

Assume that for some $l \leq L'$ we have

$$h_\theta^{(l-1)}(\mathbf{x}) = \begin{pmatrix} h_1^{(l-1)}(\mathbf{x}) \\ h_2^{(l-1)}(\mathbf{x}) \\ \mathbf{0}_{d_l - d_l^\star - d_l^f} \end{pmatrix}.$$

Then,

$$h_\theta^{(l)}(\mathbf{x}) = \gamma_l \odot \sigma\left(\mathbf{W}_l h_\theta^{(l-1)}(\mathbf{x}) + \mathbf{b}_l\right)$$

$$= \begin{pmatrix} \mathbf{1}_{d_l^{(1)}} \\ \mathbf{1}_{d_l^{(2)}} \\ \mathbf{0}_{d_l - d_l^{(1)} - d_l^{(2)}} \end{pmatrix} \odot \sigma\left( \begin{pmatrix} \mathbf{W}_l^{(1)} & \mathbf{0}_{d_l^{(1)} \times d_{l-1}^{(2)}} & \tilde{\mathbf{W}}_l^1 \\ \mathbf{0}_{d_l^{(2)} \times d_{l-1}^{(1)}} & \mathbf{W}_l^{(2)} & \tilde{\mathbf{W}}_l^2 \\ \tilde{\mathbf{W}}_l^3 & \tilde{\mathbf{W}}_l^4 & \tilde{\mathbf{W}}_l^5 \end{pmatrix} \begin{pmatrix} h_1^{(l-1)}(\mathbf{x}) \\ h_2^{(l-1)}(\mathbf{x}) \\ \mathbf{0}_{d_l - d_l^{(1)} - d_l^{(2)}} \end{pmatrix} + \begin{pmatrix} \mathbf{b}_l^{(1)} \\ \mathbf{b}_l^{(2)} \\ \tilde{\mathbf{b}}_l \end{pmatrix} \right)$$

$$= \begin{pmatrix} \sigma\left(\mathbf{W}_l^{(1)} h_1^{(l-1)}(\mathbf{x}) + \mathbf{b}_l^{(1)}\right) \\ \sigma\left(\mathbf{W}_l^{(2)} h_2^{(l-1)}(\mathbf{x}) + \mathbf{b}_l^{(2)}\right) \\ \mathbf{0}_{d_l - d_l^{(1)} - d_l^{(2)}} \end{pmatrix} = \begin{pmatrix} h_1^{(l)}(\mathbf{x}) \\ h_2^{(l)}(\mathbf{x}) \\ \mathbf{0}_{d_l - d_l^{(1)} - d_l^{(2)}} \end{pmatrix}.$$

It is left to show that the claim holds for $l > L'$. By the previous steps, $h_\theta^{(L')}(\mathbf{x}) = \begin{pmatrix} h_1(\mathbf{x}) \\ h_2(\mathbf{x}) \\ \mathbf{0}_{d_{L'} - 2} \end{pmatrix}$.

Under the assumptions on $\mathbf{W}_{L'+k}, \mathbf{b}_{L'+k}$ and $\gamma_{L'+k}$, $k = 1, 2$ it holds that

$$h_\theta^{(L'+2)}(\mathbf{x}) = \begin{pmatrix} h_1(\mathbf{x}) \oplus h_2(\mathbf{x}) \\ \mathbf{0}_{d_{L'}-1} \end{pmatrix}.$$

Under the assumptions on layers $l > L' + 2$,

$$h_\theta^{(l)}(\mathbf{x}) = \begin{pmatrix} h_1(\mathbf{x}) \oplus h_2(\mathbf{x}) \\ \mathbf{0}_{d_l - 1} \end{pmatrix}.$$

In particular, assuming that $d_L = 1$, $h_\theta(\mathbf{x}) = h_1(\mathbf{x}) \oplus h_2(\mathbf{x})$. $\qquad\square$

**Corollary D.27.** *Let $h_1, h_2$ be networks with depths $L_1, L_2$ and widths $\underline{d}^{(1)}, \underline{d}^{(2)}$. Then $h_1 \oplus h_2$ can be implemented with a network $h$ of depth $L = \max\{L_1, L_2\} + 2$ and widths $\underline{d}$ such that*

$$w\left(\underline{d}\right) \leq w\left(\underline{d}^{(1)}\right) + w\left(\underline{d}^{(2)}\right) + 2\underline{d}_{\max}^{(2)} \cdot n\left(\underline{d}^{(1)}\right) + O(1)$$

*and*

$$\underline{d}_{\max} \leq \underline{d}_{\max}^{(1)} + \underline{d}_{\max}^{(2)}.$$

*Proof.* Following D.25 we assume shall assume that $L_1 = L_2 = L$. We know from D.26 that there exists a network $h$ with dimensions $\underline{d} = \left(\underline{d}^{(1)} + \underline{d}^{(2)}, 2, 1\right)$ such that $h = h_1 \oplus h_2$. Therefore

$$w\left(\underline{d}\right) = \left(d_1^{(1)} + d_1^{(2)}\right) d_0 + \sum_{l=2}^{L} \left(d_l^{(1)} + d_l^{(2)}\right)\left(d_{l-1}^{(1)} + d_{l-1}^{(2)}\right) + O(1)$$

$$= d_1^{(1)} d_0 + \sum_{l=2}^{L} d_l^{(1)} d_{l-1}^{(1)} + d_1^{(2)} d_0 + \sum_{l=2}^{L} d_l^{(2)} d_{l-1}^{(2)} + \sum_{l=2}^{L} \left[d_l^{(1)} d_{l-1}^{(2)} + d_l^{(2)} d_{l-1}^{(1)}\right] + O(1)$$

$$= w\left(\underline{d}^{(1)}\right) + w\left(\underline{d}^{(2)}\right) + \sum_{l=2}^{L} \left[d_l^{(1)} d_{l-1}^{(2)} + d_l^{(2)} d_{l-1}^{(1)}\right] + O(1)$$

$$\leq w\left(\underline{d}^{(1)}\right) + w\left(\underline{d}^{(2)}\right) + \sum_{l=2}^{L} \left[d_l^{(1)} \underline{d}_{\max}^{(2)} + \underline{d}_{\max}^{(2)} d_{l-1}^{(1)}\right] + O(1)$$

$$= w\left(\underline{d}^{(1)}\right) + w\left(\underline{d}^{(2)}\right) + \underline{d}_{\max}^{(2)} \sum_{l=2}^{L} \left[d_l^{(1)} + d_{l-1}^{(1)}\right] + O(1)$$

$$\leq w\left(\underline{d}^{(1)}\right) + w\left(\underline{d}^{(2)}\right) + 2\underline{d}_{\max}^{(2)} \cdot n\left(\underline{d}^{(1)}\right) + O(1).$$

In addition, $\underline{d}_{\max} \leq \underline{d}_{\max}^{(1)} + \underline{d}_{\max}^{(2)}$ and $n\left(\underline{d}\right) = n\left(\underline{d}^{(1)}\right) + n\left(\underline{d}^{(2)}\right)$. $\qquad \square$

**Recall Corollary 3.4.** For any teacher $h^\star$ of depth $L^\star$ and dimensions $\underline{d}^\star$ and any consistent training set $S$ generated from it, there exists an interpolating network $h$ (*i.e.,* $\mathcal{L}_S(h) = 0$) of depth $L = \max\{L^\star, 14\} + 2$ and dimensions $\underline{d}$, such that the number of edges is

$$w\left(\underline{d}\right) \leq w\left(\underline{d}^\star\right) + N \cdot H\left(\mathcal{L}_S\left(h^\star\right)\right) + 2n\left(\underline{d}^\star\right) N^{3/4} H\left(\mathcal{L}_S\left(h^\star\right)\right)^{3/4} \text{polylog} N$$
$$+ O\left(d_0\left(d_0 + n\left(\underline{d}^\star\right)\right) \cdot \log N\right)$$

and the dimensions are

$$\underline{d}_{\max} \leq \underline{d}_{\max}^\star + N^{3/4} \cdot H\left(\mathcal{L}_S\left(h^\star\right)\right)^{3/4} \cdot \text{polylog}\left(N\right) + O\left(d_0 \cdot \log\left(N\right)\right).$$

*Proof.* We use Corollary D.27 with $h_1 = h^\star$ and $h_2 = \tilde{h}_S$, the noise memorizing network from Theorem 3.1, to get

$$w\left(\underline{d}\right) \leq w\left(\underline{d}^\star\right) + w\left(\tilde{\underline{d}}_S\right) + 2\tilde{\underline{d}}_{S,\max} \cdot n\left(\underline{d}^\star\right) + O(1)$$

$$\leq w\left(\underline{d}^\star\right) + \log\binom{N}{N_1} + \left(\log\binom{N}{N_1}\right)^{3/4} \cdot \text{polylog} N + O(d_0^2 \cdot \log N)$$

$$+ 2n\left(\underline{d}^\star\right)\left(\left(\log\binom{N}{N_1}\right)^{3/4} \cdot \text{polylog} N + O(d_0 \cdot \log N)\right) + O(1).$$

Using Stirling's approximation

$$\log\binom{N}{N_1} = N \cdot H\left(\frac{N_1}{N}\right) + O\left(\log\left(N\right)\right) = N \cdot H\left(\mathcal{L}_S\left(h^\star\right)\right) + O\left(\log\left(N\right)\right).$$

Therefore

$$w\left(\underline{d}\right) \leq w\left(\underline{d}^{\star}\right) + N \cdot H\left(\mathcal{L}_S\left(h^{\star}\right)\right) + O\left(\log\left(N\right)\right) + N^{3/4} \cdot H\left(\mathcal{L}_S\left(h^{\star}\right)\right)^{3/4} \cdot \mathrm{polylog}\, N$$
$$+ O\left(d_0^2 \cdot \log N\right) + 2n\left(\underline{d}^{\star}\right)\left(N^{3/4} \cdot \mathrm{polylog}\, N + O\left(d_0 \cdot \log N\right)\right)$$
$$= w\left(\underline{d}^{\star}\right) + N \cdot H\left(\mathcal{L}_S\left(h^{\star}\right)\right) + 2n\left(\underline{d}^{\star}\right) N^{3/4} H\left(\mathcal{L}_S\left(h^{\star}\right)\right)^{3/4} \mathrm{polylog} N$$
$$+ O\left(d_0\left(d_0 + n\left(\underline{d}^{\star}\right)\right) \cdot \log N\right) .$$

The bound of $\underline{d}_{\max}$ is derived similarly. $\qquad\square$

# E  The label-flip-memorization network's dependence on the dimension

In Theorem 3.1, the wire bound has an $O(d_0^2 \cdot \log N)$ term. (Recall that $d_0$ is the input dimension and $N$ is the domain size.) In this section, we discuss (a) approaches for improving this term and (b) a lower bound showing that it cannot be significantly improved.

## E.1  Improving the $O(d_0^2 \cdot \log N)$ Term

The $O(d_0^2 \cdot \log N)$ term in Theorem 3.1 can be improved by using the following fact.

**Lemma E.1** (Using a sign matrix for preprocessing). *Let $d_0 \in \mathbb{N}$, let $\hat{\mathcal{X}} \subseteq \{0,1\}^{d_0}$, and let $N = |\hat{\mathcal{X}}|$. There exists $d_1 = O(\sqrt{d_0} \cdot \log N)$ and there exists a matrix $\mathbf{W} \in \{\pm 1\}^{d_1 \times d_0}$ such that the function $C_0 \colon \{0,1\}^{d_0} \to \{0,1\}^{d_1}$ defined by $C_0(\mathbf{x}) = \mathbb{I}\{\mathbf{W}\mathbf{x} > 0\}$ is injective on $\hat{\mathcal{X}}$.*

*Proof.* Pick $\mathbf{W} \in \{\pm 1\}^{d_1 \times d_0}$ uniformly at random. We will show that there is a nonzero chance that $C_0$ is injective on $\hat{\mathcal{X}}$.

Let $\mathbf{x}, \mathbf{x}'$ be any two distinct points in $\hat{\mathcal{X}}$. Consider a single row $\mathbf{W}_i$ of $\mathbf{W}$. Let $E$ be the good event that

$$\mathbf{W}_i \cdot (\mathbf{x} \odot \mathbf{x}') \in \{0,1\}.$$

Then $\Pr[E] \geq \Omega(1/\sqrt{d_0})$, because we are taking a simple one-dimensional random walk of length at most $d_0$. Conditioned on $E$, there is an $\Omega(1)$ chance that $\mathbb{I}\{\mathbf{W}_i \cdot \mathbf{x} > 0\} \neq \mathbb{I}\{\mathbf{W}_i \cdot \mathbf{x}' > 0\}$, because we are taking two independent one-dimensional random walks starting from either $0$ or $1$, at least one of which has nonzero length, and asking whether they land on the same side of $1/2$. Therefore, unconditionally, $\Pr[\mathbb{I}\{\mathbf{W}_i \cdot \mathbf{x} > 0\} \neq \mathbb{I}\{\mathbf{W}_i \cdot \mathbf{x}' > 0\}] \geq \Omega(1/\sqrt{d_0})$. Consequently, by independence,

$$\Pr[C_0(\mathbf{x}) = C_0(\mathbf{x}')] \leq (1 - \Omega(1/\sqrt{d_0}))^{d_1} < 1/N^2,$$

provided we choose a suitable value $d_1 = O(\sqrt{d_0} \cdot \log N)$. By the union bound over all pairs $\mathbf{x}, \mathbf{x}'$, it follows that there is a nonzero chance that $C_0$ is injective on $\hat{\mathcal{X}}$. $\square$

There are two approaches to using Lemma E.1 for the sake of improving the $O(d_0^2 \cdot \log N)$ term in Theorem 3.1.

- One approach would be to start with a trivial layer that copies the input $\mathbf{x} \in \{0,1\}^{d_0}$ as well as computing all the negations of the bits of $\mathbf{x}$; then we have a layer that applies the function $C_0$ from Lemma E.1 (using negated variables to implement $-1$ weights); and then we continue with the network of Theorem 3.1. The net effect is that the depth has increased by two (so the network now has depth 16 instead of 14), and in the weights bound, the $O(d_0^2 \cdot \log N)$ term has been slightly improved to $O(d_0^2 + d_0^{3/2} \cdot \log N + d_0 \cdot \log^3 N)$.

- A second approach would be to change the model. If we permit ternary edge weights (i.e., weights in the set $\{-1, 0, 1\}$), then the function $C_0$ of Lemma E.1 can be implemented as the very first layer of our network, and then we can continue with the network of Theorem 3.1. Note that we need ternary edge weights only in the first layer; the edge weights in all subsequent layers are binary. The benefit of this approach is in the weights bound, the $O(d_0^2 \cdot \log N)$ term of Theorem 3.1 would be improved to $O(d_0^{3/2} \cdot \log N + d_0 \cdot \log^3 N)$.

## E.2 A $d_0^2$ Lower Bound on the Number of Weights

We now show that the $O(d_0^2 \cdot \log N)$ term in Theorem 3.1 cannot be improved to something better than $d_0^2$, if we insist on using the "binary threshold network" model. The argument is elementary.

**Proposition E.2** ($d_0^2$ wire lower bound). *For every $d_0 \in \mathbb{N}$, there exists a partial Boolean function $f \colon \{0,1\}^{d_0} \to \{0,1,\star\}$, defined on a domain $\hat{\mathcal{X}}$ of size $d_0 + 1$, such that for every binary threshold network $\tilde{h}$, if $\tilde{h}$ agrees with $f$ everywhere in its domain and $\underline{d}$ is the widths of $\tilde{h}$, then $w\,(\underline{d}) \geq d_0^2$.*

*Proof.* For each $i \in \{0, 1, \dots, d_0\}$, let $\mathbf{x}^{(i)}$ be the vector consisting of $i$ zeroes followed by $d_0 - i$ ones. Let $\hat{\mathcal{X}} = \{\mathbf{x}^{(i)} : 0 \leq i \leq d_0\}$, and let

$$f(\mathbf{x}) = \begin{cases} \mathsf{PARITY}(\mathbf{x}) & \text{if } \mathbf{x} \in \hat{\mathcal{X}} \\ \star & \text{otherwise.} \end{cases}$$

For the analysis, let $\tilde{h}$ be a fully connected binary threshold network that agrees with $f$ on all points in $\hat{\mathcal{X}}$. Consider the layer immediately following the input layer. Each node $g$ in this layer computes either a monotone Boolean function or an anti-monotone Boolean function of the input variables. Therefore, there is at most one value $i \in \{1, 2, \dots, d_0\}$ such that $g(\mathbf{x}^{(i-1)}) \neq g(\mathbf{x}^{(i)})$. On the other hand, for every $i \in \{1, 2, \dots, d_0\}$, we have $\tilde{h}(\mathbf{x}^{(i-1)}) \neq \tilde{h}(\mathbf{x}^{(i)})$, and hence there must be at least one node $g$ in this layer such that $g(\mathbf{x}^{(i-1)}) \neq g(\mathbf{x}^{(i)})$. Therefore, there are at least $d_0$ many nodes $g$.

Thus, the first two layers of $\tilde{h}$ both have widths of at least $d_0$, demonstrating that $\tilde{h}$ has at least $d_0^2$ many weights. $\qquad\square$

# F Generalization results (Proofs for Section 4)

Denote by $\mathcal{H}_{\underline{d}}^{\text{BTN}}$ the set of functions representable as binary threshold networks with dimensions $\underline{d}$ (given a fixed depth $L$). We start by bounding the cardinality $\left|\mathcal{H}_{\underline{d}}^{\text{BTN}}\right|$ in terms of the number of edges $w(\underline{d})$.

**Lemma F.1.** *Let $\underline{d}$ be the dimensions of a binary threshold network with $w \triangleq w(\underline{d})$ edges. Then there are $2^{w+O\left(\sqrt{w}\log(w)\right)}$ functions representable as networks with dimensions $\underline{d}$.*

*Proof.* We bound the number of function representable as binary threshold networks with dimensions $\underline{d}$ having $w$ edges by suggesting a way to encode them, and then bounding the number of bits in the encoding. First, permute each layer so the neurons are sorted by the bias and neuron scaling terms $(b_{li}, \gamma_{li})$. As NNs are invariant to permutations, this does not change the function. Now, at each layer we encode the bias term based on one of two encodings.

- If $d_l < d_{l-1}$, then list each of the bias terms as a number with $O(\log(d_{l-1}))$ bits plus 2 bits for the scaling term for a total of $O(d_l(\log(d_{l-1})+2)) \leq O(\sqrt{d_l d_{l-1}}\log(d_{l-1}))$, where the inequality is due to $d_l < d_{l-1}$.

- If $d_l \geq d_{l-1}$, then we encode the bias and scaling terms by listing the number of times each pair $(b_{li}, \gamma_{li}) \in \{-d_{l-1}, \ldots, d_{l-1} - 1\} \times \{-1, 0, 1\}$ appears in $(\mathbf{b}_l, \boldsymbol{\gamma}_l)$ (recall that the neurons are ordered according to these pairs). Each pair can appear at most $d_l$ times and so requires $O(\log(d_l))$ bits to encode for a total of $O(6d_{l-1}\log(d_l)) = O(d_{l-1}\log(d_l)) \leq O(\sqrt{d_l d_{l-1}}\log(d_l d_{l-1}))$.

By encoding each weight with a single bit, this means that for all layers, we can encode the weights, biases and scaling terms using $d_l d_{l-1} + O\left(\sqrt{d_l d_{l-1}}\log(d_l d_{l-1})\right)$ bits for a total of

$$
\sum_{l=1}^{L} d_l d_{l-1} + O\left(\sqrt{d_l d_{l-1}}\log(d_l d_{l-1})\right) = w + O\left(\sum_{l=1}^{L}\sqrt{d_l d_{l-1}}\log(d_l d_{l-1})\right)
$$

$$
\leq w + O\left(\sum_{l=1}^{L}\sqrt{d_l d_{l-1}}\log\left(\sum_{l=1}^{L} d_l d_{l-1}\right)\right)
$$

$$
\leq w + O\left(\sum_{l=1}^{L}\sqrt{d_l d_{l-1}}\log(w)\right) = w + O\left(\log(w)\cdot L\sum_{l=1}^{L}\frac{1}{L}\sqrt{d_l d_{l-1}}\right)
$$

$$
[\text{Jensen}] \leq w + O\left(\log(w)\cdot L\sqrt{\sum_{l=1}^{L}\frac{1}{L}d_l d_{l-1}}\right) = w + O\left(\log(w)\cdot\sqrt{L}\sqrt{\sum_{l=1}^{L} d_l d_{l-1}}\right)
$$

$$
= w + O\left(\log(w)\cdot\sqrt{L}\sqrt{w}\right)
$$

$$
= w + O\left(\log(w)\cdot\sqrt{w}\right).
$$

$\square$

**Corollary F.2.** *Assuming that the depth $L$ is fixed and known, a binary threshold network of depth $L$ with unknown number of weights $w$, can be encoded with $w + O(\sqrt{w}\log(w))$ bits.*

*Proof.* After specifying the architecture $\underline{d}$, from Lemma F.1 we require $w + O(\sqrt{w}\log(w))$ bits. Therefore it remains to bound the length of the encoding of $\underline{d}$. We first use $O(\log(w))$ bits to encode the number of weights, then, since $\underline{d} \in [w]^L$, we only need $O\left(\log\left(w^L\right)\right) = O(\log(w))$ additional bits for a total of $w + O(\sqrt{w}\log(w)) + O(\log(w)) = w + O(\sqrt{w}\log(w))$. $\square$

## F.1 Derivation of the min-size generalization bounds (Proofs for Section 4.1)

Throughout this subsection, we use $A(S)$ to denote the min-size interpolating NN of depth $L$, $A_L(S)$.

**Lemma F.3.** *Let $L \geq 16$ be fixed. Then*

$$I(S; A(S)) \leq w(\underline{d}^\star) + N \cdot H(\varepsilon^\star) + O(\delta(N, d_0, \underline{d}^\star))$$

*where*

$$\delta(N, d_0, \underline{d}^\star) = n(\underline{d}^\star) \cdot N^{3/4} H(\varepsilon^\star)^{3/4} \cdot \text{polylog}(N + n(\underline{d}^\star) + d_0)$$
$$+ d_0^2 \cdot \log N + d_0 n(\underline{d}^\star) \log(n(\underline{d}^\star) + N + d_0)^{3/2}.$$

*Proof.* Using Shannon's source coding theorem:

$$I(S; A(S)) \leq H(A(S)) \leq \mathbb{E}|A(S)|,$$

where $|A(S)|$ denotes the number of bits in the encoding of $A(S)$. Following Corollary 3.4, for a consistent $S$, $A(S)$ is a network with fixed depth and at most

$$w \triangleq w(\underline{d}^\star) + N \cdot H(\mathcal{L}_S(h^\star)) + 2n(\underline{d}^\star) N^{3/4} H(\mathcal{L}_S(h^\star))^{3/4} \text{polylog} N$$
$$+ O(d_0(d_0 + n(\underline{d}^\star)) \cdot \log N)$$

weights and therefore, using the result from Corollary F.2 and $\sqrt{w(\underline{d}^\star)} \leq d_0 + n(\underline{d}^\star)$,

$$|A(S)| \leq w + O(\sqrt{w}\log(w))$$
$$= w(\underline{d}^\star) + N \cdot H(\mathcal{L}_S(h^\star)) + O\left(n(\underline{d}^\star) \cdot N^{3/4} H(\mathcal{L}_S(h^\star))^{3/4} \cdot \text{polylog}(N + n(\underline{d}^\star) + d_0)\right)$$
$$+ O\left(d_0^2 \cdot \log N + d_0 n(\underline{d}^\star) \log(n(\underline{d}^\star) + N + d_0)^{3/2}\right)$$
$$= w(\underline{d}^\star) + N \cdot H(\mathcal{L}_S(h^\star)) + O\left(\tilde{\delta}(N, d_0, \underline{d}^\star)\right)$$

where we grouped all lower order terms in $\tilde{\delta}$. In case $S$ is inconsistent, $A(S) = \star$ so $|A(S)| = O(1)$. Taking the expected value and using Jensen's inequality gives

$$\mathbb{E}|A(S)| = \mathbb{E}[|A(S)| \cdot \mathbb{I}\{\text{inconsistent } S\}] + \mathbb{E}[|A(S)| \cdot \mathbb{I}\{\text{consistent } S\}]$$
$$\leq O(1) + \mathbb{E}\big[\underbrace{\mathbb{I}\{\text{consistent } S\}}_{\leq 1} \underbrace{\left(w(\underline{d}^\star) + N \cdot H(\mathcal{L}_S(h^\star)) + O\left(\tilde{\delta}(N, d_0, \underline{d}^\star)\right)\right)}_{\geq 0}\big]$$
$$\leq O(1) + \mathbb{E}\left[w(\underline{d}^\star) + N \cdot H(\mathcal{L}_S(h^\star)) + O\left(\tilde{\delta}(N, d_0, \underline{d}^\star)\right)\right]$$
$$[\text{Jensen}] \leq w(\underline{d}^\star) + N \cdot H(\mathbb{E}[\mathcal{L}_S(h^\star)]) + O(\delta(N, d_0, \underline{d}^\star))$$
$$= w(\underline{d}^\star) + N \cdot H(\varepsilon^\star) + O(\delta(N, d_0, \underline{d}^\star)).$$

$\square$

With this result, we are ready to derive the generalization results.

**Recall Theorem 4.2.** Consider a distribution $\mathcal{D}$ induced by a noisy teacher model of depth $L^\star$ and widths $\underline{d}^\star$ (Assumption 2.4) with a noise level of $\varepsilon^\star < 1/2$. Let $S \sim \mathcal{D}^N$ be a training set such that $N = o(\sqrt{1/\mathcal{D}_{\max}})$. Then, for any fixed depth $L \geq \max\{L^\star, 14\} + 2$, the generalization error of the min-size depth-$L$ NN interpolator satisfies the following.

- **Under arbitrary label noise,**

$$\mathbb{E}_S\left[\mathcal{L}_{\mathcal{D}}\left(A\left(S\right)\right)\right] \leq 1 - 2^{-H(\varepsilon^\star)/\mathbb{P}_S(\text{consistent } S)} + \mathbb{P}\left(\text{inconsistent } S\right) + O\left(C_{\min}\left(N, d_0, \underline{d}^\star\right)\right).$$

- **Under independent label noise,**

$$\left|\mathbb{E}_S\left[\mathcal{L}_{\mathcal{D}}\left(A\left(S\right)\right)\right] - 2\varepsilon^\star\left(1-\varepsilon^\star\right)\right|$$
$$\leq \left(1 - 2\varepsilon^\star\right)\sqrt{\frac{O(C_{\min}(N,d_0,\underline{d}^\star))+\mathbb{P}(\text{inconsistent } S)}{\mathbb{P}(\text{consistent } S)}} + \frac{\left(N-1\right)\mathcal{D}_{\max}}{3} + \mathbb{P}\left(\text{inconsistent } S\right),$$

where

$$C_{\min}\left(N, d_0, \underline{d}^\star\right) = \frac{w\left(\underline{d}^\star\right) + \delta\left(N, d_0, \underline{d}^\star\right)}{N}$$

with $\delta\left(N, d_0, \underline{d}^\star\right)$ as defined in Lemma F.3.

*Remark F.4.* The bound shown in Section 4.1 is found by bounding $\mathbb{P}(\text{inconsistent } S) \leq \frac{1}{2}N^2\mathcal{D}_{\max}$ as in Lemma B.1. Then using the Taylor approximation with small $N^2\mathcal{D}_{\max}$

$$1 - 2^{-\frac{H(\varepsilon^\star)}{\mathbb{P}(\text{consistent } S)}} \leq 1 - 2^{-\frac{H(\varepsilon^\star)}{1-\frac{1}{2}N^2\mathcal{D}_{\max}}}$$
$$= 1 - 2^{-H(\varepsilon^\star)\left(1+O\left(N^2\mathcal{D}_{\max}\right)\right)}$$
$$= 1 - 2^{-H(\varepsilon^\star)}\left(1 + O\left(N^2\mathcal{D}_{\max}\right)\right)$$
$$= 1 - 2^{-H(\varepsilon^\star)} + O\left(N^2\mathcal{D}_{\max}\right).$$

Lemma B.1 is used similarly to bound the error in the independent noise case. Assuming that $N = \omega\left(n\left(\underline{d}^\star\right)^4 H\left(\varepsilon^\star\right)^3 \text{polylog}\left(n\left(\underline{d}^\star\right)\right) + d_0^2 \log d_0\right)$ when $\varepsilon^\star > 0$ we can deduce that $N = \omega\left(w\left(\underline{d}^\star\right)\right)$ as well since

$$w\left(\underline{d}^\star\right) \leq \left(n\left(\underline{d}^\star\right) + d_0\right)^2 \leq 4\left(\max\left\{n\left(\underline{d}^\star\right), d_0\right\}\right)^2.$$

Together with $N = o\left(\sqrt{1/\mathcal{D}_{\max}}\right)$ we get the desired form of the bounds. Finally, note that when $\varepsilon^\star = 0$, the convergence rate of $\tilde{O}\left(1/N\right)$ instead of $\tilde{O}\left(1/\sqrt[4]{N}\right)$, where $\tilde{O}$ hides logarithmic terms arising as artifacts of our analysis, and dependence on other parameters such as the input dimension $d_0$.

*Proof.* Starting with the bound in the arbitrary noise setting, we combine C.2 with F.3

$$-\log\left(1 - \mathbb{E}_S\left[\mathcal{L}_{\mathcal{D}}\left(A\left(S\right)\right) \mid \text{consistent } S\right]\right) \leq \frac{I\left(S; A\left(S\right)\right)}{N \cdot \mathbb{P}_S\left(\text{consistent } S\right)}$$
$$\leq \frac{w\left(\underline{d}^\star\right) + N \cdot H\left(\varepsilon^\star\right) + O\left(\delta\left(N, d_0, \underline{d}^\star\right)\right)}{N \cdot \mathbb{P}_S\left(\text{consistent } S\right)}$$
$$= \frac{1}{\mathbb{P}_S\left(\text{consistent } S\right)} \cdot \left(H\left(\varepsilon^\star\right) + O\left(C_{\min}\left(N, d_0, \underline{d}^\star\right)\right)\right).$$

Rearranging the above inequality and recalling Remark C.1, we have,

$$\mathbb{E}_S\left[\mathcal{L}_{\mathcal{D}}\left(A\left(S\right)\right) \mid \text{consistent } S\right] \leq 1 - 2^{-\frac{H(\varepsilon^\star)}{\mathbb{P}_S(\text{consistent } S)} - O(C_{\min}(N,d_0,\underline{d}^\star))}.$$

Then, using Lemma A.6, we get,

$$\mathbb{E}_S\left[\mathcal{L}_\mathcal{D}\left(A\left(S\right)\right)\mid \text{consistent } S\right] \leq 1 - 2^{-\frac{H(\varepsilon^\star)}{\mathbb{P}_S(\text{consistent } S)}} + O\left(C_{\min}\left(N, d_0, \underline{d}^\star\right)\right).$$

The bound is derived using the following observation. Since for a RV $X$ in $[0,1]$ and a binary RV $Y$ we have

$$\mathbb{E}[X] = \mathbb{E}[X \mid Y]\underbrace{\mathbb{P}(Y)}_{\leq 1} + \underbrace{\mathbb{E}[X \mid \neg Y]}_{\leq 1}\mathbb{P}(\neg Y) \leq \mathbb{E}[X \mid Y] + \mathbb{P}[\neg Y],$$

we conclude the proof as

$$\mathbb{E}_S\left[\mathcal{L}_\mathcal{D}\left(A\left(S\right)\right)\right] \leq \mathbb{E}_S\left[\mathcal{L}_\mathcal{D}\left(A\left(S\right)\right)\mid \text{consistent } S\right] + \mathbb{P}\left(\text{inconsistent } S\right).$$

For the independent noise setting, we combine Lemma C.3 and Lemma F.3 to get

$$\left|\mathbb{E}_S\left[\mathcal{L}_\mathcal{D}\left(A\left(S\right)\right)\mid \text{consistent } S\right] - 2\varepsilon^\star\left(1 - \varepsilon^\star\right)\right|$$

$$\leq \left(1 - 2\varepsilon^\star\right)O\left(\sqrt{C\left(N\right)}\right) + \frac{\left(N-1\right)\mathcal{D}_{\max}}{3},$$

where

$$C\left(N\right) = \frac{I\left(S; A\left(S\right)\right) - N \cdot \left(H\left(\varepsilon^\star\right) - \mathbb{P}\left(\text{inconsistent } S\right)\right)}{N\left(1 - \mathbb{P}\left(\text{inconsistent } S\right)\right)}$$

$$\leq \frac{w(\underline{d}^\star)+N\cdot H(\varepsilon^\star)+O(\delta(N,d_0,\underline{d}^\star))-N\cdot(H(\varepsilon^\star)-\mathbb{P}(\text{inconsistent } S))}{N(1-\mathbb{P}(\text{inconsistent } S))}$$

$$= \frac{O\left(\frac{w(\underline{d}^\star)+\delta(N,d_0,\underline{d}^\star)}{N}\right) + \mathbb{P}\left(\text{inconsistent } S\right)}{\mathbb{P}\left(\text{consistent } S\right)}$$

$$= \frac{O\left(C_{\min}\left(N, d_0, \underline{d}^\star\right)\right) + \mathbb{P}\left(\text{inconsistent } S\right)}{\mathbb{P}\left(\text{consistent } S\right)}$$

Finally, using the inequality from Lemma C.4, we have,

$$\left|\mathbb{E}_{S,A(S)}\left[\mathcal{L}_\mathcal{D}\left(A\left(S\right)\right)\right] - 2\varepsilon^\star\left(1 - \varepsilon^\star\right)\right|$$

$$\leq \left|\mathbb{E}_{S,A(S)}\left[\mathcal{L}_\mathcal{D}\left(A\left(S\right)\right)\mid \text{consistent } S\right] - 2\varepsilon^\star\left(1 - \varepsilon^\star\right)\right| + \mathbb{P}(\text{inconsistent } S)$$

$$\square$$

## F.2 Derivation of the posterior sampling generalization bounds (Section 4.2)

**Lemma F.5.** *For the posterior sampling algorithm*

$$I\left(S; A\left(S\right)\right) \leq \mathbb{E}_S \left[\log\left(\frac{1}{p_S}\right) \middle| \text{consistent } S\right] \mathbb{P}_S\left(\text{consistent } S\right) + \frac{2}{e \ln 2} .$$

*Proof.* Recall the definition of the marginal distribution of the algorithm's output (a hypothesis $h$) is

$$d\nu\left(h\right) = \sum_s dp\left(s, h\right) ,$$

where $s$ are all possible realizations of a (training) sample of size $N$.

For $h = \star$, we have $d\nu\left(\star\right) = \mathbb{P}_S\left(\text{inconsistent } S\right)$.

For $h \neq \star$, since $\mathcal{L}_s\left(h\right) = 0$ implies that $s$ is consistent, we have

$$d\nu\left(h\right) \triangleq \sum_s dp\left(s, h\right) = \sum_s \frac{\mathbb{I}\left\{\mathcal{L}_s\left(h\right) = 0\right\}}{p_s} d\mathcal{P}\left(h\right) d\mathcal{D}^N\left(s\right)$$

$$= \sum_{s:p_s>0} \frac{\mathbb{I}\left\{\mathcal{L}_s\left(h\right) = 0\right\}}{p_s} d\mathcal{P}\left(h\right) d\mathcal{D}^N\left(s\right)$$

$$= d\mathcal{P}\left(h\right) \sum_{s:p_s>0} \frac{\mathbb{I}\left\{\mathcal{L}_s\left(h\right) = 0\right\}}{p_s} d\mathcal{D}^N\left(s\right)$$

$$= d\mathcal{P}\left(h\right) \mathbb{E}_{S \sim \mathcal{D}^N} \left[\frac{\mathbb{I}\left\{p_S > 0\right\}}{p_S} \mathbb{I}\left\{\mathcal{L}_S\left(h\right) = 0\right\}\right] .$$

where, for ease of notation, we use the convention that $\frac{\mathbb{I}\{p_s>0\}}{p_s} = 0$ when $p_s = 0$. Denoting

$$\pi\left(h\right) \triangleq \mathbb{E}_{S \sim \mathcal{D}^N} \left[\frac{\mathbb{I}\left\{p_S > 0\right\}}{p_S} \mathbb{I}\left\{\mathcal{L}_S\left(h\right) = 0\right\}\right] ,$$

we get

$$d\nu\left(h\right) = d\mathcal{P}\left(h\right) \pi\left(h\right) .$$

Notice that if there exists some $s \in \text{supp}\left(\mathcal{D}^N\right)$ such that $\mathcal{L}_s\left(h\right) = 0$ then $\pi\left(h\right) > 0$. Using the definition of the mutual information:

$$I\left(S; A\left(S\right)\right) = \sum_s \sum_{h \in \mathcal{H} \cup \{\star\}} dp\left(s, h\right) \log\left(\frac{dp\left(s, h\right)}{d\nu\left(h\right) d\mathcal{D}\left(s\right)}\right)$$

$$= \sum_{s:p_s=0} dp\left(s, \star\right) \log\left(\frac{dp\left(s, \star\right)}{d\nu\left(\star\right) d\mathcal{D}\left(s\right)}\right) + \sum_{s:p_s>0} \sum_{h \in \mathcal{H}} dp\left(s, h\right) \log\left(\frac{dp\left(s, h\right)}{d\nu\left(h\right) d\mathcal{D}\left(s\right)}\right)$$

$$= \sum_{s:p_s=0} d\mathcal{D}\left(s\right) \log\left(\frac{d\mathcal{D}\left(s\right)}{\mathbb{P}_S\left(\text{inconsistent } S\right) d\mathcal{D}\left(s\right)}\right) +$$

$$\sum_{s:p_s>0} \sum_{h:\mathcal{L}_s(h)=0} \frac{1}{p_s} d\mathcal{P}\left(h\right) d\mathcal{D}\left(s\right) \log\left(\frac{\frac{1}{p_s} d\mathcal{P}\left(h\right) d\mathcal{D}\left(s\right)}{d\mathcal{P}\left(h\right) \pi\left(h\right) d\mathcal{D}\left(s\right)}\right)$$

$$= \sum_{s:p_s=0} d\mathcal{D}\left(s\right) \log\left(\frac{1}{\mathbb{P}_S\left(\text{inconsistent } S\right)}\right) + \sum_{s:p_s>0} \sum_{h:\mathcal{L}_s(h)=0} \frac{1}{p_s} d\mathcal{P}\left(h\right) d\mathcal{D}\left(s\right) \log\left(\frac{1}{p_s \pi\left(h\right)}\right) .$$

Simplifying each term separately, the first sum immediately simplifies to

$$-\mathbb{P}_S\left(\text{inconsistent } S\right) \log\left(\mathbb{P}_S\left(\text{inconsistent } S\right)\right) \leq \frac{1}{e \ln 2} ,$$

and

$$\sum_{s:p_s>0} \sum_{h:\mathcal{L}_s(h)=0} \frac{1}{p_s} d\mathcal{P}(h) \, d\mathcal{D}(s) \log\left(\frac{1}{p_s \pi(h)}\right)$$

$$= -\sum_{s:p_s>0} \sum_{h:\mathcal{L}_s(h)=0} \frac{1}{p_s} d\mathcal{P}(h) \, d\mathcal{D}(s) \log(p_s) - \sum_{s:p_s>0} \sum_{h:\mathcal{L}_s(h)=0} \frac{1}{p_s} d\mathcal{P}(h) \, d\mathcal{D}(s) \log(\pi(h))$$

$$= -\sum_{s:p_s>0} \frac{1}{p_s} \log(p_s) \, d\mathcal{D}(s) \underbrace{\sum_{h:\mathcal{L}_s(h)=0} d\mathcal{P}(h)}_{=p_s}$$

$$\quad - \sum_{s:p_s>0} \sum_{h:\pi(h)>0} \frac{\mathbb{I}\{\mathcal{L}_s(h)=0\}}{p_s} d\mathcal{P}(h) \, d\mathcal{D}(s) \log(\pi(h))$$

$$= -\sum_{s:p_s>0} \frac{1}{p_s} \log(p_s) \, d\mathcal{D}(s) \, p_s - \sum_{h:\pi(h)>0} \log(\pi(h)) \, d\mathcal{P}(h) \underbrace{\sum_{s:p_s>0} \frac{\mathbb{I}\{\mathcal{L}_s(h)=0\}}{p_s} d\mathcal{D}(s)}_{=\pi(h)}$$

$$= -\sum_{s:p_s>0} \log(p_s) \, d\mathcal{D}(s) - \sum_{h:\pi(h)>0} \pi(h) \log(\pi(h)) \, d\mathcal{P}(h)$$

$$= -\mathbb{E}_S\left[\log(p_S)\, \mathbb{I}\{p_S>0\}\right] - \mathbb{E}_{h\sim\mathcal{P}}\left[\mathbb{I}\{\pi(h)>0\}\, \pi(h) \log(\pi(h))\right]$$

$$= \mathbb{E}_S\left[\log\left(\frac{1}{p_S}\right) \mid p_S>0\right] \mathbb{P}_S(p_S>0) + \underbrace{\mathbb{E}_{h\sim\mathcal{P}}\left[-\pi(h)\log(\pi(h))\, \mathbb{I}\{\pi(h)>0\}\right]}_{\leq 1/e\ln 2}$$

$$\leq \mathbb{E}_S\left[\log\left(\frac{1}{p_S}\right) \mid \text{consistent } S\right] \mathbb{P}_S(\text{consistent } S) + \frac{1}{e\ln 2}\,.$$

Putting all of this together,

$$I(S; A(S)) \leq \mathbb{E}_S\left[\log\left(\frac{1}{p_S}\right) \middle| \text{consistent } S\right] \mathbb{P}_S(\text{consistent } S) + \frac{2}{e\ln 2}\,.$$

$\qquad\qquad\qquad\qquad\qquad\qquad\qquad\qquad\qquad\qquad\qquad\qquad\qquad\qquad\qquad\qquad\qquad\qquad\qquad\qquad\square$

**Corollary F.6.** *The generalization of posterior sampling satisfies*

$$-\log\left(1 - \mathbb{E}_{S,A(S)}\left[\mathcal{L}_{\mathcal{D}}\left(A\left(S\right)\right) \mid \text{consistent } S\right]\right) \leq \frac{\mathbb{E}_S\left[\log\left(\frac{1}{p_S}\right)\Big|\text{consistent } S\right] + 3}{N}.$$

*Proof.* Combining Lemma C.2 and Lemma F.5 we get

$$I\left(S; A\left(S\right)\right) \geq -N\log\left(1 - \mathbb{E}_{S,A(S)}\left[\mathcal{L}_{\mathcal{D}}\left(A\left(S\right)\right) \mid \text{consistent } S\right]\right)\mathbb{P}_S\left(\text{consistent } S\right)$$

and

$$I\left(S; A\left(S\right)\right) \leq \mathbb{E}_S\left[\log\left(\frac{1}{p_S}\right)\Big|\text{consistent } S\right]\mathbb{P}_S\left(\text{consistent } S\right) + \frac{2}{e\ln 2}$$

so

$$-N\log\left(1 - \mathbb{E}_{S,A(S)}\left[\mathcal{L}_{\mathcal{D}}\left(A\left(S\right)\right) \mid \text{consistent } S\right]\right)\mathbb{P}_S\left(\text{consistent } S\right)$$

$$\leq \mathbb{E}_S\left[\log\left(\frac{1}{p_S}\right)\Big|\text{consistent } S\right]\mathbb{P}_S\left(\text{consistent } S\right) + \frac{2}{e\ln 2}$$

and finally, using $2/e\ln 2 \leq 1.5$ and recalling C.1 we get

$$-\log\left(1 - \mathbb{E}_{S,A(S)}\left[\mathcal{L}_{\mathcal{D}}\left(A\left(S\right)\right) \mid \text{consistent } S\right]\right) \leq \frac{\mathbb{E}_S\left[\log\left(\frac{1}{p_S}\right)\Big|\text{consistent } S\right] + 3}{N}.$$

$\square$

Let $\bar{h}$ be a network with depth $L$, dimensions $\bar{d}$, and parameters $\bar{\theta} = \{\bar{\mathbf{W}}_l, \bar{\mathbf{b}}_l, \bar{\gamma}_l\} \in \Theta^{\mathrm{BTN}}(\bar{d})$. Let $\underline{d} \geq \bar{d}$. Similar to $\Theta^{\mathrm{BTN}}(\underline{d}; h_1, h_2)$ introduced in Lemma D.26, let $\Theta^{\mathrm{BTN}}(\underline{d}; \bar{h}) \subset \Theta^{\mathrm{BTN}}(\underline{d})$ be the set of parameters $\theta$ that implement $\bar{h}$ by setting a subset of the parameters to be equal to $\bar{\theta}$, and zero the effect of redundant neurons by setting their bias and neuron scaling terms to be 0. This is illustrated in Figure 4. In particular, in our notation, $\Theta^{\mathrm{BTN}}(\underline{d}; h_1, h_2) = \Theta^{\mathrm{BTN}}(\underline{d}; h_1 \oplus h_2)$.

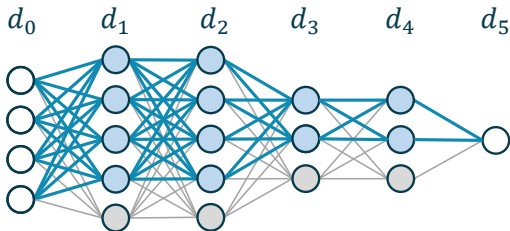

Figure 4: **Implementing a narrow network with a wider network.** Blue edges represent parameters set to equal the parameters of $\bar{h}$, gray nodes represent zero neuron scaling, and gray edges represent unconstrained parameters.

**Lemma F.7.** *Let $h$ be a network with depth $L$ and dimensions $\bar{d}$. Let $\underline{d} \geq \bar{d}$. Then*

$$-\log\left(\frac{|\Theta^{BTN}(\underline{d}; \bar{h})|}{|\Theta^{BTN}(\underline{d})|}\right) \leq w(\bar{d}) + O(n(\underline{d}) \cdot \log(\underline{d}_{\max} + d_0)).$$

*Proof.* We prove this by counting the number of constrained parameters in $\Theta^{\mathrm{BTN}}(\underline{d}; \bar{h})$. The number of constrained weights is

$$\bar{d}_1 d_0 + \sum_{l=2}^{L} \bar{d}_l \bar{d}_{l-1},$$

which is exactly $w(\bar{d})$. In addition, there are $n(\underline{d})$ constrained bias terms, and $n(\underline{d})$ constrained scaling terms. In total, after accounting for the quantization of each parameter, this means that

$$\frac{|\Theta^{\mathrm{BTN}}(\underline{d}; \bar{h})|}{|\Theta^{\mathrm{BTN}}(\underline{d})|} \geq \left(\underbrace{2^{w(\bar{d})}}_{\text{weights}} \cdot \underbrace{3^{n(\underline{d})}}_{\text{scaling terms}} \cdot \prod_{l=1}^{L}(2d_{l-1})^{d_l}\right)^{-1}$$

so

$$-\log\left(\frac{|\Theta^{\mathrm{BTN}}(\underline{d}; \bar{h})|}{|\Theta^{\mathrm{BTN}}(\underline{d})|}\right)$$

$$\leq w(\bar{d}) + n(\underline{d}) \cdot \log 3 + \sum_{l=1}^{L} \bar{d}_l \cdot \log(2d_{l-1})$$

$$\leq w(\bar{d}) + n(\underline{d}) \cdot \log 3 + n(\underline{d}) \cdot \log(2\underline{d}_{\max} + 2d_0)$$

$$= w(\bar{d}) + O(n(\underline{d}) \cdot \log(\underline{d}_{\max} + d_0)).$$

$\square$

Combining Lemma F.7 with Assumption 2.4, and Corollary 3.4 gives the following lemma.

**Lemma F.8.** *Consider a distribution $\mathcal{D}$ induced by a noisy teacher model of depth $L^\star$ and widths $\underline{d}^\star$ (Assumption 2.4) with a noise level of $\varepsilon^\star < 1/2$. Let $S \sim \mathcal{D}^N$ be a training set with effective training set label noise $\hat{\varepsilon}_{\mathrm{tr}}$ as defined in (4). Then there exist constants $c_1, c_2 > 0$ such that for any student network of depth $L \geq \max\{L^\star, 14\} + 2$ and widths $\underline{d} \in \mathbb{N}^L$ satisfying*

$$\forall l = 1, \ldots, L^\star - 1 \quad d_l \geq d_l^\star + N^{3/4} \cdot (\log N)^{c_1} + c_2 \cdot d_0 \cdot \log(N) \ ,$$

*it holds for posterior sampling with a uniform prior over parameters that*

$$\mathbb{E}_S\left[\log\left(\frac{1}{p_S}\right) \mid consistent\ S\right]$$
$$\leq w(\underline{d}^\star) + N \cdot H(\hat{\varepsilon}_{\mathrm{tr}}) + 2n(\underline{d}^\star) N^{3/4} \mathrm{polylog}N$$
$$+ O(d_0(d_0 + n(\underline{d}^\star)) \cdot \log(N) + n(\underline{d}) \cdot \log(\underline{d}_{\max} + d_0)) \ .$$

*Remark* F.9. Unlike the bounds for min-size interpolators, there is no $H(\hat{\varepsilon}_{\mathrm{tr}})^{3/4}$ term multiplying the $N^{3/4}$ term. This is because the architecture of random interpolators is fixed, so in our setting we must assume that it is wide enough in order to guarantee interpolation of any noisy training set.

*Proof.* Notice that for posterior sampling with uniform distribution over parameters, the interpolation probability $p_S$ can be lower bounded as

$$p_S \geq \frac{\left|\Theta^{\mathrm{BTN}}\left(\underline{d}; h^\star \oplus \tilde{h}_S\right)\right|}{\left|\Theta^{\mathrm{BTN}}(\underline{d})\right|}$$

and therefore

$$\log\left(\frac{1}{p_S}\right) \leq -\log\left(\frac{\left|\Theta^{\mathrm{BTN}}\left(\underline{d}; h^\star \oplus \tilde{h}_S\right)\right|}{\left|\Theta^{\mathrm{BTN}}(\underline{d})\right|}\right) \ .$$

Then, using the bounds from Lemma F.7 with the one from Corollary 3.4

$$\log\left(\frac{1}{p_S}\right) \leq w(\underline{d}^\star) + N \cdot H(\mathcal{L}_S(h^\star)) + 2n(\underline{d}^\star) N^{3/4} \mathrm{polylog}N$$
$$+ O(d_0(d_0 + n(\underline{d}^\star)) \cdot \log N + n(\underline{d}) \cdot \log(\underline{d}_{\max} + d_0)) \ .$$

By taking the expectation and using Jensen's inequality with the concave $H$ we arrive at

$$\mathbb{E}_S[H(\mathcal{L}_S(h^\star)) \mid consistent\ S] \leq H(\mathbb{E}_S[\mathcal{L}_S(h^\star) \mid consistent\ S]) = H(\hat{\varepsilon}_{\mathrm{tr}}) \ .$$

Hence

$$\mathbb{E}_S\left[\log\left(\frac{1}{p_S}\right) \mid consistent\ S\right]$$
$$\leq w(\underline{d}^\star) + N \cdot H(\hat{\varepsilon}_{\mathrm{tr}}) + 2n(\underline{d}^\star) N^{3/4} \mathrm{polylog}N$$
$$+ O(d_0(d_0 + n(\underline{d}^\star)) \cdot \log(N) + n(\underline{d}) \cdot \log(\underline{d}_{\max} + d_0)) \ .$$

$\square$

**Recall Theorem 4.4.** Consider a distribution $\mathcal{D}$ induced by a noisy teacher model of depth $L^\star$ and widths $\underline{d}^\star$ (Assumption 2.4) with a noise level of $\varepsilon^\star < 1/2$. Let $S \sim \mathcal{D}^N$ be a training set such that $N = o(\sqrt{1/\mathcal{D}_{\max}})$. Then, there exist constants $c_1, c_2 > 0$ such that for any student network of depth $L \geq \max\{L^\star, 14\} + 2$ and widths $\underline{d} \in \mathbb{N}^L$ holding

$$\forall l = 1, \ldots, L^\star - 1 \quad d_l \geq d_l^\star + N^{3/4} \cdot (\log N)^{c_1} + c_2 \cdot d_0 \cdot \log(N) , \tag{11}$$

the generalization error of posterior sampling satisfies the following.

- **Under arbitrary label noise,**

$$\mathbb{E}_{S,A(S)}\left[\mathcal{L}_{\mathcal{D}}\left(A\left(S\right)\right)\right] \leq 1 - 2^{-H(\varepsilon^\star)} + 2N^2 \mathcal{D}_{\max} + O\left(C_{\mathrm{rand}}\left(N\right)\right) .$$

- **Under independent label noise,**

$$\left|\mathbb{E}_{S,A(S)}\left[\mathcal{L}_{\mathcal{D}}\left(A\left(S\right)\right)\right] - 2\varepsilon^\star\left(1 - \varepsilon^\star\right)\right|$$

$$\leq \left(1 - 2\varepsilon^\star\right)\sqrt{\frac{O\left(C_{\mathrm{rand}}\left(N\right)\right) + \mathbb{P}\left(\text{inconsistent } S\right)}{\mathbb{P}\left(\text{consistent } S\right)}} + \frac{\left(N - 1\right)\mathcal{D}_{\max}}{3} + \mathbb{P}\left(\text{inconsistent } S\right) ,$$

where

$$C_{\mathrm{rand}}\left(N\right) = \frac{n\left(\underline{d}^\star\right) \cdot \mathrm{polylog}\left(N\right)}{\sqrt[4]{N}} + \frac{w\left(\underline{d}^\star\right) + d_0\left(d_0 + n\left(\underline{d}^\star\right)\right) \cdot \log\left(N\right) + n\left(\underline{d}\right) \cdot \log\left(d_{\max} + d_0\right)}{N} .$$

*Remark* F.10. The bound shown in Section 4.2 is found by bounding $\mathbb{P}\left(\text{inconsistent } S\right)$ as in Lemma B.1. Assuming that $N = \omega\left(n\left(\underline{d}^\star\right)^4 \mathrm{polylog}\left(n\left(\underline{d}^\star\right)\right) + d_0^2 \log d_0\right)$ we can deduce that $N = \omega\left(w\left(\underline{d}^\star\right)\right)$ as well since

$$w\left(\underline{d}^\star\right) \leq \left(n\left(\underline{d}^\star\right) + d_0\right)^2 \leq 4\left(\max\left\{n\left(\underline{d}^\star\right), d_0\right\}\right)^2 .$$

Together with $N = o\left(\sqrt{1/\mathcal{D}_{\max}}\right)$ we get the desired form of the bounds.

*Proof.* Corollary 3.4 implies that there exist $c_1, c_2 > 0$ such that a student NN satisfying (11) can interpolate any consistent dataset, and so posterior sampling is interpolating for all consistent datasets.

We start by proving the bound for arbitrary label noise. First, we notice that

$$\hat{\varepsilon}_{\mathrm{tr}} = \mathbb{P}(Y_1 \neq h^\star(X_1) \mid \text{consistent } S) = \frac{\mathbb{P}(Y_1 \neq h^\star(X_1), \text{consistent } S)}{\mathbb{P}\left(\text{consistent } S\right)}$$

$$\leq \frac{\mathbb{P}(Y_1 \neq h^\star(X_1))}{\mathbb{P}\left(\text{consistent } S\right)} = \frac{\varepsilon^\star}{\mathbb{P}\left(\text{consistent } S\right)} .$$

The entropy function $H$ is increasing in $\left[0, \frac{1}{2}\right]$ and achieves its maximum at $\frac{1}{2}$, so together with the inequality above, we get,

$$H\left(\hat{\varepsilon}_{\mathrm{tr}}\right) \leq H\left(\min\left\{\frac{\varepsilon^\star}{\mathbb{P}(\text{consistent } S)}, \frac{1}{2}\right\}\right) = H\left(\varepsilon^\star + \min\left\{\frac{\varepsilon^\star}{\mathbb{P}(\text{consistent } S)} - \varepsilon^\star, \frac{1}{2} - \varepsilon^\star\right\}\right)$$

$$= H\left(\varepsilon^\star + \underbrace{\min\left\{\varepsilon^\star \frac{\mathbb{P}(\text{inconsistent } S)}{\mathbb{P}(\text{consistent } S)}, \frac{1}{2} - \varepsilon^\star\right\}}_{\triangleq \Delta}\right) = H\left(\varepsilon^\star + \Delta\right) .$$

Employing the concavity of the entropy function, we get,

$$H\left(\hat{\varepsilon}_{\mathrm{tr}}\right) \leq H\left(\varepsilon^\star + \Delta\right) \leq H(\varepsilon^\star) + H'(\varepsilon^\star) \cdot \Delta \leq H(\varepsilon^\star) + \underbrace{H'(\varepsilon^\star) \cdot \varepsilon^\star}_{\leq \frac{1}{2}, \text{algebraically}} \cdot \frac{\mathbb{P}\left(\text{inconsistent } S\right)}{\mathbb{P}\left(\text{consistent } S\right)} .$$

By combining the above with Corollary F.6, Lemma F.8, we have that

$$-\log\left(1 - \mathbb{E}_{(S,A(S))}\left[\mathcal{L}_{\mathcal{D}}\left(A\left(S\right)\right) \mid \text{consistent } S\right]\right) \leq \frac{\mathbb{E}_S\left[\log\left(1/p_S\right) \mid \text{consistent } S\right] + 3}{N}$$

$$\leq H\left(\hat{\varepsilon}_{\text{tr}}\right) + \frac{1}{N}\left(w\left(\underline{d}^{\star}\right) + N \cdot H\left(\hat{\varepsilon}_{\text{tr}}\right) + 2n\left(\underline{d}^{\star}\right)N^{3/4}\text{polylog}N\right.$$

$$\left. + O\left(d_0\left(d_0 + n\left(\underline{d}^{\star}\right)\right) \cdot \log\left(N\right) + n\left(\underline{d}\right) \cdot \log\left(\underline{d}_{\max} + d_0\right)\right)\right)$$

$$\leq H\left(\hat{\varepsilon}_{\text{tr}}\right) + O\left(\frac{n(\underline{d}^{\star}) \cdot \text{polylog}(N)}{\sqrt[4]{N}} + \frac{w(\underline{d}^{\star}) + d_0(d_0 + n(\underline{d}^{\star})) \cdot \log(N) + n(\underline{d}) \cdot \log\left(\underline{d}_{\max} + d_0\right)}{N}\right)$$

$$\leq H(\varepsilon^{\star}) + \frac{\mathbb{P}(\text{inconsistent } S)}{2\mathbb{P}(\text{consistent } S)}$$

$$+ O\left(\frac{n(\underline{d}^{\star}) \cdot \text{polylog}(N)}{\sqrt[4]{N}} + \frac{w(\underline{d}^{\star}) + d_0(d_0 + n(\underline{d}^{\star})) \cdot \log(N) + n(\underline{d}) \cdot \log\left(\underline{d}_{\max} + d_0\right)}{N}\right)$$

$$= H(\varepsilon^{\star}) + \frac{\mathbb{P}(\text{inconsistent } S)}{2\mathbb{P}(\text{consistent } S)} + O\left(C_{\text{rand}}\left(N\right)\right).$$

Rearranging the inequality results in

$$\mathbb{E}_{(S,A(S))}\left[\mathcal{L}_{\mathcal{D}}\left(A\left(S\right)\right) \mid \text{consistent } S\right]$$

$$\leq 1 - 2^{-H(\varepsilon^{\star}) - \frac{\mathbb{P}(\text{inconsistent } S)}{2\mathbb{P}(\text{consistent } S)} - O(C_{\text{rand}}(N))}$$

Then, using Lemma A.6, we get,

$$\mathbb{E}_{(S,A(S))}\left[\mathcal{L}_{\mathcal{D}}\left(A\left(S\right)\right) \mid \text{consistent } S\right]$$

$$\leq 1 - 2^{-H(\varepsilon^{\star})} + \frac{\mathbb{P}\left(\text{inconsistent } S\right)}{2\mathbb{P}\left(\text{consistent } S\right)} + O\left(C_{\text{rand}}\left(N\right)\right).$$

Repeating the argument from the proof of Theorem 4.2, since for an RV $X$ in $[0,1]$ and a binary RV $Y$ we have

$$\mathbb{E}[X] = \mathbb{E}[X \mid Y]\underbrace{\mathbb{P}(Y)}_{\leq 1} + \underbrace{\mathbb{E}[X \mid \neg Y]}_{\leq 1}\mathbb{P}(\neg Y) \leq \mathbb{E}[X \mid Y] + \mathbb{P}(\neg Y),$$

we have,

$$\mathbb{E}_{(S,A(S))}\left[\mathcal{L}_{\mathcal{D}}\left(A\left(S\right)\right)\right] \leq \mathbb{E}_{(S,A(S))}\left[\mathcal{L}_{\mathcal{D}}\left(A\left(S\right)\right) \mid \text{consistent } S\right] + \mathbb{P}\left(\text{inconsistent } S\right)$$

$$\leq 1 - 2^{-H(\varepsilon^{\star})} + \frac{\mathbb{P}\left(\text{inconsistent } S\right)}{2\mathbb{P}\left(\text{consistent } S\right)} + \mathbb{P}\left(\text{inconsistent } S\right) + O\left(C_{\text{rand}}\left(N\right)\right)$$

$$\leq 1 - 2^{-H(\varepsilon^{\star})} + 2\frac{\mathbb{P}\left(\text{inconsistent } S\right)}{\mathbb{P}\left(\text{consistent } S\right)} + O\left(C_{\text{rand}}\left(N\right)\right)$$

$$\leq 1 - 2^{-H(\varepsilon^{\star})} + 2\frac{\frac{1}{2}N^2\mathcal{D}_{\max}}{1 - \frac{1}{2}N^2\mathcal{D}_{\max}} + O\left(C_{\text{rand}}\left(N\right)\right)$$

$$\leq 1 - 2^{-H(\varepsilon^{\star})} + 2N^2\mathcal{D}_{\max} + O\left(C_{\text{rand}}\left(N\right)\right)$$

where in the last inequality we used $t/\left(1 - t\right) \leq 2t$ for $t \in [0, 1/2]$.

Moving on to the independent noise setting, we combine Lemma F.5, Lemma F.8, and $\hat{\varepsilon}_{\text{tr}} \leq \varepsilon^{\star} < \frac{1}{2}$ from Lemma B.2, to bound the mutual information as

$$I\left(S; A\left(S\right)\right) \leq \mathbb{E}_S\left[\log\left(\frac{1}{p_S}\right) \mid \text{consistent } S\right]\overbrace{\mathbb{P}_S\left(\text{consistent } S\right)}^{\leq 1} + \frac{2}{e\ln 2}$$

$$\leq \mathbb{E}_S\left[\log\left(\frac{1}{p_S}\right) \mid \text{consistent } S\right] + 1.1$$

$$\leq N \cdot H\left(\varepsilon^{\star}\right) + O\left(N \cdot C_{\text{rand}}\left(N\right)\right).$$

Plugging the above into $C(N)$ of Lemma C.3, we get,

$$C\left(N\right) = \frac{I\left(S; A\left(S\right)\right) - N \cdot H\left(\varepsilon^{\star}\right) + N \cdot \mathbb{P}_{S \sim \mathcal{D}^N}\left(\text{inconsistent } S\right)}{N \cdot \mathbb{P}\left(\text{consistent } S\right)}$$

$$\leq \frac{N \cdot H(\varepsilon^{\star}) + O(N \cdot C_{\text{rand}}(N)) - N \cdot H(\varepsilon^{\star}) + N \cdot \mathbb{P}(\text{inconsistent } S)}{N \cdot \mathbb{P}(\text{consistent } S)}$$

$$= \frac{O\left(C_{\text{rand}}\left(N\right)\right) + \mathbb{P}\left(\text{inconsistent } S\right)}{\mathbb{P}\left(\text{consistent } S\right)}.$$

Then we continue as in the arbitrary noise setting to get the desired bound. $\qquad\square$

# G Alignment with Dale's Law

In this section, we show that our results apply to a model resembling "Dale's Law" [82], *i.e.,* such that for each neuron, all outgoing weights have the same sign. To this end, we define the following model, in which the main difference from Def. 2.1 is that neuron scaling is applied after the threshold activation.

**Definition G.1** (Binary threshold networks with outgoing scaling)**.** For a depth $L$, widths $\underline{d} = (d_1, \ldots, d_L)$, input dimension $d_0$, a scaled-neuron fully connected binary threshold NN with outgoing weight scaling (oBTN), is a mapping $\boldsymbol{\theta} \mapsto g_{\boldsymbol{\theta}}$ such that $g_{\boldsymbol{\theta}} : \{0, 1\}^{d_0} \to \{-1, 0, 1\}^{d_L}$, parameterized by

$$\boldsymbol{\theta} = \left\{ \mathbf{W}^{(l)}, \mathbf{b}^{(l)}, \boldsymbol{\gamma}^{(l)} \right\}_{l=1}^{L} ,$$

where for every layer $l \in [L]$,

$$\mathbf{W}^{(l)} \in \mathcal{Q}_l^W = \{0, 1\}^{d_l \times d_{l-1}} , \quad \boldsymbol{\gamma}^{(l)} \in \mathcal{Q}_l^{\gamma} = \{-1, 0, 1\}^{d_l} , \quad \mathbf{b}^{(l)} \in \mathcal{Q}_l^b = \{-d_{l-1} + 1, \ldots, d_{l-1}\}^{d_l} .$$

This mapping is defined recursively as $g_{\boldsymbol{\theta}}(\mathbf{x}) = g^{(L)}(\mathbf{x})$ where

$$g^{(0)}(\mathbf{x}) = \mathbf{x} ,$$
$$\forall l \in [L] \quad g^{(l)}(\mathbf{x}) = \boldsymbol{\gamma}^{(l)} \odot \mathbb{I} \left\{ \mathbf{W}^{(l)} g^{(l-1)}(\mathbf{x}) + \mathbf{b}^{(l)} > \mathbf{0} \right\} .$$

**Lemma G.2.** *Let $g_{\boldsymbol{\theta}}$ be an oBTN as in Def. G.1. Then there exists a BTN $h_{\boldsymbol{\theta}'}$ with the same dimensions and $\mathbf{b}'^{(l)} \in \mathcal{Q}_l^{2b} \triangleq \{-2d_{l-1} + 1, \ldots, 2d_l\}$ such that $h_{\boldsymbol{\theta}'} \equiv g_{\boldsymbol{\theta}} + s$ for $s \in \{0, 1\}^{d_L}$ such that for all $i = 1, \ldots, d_L$, $s_i = 1$ only if $\gamma_i^{(L)} = -1$.*

*Proof.* We prove the lemma by induction on depth. As we will see, the base case is a particular case of the step of the induction, so we start with the latter. Let $l = 1, \ldots, L$. For ease of notation, we denote $C = g^{(l)}$ and $A = g^{(l-1)}$, as well as $C' = h^{(l)}$, $A' = h^{(l-1)}$. In addition, we omit the superscripts from the $l^{th}$ layer's parameters. Let $i = 1, \ldots, d_l$, then by the induction hypothesis there exists some $a \in \{0, 1\}^{d_{l-1}}$ such that $A(\mathbf{x}) = A'(\mathbf{x}) - a$, for all inputs $\mathbf{x}$,

$$C(\mathbf{x})_i = \gamma_i \cdot \mathbb{I} \left\{ b_i + \sum_{j=1}^{d_{l-1}} w_{ij} A(\mathbf{x})_j > 0 \right\} = \gamma_i \cdot \mathbb{I} \left\{ b_i + \sum_{j=1}^{d_{l-1}} w_{ij} \left( A'(\mathbf{x})_j - a_j \right) > 0 \right\}$$
$$= \gamma_i \cdot \mathbb{I} \left\{ \left( b_i - \sum_{j=1}^{d_{l-1}} w_{ij} a_j \right) + \sum_{j=1}^{d_{l-1}} w_{ij} A'(\mathbf{x})_j > 0 \right\} .$$

If $\gamma_i = +1$, choose $\gamma_i' = +1$, and $b_i' = b_i - \sum_{j=1}^{d_{l-1}} w_{ij} a_j$. Clearly, since $w_{ij}, a_j \in \{0, 1\}$, it holds that $\left| \sum_{j=1}^{d_{l-1}} w_{ij} a_j \right| \leq d_{l-1}$ so $b_i' \in \mathcal{Q}_l^{2b}$. Then

$$C(\mathbf{x})_i = \mathbb{I} \left\{ b_i' + \gamma_i' \cdot \sum_{j=1}^{d_{l-1}} w_{ij} A'(\mathbf{x})_j > 0 \right\} = C'(\mathbf{x})_i$$

*i.e.*, the claim holds with $s_i = 0$. If $\gamma_i = -1$ then

$$
\begin{aligned}
C\left(\mathbf{x}\right)_i &= \gamma_i \cdot \mathbb{I}\left\{\left(b_i - \sum_{j=1}^{d_{l-1}} w_{ij} a_j\right) + \sum_{j=1}^{d_{l-1}} w_{ij} A'\left(\mathbf{x}\right)_j > 0\right\} \\
&= -\mathbb{I}\left\{\left(b_i - \sum_{j=1}^{d_{l-1}} w_{ij} a_j\right) + \sum_{j=1}^{d_{l-1}} w_{ij} A'\left(\mathbf{x}\right)_j > 0\right\} \\
&= -1 + \mathbb{I}\left\{\left(b_i - \sum_{j=1}^{d_{l-1}} w_{ij} a_j\right) + \sum_{j=1}^{d_{l-1}} w_{ij} A'\left(\mathbf{x}\right)_j \le 0\right\} \\
&= -1 + \mathbb{I}\left\{-\left(b_i - \sum_{j=1}^{d_{l-1}} w_{ij} a_j\right) - \sum_{j=1}^{d_{l-1}} w_{ij} A'\left(\mathbf{x}\right)_j \ge 0\right\} \\
&= -1 + \mathbb{I}\left\{1 - \left(b_i - \sum_{j=1}^{d_{l-1}} w_{ij} a_j\right) - \sum_{j=1}^{d_{l-1}} w_{ij} A'\left(\mathbf{x}\right)_j > 0\right\}.
\end{aligned}
$$

Thus, we can construct the $l^{th}$ layer of $h$ by choosing $\gamma_i' = -1$ and $b_i' = 1 - \left(b_i - \sum_{j=1}^{d_{l-1}} w_{ij} a_j\right)$ so

$$
\begin{aligned}
C\left(\mathbf{x}\right)_i &= -1 + \mathbb{I}\left\{1 - \left(b_i - \sum_{j=1}^{d_{l-1}} w_{ij} a_j\right) - \sum_{j=1}^{d_{l-1}} w_{ij} A'\left(\mathbf{x}\right)_j > 0\right\} \\
&= -1 + \mathbb{I}\left\{b_i' + \gamma_i' \sum_{j=1}^{d_{l-1}} w_{ij} A'\left(\mathbf{x}\right)_j > 0\right\} \\
&= C'\left(\mathbf{x}\right)_i - s_i
\end{aligned}
$$

with $s_i = 1$. Finally, if $\gamma_i = 0$, then $C_i$ is identically 0, so we can choose $\gamma_i' = b_i' = s_i = 0$. Notice that this construction also proves the base case $l = 1$ where $a = \mathbf{0}$. $\qquad\square$

**Corollary G.3.** *Let $\Theta'$ be the set of oBTN parameters such that for all $\boldsymbol{\theta} \in \Theta'$, $g_{\boldsymbol{\theta}} : \{0,1\}^{d_0} \to \{0,1\}^{d_L}$. Then there exists a BTN, $h$ as in Lemma G.2 such that $g_{\boldsymbol{\theta}} \equiv h$.*

*Proof.* Let $\boldsymbol{\theta} \in \Theta'$. Since $g_{\boldsymbol{\theta}}\left(\mathbf{x}\right) \ne -1$ for all $\mathbf{x}$, there exist parameters $\boldsymbol{\theta}' \in \Theta'$ such that $g_{\boldsymbol{\theta}'} \equiv g_{\boldsymbol{\theta}}$, and $\boldsymbol{\gamma}'^{(L)} \ge 0$. Hence, by Lemma G.2 there exists a BTN $h$ such that $h \equiv g_{\boldsymbol{\theta}'}$, *i.e.*, with $s = \mathbf{0}$. $\quad\square$

*Remark* G.4. Similar results can be shown in the other direction. That is, that BTNs can be represented as slightly larger oBTNs.

Finally, recall from Appendix F, that the cardinality of the hypothesis class is related to the error terms of Theorem 4.2 and Theorem 4.4 only logarithmically, meaning that we can apply the results to Def. G.1 without qualitatively changing them.

