# OpenReview forum: "Provable Tempered Overfitting of Minimal Nets and Typical Nets"
_NeurIPS.cc/2024/Conference — NeurIPS 2024 poster_

### Official Review · Reviewer_X2nX · 2024-06-12

**Soundness:** 3
**Presentation:** 2
**Contribution:** 2
**Rating:** 6
**Confidence:** 3

**Summary:**

This paper studies the tempered overfitting phenomenon in deep neural networks. The authors show the upper bound of test loss in both Min-size interpolators and random interpolators, which indicates the tempered overfitting.

**Strengths:**

The authors considered the quantized networks, and gave theoretical results on the tempered overfitting in DNNs without requiring the data dimension to be very low or very high.

**Weaknesses:**

a) I am concerned about the claim made in your contribution (Line 40) that "any noisy dataset can be interpolated using a neural network of constant depth with threshold activations and binary weights." Doesn't this only hold under specific data settings? I would appreciate it if the authors could comment on this point.

b) The mathematical symbols are used somewhat casually. For instance, (1) in line 94-95, the meaning of $A(S)$ and $\\mathcal{H}$; (2) in line 112, the meaning of $\tilde{h}\_{S} $. I recommend that the authors provide more explanations for the mathematical symbols, as this would strengthen the paper.

c) The motivation behind selecting quantized networks is unclear. Why were the networks quantized? Were the networks designed to fit the data model? I recommend that the authors provide some explanation for the motivation behind using quantized networks. In practice, few models use quantized networks.

d)  In line 180, the statement that $d\_0=o(\sqrt{N/\log(N)})$ requires the data dimension to be low. Does this contradict the previous comment (line 39) that the input dimension is not very low?

e) I recommend that the authors provide more explanation about Theorem 3.1 to help readers better understand it. I am not sure if I fully understand Theorem 3.1.

f) I am confused of the term $N^2 \mathcal{D}\_{max}$ in Theorem 4.1 and 4.3. Is this a small term? Can we find some conditions to ensure $N^2 \mathcal{D}\_{max}$ is small? It seems that $N^2 \mathcal{D}\_{max}$ loses the tightness.

g) In Remark 4.4, what's the meaning the dimension of $\underline{d}$ is small enough? Does it mean the network depth $L$ is small enough (constant level)?

**Questions:**

Generally speaking, incorporating synthetic experiments could further support the theory. Is it possible for the authors to include some synthetic experiments to validate the theoretical results? I am not suggesting that the authors are required to add experiments. I understand that this is primarily a theoretical analysis paper, and adding synthetic experiments seems difficult.  Therefore, I am asking the authors to consider the possibility of adding synthetic experiments.

**Limitations:**

See weakness and questions above.

---

> ### Author Rebuttal · Authors · 2024-08-07
>
> We thank the reviewer for the careful reading and constructive feedback.
>
> Below we address the reviewer’s concerns and explain how they’ll help us improve our paper.
>
> > a) I am concerned about the claim made in your contribution (Line 40) ... Doesn't this only hold under specific data settings? I would appreciate it if the authors could comment on this point.
>
> Indeed, this was not clear. We refer to interpolation of training sets with binary input features which labels are generated by a constant-size teacher model with label noise (as specified in Section 2). We will rephrase this line to clarify this.
>
> > b) The mathematical symbols are used somewhat casually. For instance, (1) ... $A(S)$ and $\mathcal{H}$; (2) ... $\tilde{h}_S$. I recommend that the authors provide more explanations for the mathematical symbols, as this would strengthen the paper.
>
> Thank you, we will make sure to explain the mathematical symbols and emphasize their role.
>
> > c) The motivation behind selecting quantized networks is unclear ... I recommend that the authors provide some explanation for the motivation behind using quantized networks. In practice, few models use quantized networks.
>
> From a theoretical point of view, the focus on quantized models simplifies the analysis considerably (e.g. we follow a similar proof technique for random interpolators as in [A], which also discussed quantized and continuous neural networks). We don’t find this assumption too restrictive as most continuous models can be approximated by quantized networks with sufficiently-fine quantization, under appropriate assumptions on the data (e.g. see [B]).
>
> The restriction to networks with binary weights is stronger, and is due to Theorem 3.1 which is the main technical tool of the paper, and for which we do not have a sufficiently tight analog for networks with higher quantization levels. As we note in Section 6 (line 241), some of our results can be extended to networks with higher quantization at the cost of looser bounds. We will explicitly write them in the paper.
>
> Moreover, from a practical point of view, all practical models are quantized to some extent (e.g., using float32) and there is a vast literature which studies NNs with lower numerical precision (e.g. see [C]), some even using as few as 1 bit [D]. Numerical precision has a large impact on efficiency (memory and computation), and thus, using a lower numerical precision is a very popular method to improve this efficiency.
>
> We will clarify these important points in the paper.
>
> [A] G. Buzaglo, I. Harel, M. S. Nacson, A. Brutzkus, N. Srebro, and D. Soudry. How uniform random weights induce non-uniform bias: Typical interpolating neural networks generalize with narrow teachers. In International Conference on Machine Learning (ICML), 2024.
>
> [B] Ding Y, Liu J, Xiong J, Shi Y. On the universal approximability and complexity bounds of quantized relu neural networks. arXiv preprint arXiv:1802.03646. 2018 Feb 10.
>
> [C] Gholami A, Kim S, Dong Z, Yao Z, Mahoney MW, Keutzer K. A survey of quantization methods for efficient neural network inference. In Low-Power Computer Vision 2022 Feb 22 (pp. 291-326). Chapman and Hall/CRC.
>
> [D] Courbariaux M, Hubara I, Soudry D, El-Yaniv R, Bengio Y. Binarized neural networks: Training deep neural networks with weights and activations constrained to+ 1 or-1. arXiv preprint arXiv:1602.02830. 2016 Feb 9.
>
> > d) In line 180, the statement that $d_0 = o (\sqrt{N / \log N})$ requires the data dimension to be low. Does this contradict the previous comment (line 39) that the input dimension is not very low?
>
> The input dimension is indeed bounded, but can still be large (e.g., $O (N^\gamma)$ for any $\gamma < 1 / 2$). An upper bound is to be expected, as the size of neural networks depends on the input dimension. This does *not* contradict the “very low” dimension statement in line 39, which refers to the references [42,47] cited in the related works section, which proved tempered overfitting in the case of $d_0=1$. We will clarify this.
>
> > e) I recommend that the authors provide more explanation about Theorem 3.1 to help readers better understand it. I am not sure if I fully understand Theorem 3.1.
>
> The main takeaway from Theorem 3.1 is that label flips can be memorized with networks with a number of parameters that is optimal in the leading order $N \cdot L_S (h^\star)$, i.e., not far from the minimal information-theoretical value.
> We will emphasize this.
>
> > f) I am confused of the term $N^2 D_{max}$ in Theorem 4.1 and 4.3. Is this a small term? Can we find some conditions to ensure $N^2 D_{max}$ is small? It seems that $N^2 D_{max}$ loses the tightness.
>
> We use $N^2 D_{\max}$ as a bound on the probability to sample an inconsistent training set, which is required to be small in our analysis.
> We find that the assumption that $N^2 D_{\max}$ is small is very reasonable, as the size of the input space is *exponential* in the input dimension. For example, if we assume that $D$ is a uniform distribution over all inputs in dimension $d_0$, then $D_{\max}=2^{-d_0}$. We tried to explain this in footnote 3 on page 6, and Remark 4.4 about the required relationship between $d_0$ and $N$, but we now see that it is not clear enough, so we will clarify it further.
>
> > g) In Remark 4.4, what's the meaning the dimension of $\underline{d}$ is small enough? Does it mean the network depth $L$ is small enough (constant level)?
>
> The $O ( n(\underline{d}) \log (d_{\max}) )$ error term in Theorem 4.3 vanishes when $n(\underline{d}) \log (\underline{d}_{\max}) = o (N)$ (what we referred to by “small enough”) thus making our bounds meaningful.
>
> Regarding $L$, throughout the paper we assume a constant depth, but our results can be extended to variable depth. This is simple in the posterior sampling case (see Remark D.25 and the following paragraph, lines 1059-1063), and more subtle in the min-size setting as it changes the learning rule.

---

> ### Comment · Reviewer_X2nX · 2024-08-08
>
> Thanks for your detailed response. I am generally satisfied with your response, and I will raise the score to 6.

---

### Official Review · Reviewer_E8Kb · 2024-07-17

**Soundness:** 3
**Presentation:** 3
**Contribution:** 3
**Rating:** 7
**Confidence:** 3

**Summary:**

This paper investigates the overfitting behavior of fully connected Deep Neural Networks (DNNs) with binary weights fitted to perfectly classify a noisy training set. The authors analyze the interpolation using both the smallest DNN (having the minimal number of weights) and a random interpolating DNN. They prove that overfitting is tempered for both learning rules. The analysis is based on a new bound on the size of a threshold circuit consistent with a partial function.

**Strengths:**

1) This paper presents innovative research on the overfitting behavior of DNNs with binary weights, which adds to the existing literature on the generalization capabilities of neural networks.
2) The analysis is based on a new bound on the size of a threshold circuit consistent with a partial function, which provides a theoretical foundation for the results.
3) This paper provides a clear and well-structured explanation of the research problem and methodology, making it easy for readers to follow.

**Weaknesses:**

1) This work focuses on the learning theory and presents the first theoretical results on benign or mitigated overfitting. It would be better if authors could provide necessary evaluation studies to support theoretical results.
2) This paper could benefit from a more thorough discussion of the limitations and future directions of the research.

**Questions:**

1. What are the limitations of your research? Are there any particular scenarios or datasets where your method may not perform well? It would be helpful to have a more thorough discussion of the limitations and potential future directions of the research.

**Limitations:**

The authors addressed the limitations.

---

> ### Author Rebuttal · Authors · 2024-08-07
>
> We thank the reviewer for the careful reading and valuable feedback.
>
> **Regarding Weakness 1:**
> > This work focuses on the learning theory and presents the first theoretical results on benign or mitigated overfitting. It would be better if authors could provide necessary evaluation studies to support theoretical results.
>
> Kindly see our comment on this in the general rebuttal response.
> Briefly, our results in the independent setting agree with the linear behavior observed empirically in Mallinar et al. (2022). Following the question from the reviewer, we will explain this in the revised manuscript.
>
> **Regarding Question 1:**
> > What are the limitations of your research? Are there any particular scenarios or datasets where your method may not perform well? It would be helpful to have a more thorough discussion of the limitations and potential future directions of the research.
>
> We specify the setting in which our results apply in Section 2, and the assumptions under which our results apply in the appropriate theorems/lemmas. In short, we assume binary input features, and labels that are generated by some fixed teacher model with added noise. In the model, we assume binary $\{0, 1\}$ weights with threshold activations and $\{-1, 0, 1\}$ neuron scaling. In Theorems 4.1 and 4.3, our assumptions are (1) that the data is consistent with high probability, (2) that the training set is large enough to make our bounds non-trivial, and (3) that the model is large enough to guarantee interpolation of consistent training sets.
>
> While our results agree with previous empirical works in more realistic settings (see our general comment referenced previously), we do not claim that our theoretical results apply to them. The adaptation for scenarios other than those described in Section 2 is left for future work. We will make this point clearer in the discussion of limitations and future work in Section 6.

---

> > ### Comment · Reviewer_E8Kb · 2024-08-12
> >
> > Thanks for your response. I'll keep my score.

---

### Official Review · Reviewer_q6NB · 2024-07-21

**Soundness:** 3
**Presentation:** 3
**Contribution:** 3
**Rating:** 7
**Confidence:** 3

**Summary:**

The paper proves that both minimum size and random interpolators exhibit tempered overfitting in the case of binary classification using a threshold network with binary weights on a noisy training set.

**Strengths:**

The paper proves tempered overfitting for minimum size and random interpolators in the presence of label noise. This can be seen as an extension of previous analysis, which considers settings without label noise, or other learning rules (e.g. shortest description length of Turing machines). The proof of the results relies on a new construction of a relatively small neural network that can learn arbitrary label noise.

The paper is well-written and easy to follow.

**Weaknesses:**

The derived bounds require some additional constraints on the width of the network and the number of examples due to the presence of label noise. Such requirements may not always be satisfied in practice.

**Questions:**

I wonder if there is any way one could empirically verify the tightness of the proven results, even for small networks.

**Limitations:**

Limitations are addressed.

---

> ### Author Rebuttal · Authors · 2024-08-07
>
> We thank the reviewer for their feedback.
>
> > I wonder if there is any way one could empirically verify the tightness of the proven results, even for small networks.
>
> Kindly see our comment on this in the general rebuttal response.
> Briefly, our results in the independent setting agree with the linear behavior observed empirically in Mallinar et al. (2022). Following the question from the reviewer, we will explain this in the revised manuscript.

---

> > ### Comment · Reviewer_q6NB · 2024-08-07
> >
> > I have read the reviewers global and individual response and acknowledge their point about empirical results.

---

### Official Review · Reviewer_gENy · 2024-07-30

**Soundness:** 3
**Presentation:** 2
**Contribution:** 3
**Rating:** 5
**Confidence:** 4

**Summary:**

The paper studies the generalization ability of the interpolated minimal quantized nets. It considers the task of classification with a binary sequence input using Quantized neural networks. It establishes the generalization error of the minimal-width neural network that interpolates the dataset and claims that it belongs to the tempered overfitting regime. Overall the paper is well-written and technically solid.

**Strengths:**

(a) The paper presents a novel theoretical analysis of overfitting behaviors in neural networks with binary weights. The new approximation result Theorem 3.1 and generalization result Theorem 4.1 are all novel to some extent.

(b) The paper is well-structured, with clear definitions and thorough explanations of the models and assumptions used.

**Weaknesses:**

(a) I'm not sure how the theoretical results derived in this paper will provide any insights into how people understand interpolated neural networks even in theory. It seems that this paper just plugged in the new approximation result Theorem 3.1 in the framework of (Manoj & Srebro, 2023). In other words, the paper provides marginal new information in this regard. The main theorem Theorem 4.1 is somewhat can be expected.

(b) The paper seems to combine everything in a whole, complex theorem and the reader cannot get any insights beyond the conclusion in Theorem 4.1, and the presented results are the same and can be expected after reading (Manoj & Srebro, 2023).

(c) I strongly recommend another round of refactoring to improve the readability of the paper, for example

 i) I think in the statement, the term $2^{H(\epsilon^\star)}$ should be  $2^{-H(\epsilon^\star)}$, given the usual definition of entropy.

 ii) the notations are heavy, which prevents potential audience.

 iii) some inconsistency in notations. For example, it defines the weights in the notation section, but it uses ``edges'' in the theorem statement.

**Questions:**

(a) The first thing I'm concerned about is the definition of a consistent dataset. Does introducing the consistent dataset just make the definition of interpolator well-defined, or the proof will depend heavily on the fact that the dataset is consistent? For example, if I define the interpolator $\hat{h}$ to be the one such that $L_S(\hat{h}) = \inf_{h} {L}_S(h)$ can one expect a result similar to line 144 with the term containing N^2 D\_{max} be removed?

(b) The paper establishes the asymptotic risk of the interpolator, which is good. In the noiseless setting, the risk converges to 0 at the rate of $N^{-1/4}$ if the teacher network is fixed, this is slower than the standard parametric rate $N^{-1}$. Is this because of the fundamental limits of the proposed estimator such that the error bound is tight, or the analysis can be improved?

(c) The result of Theorem 3.1 is for a fixed-depth network. Can similar, matching results hold for the case with varying depth L and width N networks?

**Limitations:**

The authors have adequately addressed the limitations.

---

> ### Author Rebuttal · Authors · 2024-08-07
>
> We thank the reviewer for the detailed constructive feedback.
>
> Below we address the reviewer’s concerns and explain how they’ll help us improve our paper.
>
> > (Weak.a) I'm not sure how the theoretical results … provide any insights into how people understand interpolated neural networks ... In other words, the paper provides marginal new information in this regard. The main theorem Theorem 4.1 is somewhat can be expected.
>
> Our general framework is indeed similar to the one in M&S (2023). The main conceptual novelty in our paper is that we show this framework can be adapted to
>
> (1) Neural networks (using Theorem 3.1 which is the main novel technical result) and
>
> (2) Random interpolators (building upon the approach from Buzaglo et al. 2024).
>
> We agree that direct implications of our results to more practical settings (e.g., networks trained with SGD) require further investigation, but we think that these results are interesting in and of themselves as well, as they predict the same tempered behavior observed in practice (see our explanation in the general comment on the fact that our independent setting results in the linear behavior observed in Mallinar et al. (2022); we will explicitly mention this in the revised paper).
>
> > (Weak.b) The paper seems to combine everything in a whole, complex theorem and the reader cannot get any insights beyond the conclusion in Theorem 4.1, and the presented results are the same and can be expected after reading (Manoj & Srebro, 2023).
>
> While our results can perhaps be conjectured after reading M&S (2023), they are not trivial, and require the novel memorization result (Theorem 3.1), as well as additional adaptations to match our setting (e.g., when the input dimension is finite).
>
>
> > (Ques.a) The first thing I'm concerned about is the definition of a consistent dataset. Does introducing the consistent dataset just make the definition of interpolator well-defined, or the proof will depend heavily on the fact that the dataset is consistent? For example, if I define the interpolator $\hat{h}$ to be the one such that $L_S (\hat{h}) = \inf_{h} L_S (h)$ can one expect a result similar to line 144 with the term containing $N^2 D_{max}$ be removed?
>
> As in previous work, we aim to study the behavior of interpolators, and thus must require the consistency of the training set (otherwise it’s impossible to interpolate). The $N^2 D_{\max}$ term serves as a bound on the probability that the data is inconsistent, and is needed due to the fact that we examine a setting with a finite sample space.
>
> Importantly, as currently phrased, Framework 1 *allows* defining the interpolator $L_S (\hat{h}) = \inf_h L_S (h)$ as suggested by the reviewer, but our analysis requires that the value of this minimal loss equals 0 with high probability. We will explicitly explain this after presenting the framework.
> We agree that it is interesting to study the suggested learning rule, $L_S (\hat{h}) = \inf_h L_S (h)$, when the inconsistency probability is not small, and we explicitly mention it in the future work section.
>
> > (Ques.b) The paper establishes the asymptotic risk of the interpolator, which is good. In the noiseless setting, the risk converges to 0 at the rate of $N^{-1/4}$ if the teacher network is fixed, this is slower than the standard parametric rate $N^{-1}$. Is this because of the fundamental limits of the proposed estimator such that the error bound is tight, or the analysis can be improved?
>
> The $N^{-1/4}$ term comes from the lower order $N^{3/4}$ term in Corollary 3.3, and the fact that we omitted for simplicity an $H(L_S (h^\star))$ scaling term in the proof of the corollary (see the proof of Corollary 3.3 in page 43, lines 1095 and 1096). In fact, by reintroducing the entropy term with $L_S (h^\star) = 0$, the $O (1 / N)$ bound in the noiseless setting can be recovered from our results up to a logarithmic term. This is expected, as, without the noise-memorizing-network, our bounds depend only on the teacher model (up to logarithmic error terms accumulated in the analysis).
> There is a slight caveat in the random interpolator model — with a positive error rate, we require the width of the network to be at least $N^{3/4}$ in order to guarantee interpolation of any consistent training set (with any number of label flips), so the $N^{-1/4}$ rate in the bound remains. To get the desired $N^{-1}$ rate, we must assume that the width of the network is constant w.r.t N, which is reasonable if we know that there is no noise.
>
> We will clarify this in the paper.
>
> > (Ques.c) The result of Theorem 3.1 is for a fixed-depth network. Can similar results hold for the case with varying depth L and width N networks?
>
> Indeed, in Theorem 3.1, the $3/4$ exponent can be improved to roughly $2/3$ at the cost of an increased depth (see Remark D.7, page 30). That being said, we don’t know the *optimal* dimensions of an interpolating network, as our approach — calculating the XOR between the teacher and label flip memorizing networks — may not be optimal. Therefore, we leave this interesting question for future research. However, we *do* know it is not possible to decrease the max-width bound in Theorem 3.1 below $\sqrt{N}$, assuming the depth is a constant. In such cases, the total number of parameters will be too small to be able to interpolate an arbitrary configuration of label flips.
>
> Minor note: in the question the reviewer used $N$ to denote width, but the paper uses $N$ to denote the number of samples. We assumed this was unintentional (as width $N$ does not not make much sense in our approach).
>
> Finally, like the reviewer suggested, we will make another pass to improve the paper’s readability even further. Specifically, we will fix the missing minus sign and make sure to unify the terminology (e.g., edges / weights). Additionally, we will try to unburden the notations as much as possible, and verbally re-mention their ‘role’ as we use them throughout the paper.

---

> > ### Comment · Reviewer_gENy · 2024-08-11
> >
> > Thanks for your response. I'll keep my score.

---

> > > ### Author Response · Authors · 2024-08-12
> > >
> > > We thank the reviewer for the response. Could the reviewer kindly let us know if there are any remaining concerns we should address?

---

### Author Rebuttal · Authors · 2024-08-07

We would like to thank all reviewers for their constructive and valuable comments which we will use to improve our paper in its revised version.

As the subject of **empirically validating** our results appeared in multiple reviews, we address it here in this general comment.

Finding min-size interpolators and random interpolators is computationally hard. Therefore, only very small-scale experiments are feasible when directly testing our results (see for example [A] as an empirical work on random interpolators). But such small-scale experiments are typically *inadequate* to test *asymptotic* results, as we derived.

While working on our author response, we realized that our theoretical results already match very well the empirically-observed results from [B] (which was the original paper that observed the “tempered overfitting” behavior in a practical setting, i.e., using standard SGD training on various datasets). Specifically, denote by $D$ a noiseless distribution over features and labels (without label noise), and by $D_{\varepsilon^\star}$ the corrupted distribution with independent label flip noise with error rate $\varepsilon^\star$. Then, Mallinar et al. [B] found empirically that, in standard binary classification settings with interpolating neural networks, the clean test error is approximately the training set’s error rate, $L_D (h) \approx \varepsilon^\star$ (see Figures 2, 3, and 6 in [B]). In the independent noise setting (used in [B]) it is easy to show that
$$
L_{D_{\varepsilon^\star}} = \varepsilon^\star + (1 - 2 \varepsilon^\star) L_{D} (h) ,
$$
so plugging in $L_{D} (h) \approx \varepsilon^\star$ implies that $L_{D_{\varepsilon^\star}} \approx 2 \varepsilon^\star (1 - \varepsilon^\star)$, and so, their empirical $L_D (h)$ behaves exactly as our analysis dictates in Theorems 4.1 and 4.3.

Thank you for making us realize this very important point. We will add this discussion to the revised paper.


[A] Chiang PY, Ni R, Miller DY, Bansal A, Geiping J, Goldblum M, Goldstein T. Loss landscapes are all you need: Neural network generalization can be explained without the implicit bias of gradient descent. InThe Eleventh International Conference on Learning Representations 2022 Sep 29.

[B]  N. R. Mallinar, J. B. Simon, A. Abedsoltan, P. Pandit, M. Belkin, and P. Nakkiran. Benign,
tempered, or catastrophic: Toward a refined taxonomy of overfitting. In Advances in Neural
Information Processing Systems, 2022.

---

### Decision · Program_Chairs · 2024-09-25

**Decision:**

Accept (poster)

**Comment:**

This paper investigates the overfitting behavior of fully-connected binary neural networks with threshold activation, specifically networks trained/sampled to perfectly interpolate a noisy dataset. The paper demonstrates that overfitting is tempered, meaning that the interpolation is not optimal but still much better than trivial.

Overall, there is a consensus on the quality and relevance of the paper.

The reviewers have outlined that the paper is well-written and clear (*gENy*, *q6NB*, *E8Kb*). The results derived for neural network interpolators, specifically in the context of label noise and without imposing unrealistic constraints on input dimensions, have been identified as novel (*gENy*, *q6NB*, *E8Kb*, *X2nX*). The result on the memorization of random noise by binary threshold networks have also been highlighted as novel and valuable (*gENy*, *q6NB*, *E8Kb*).

This is a strong work, and I recommend acceptance.

The authors are encouraged to address the recommended proofreading checks (*gENy*) and to incorporate the clarifications they provided in their rebuttal, particularly regarding the reasoning behind the empirical validation discussed in their global response.